# PRIVATE OVERPARAMETERIZED LINEAR REGRESSION WITHOUT SUFFERING IN HIGH DIMENSIONS

## ABSTRACT

This study focuses on differentially private linear regression in the over-parameterized regime. We propose a new variant of the differentially private Follow-The-Regularized-Leader (DP-FTRL) algorithm that uses a random noise with a general covariance matrix for differential privacy. This leads to improved privacy and utility (excess risk) trade-offs. Firstly, even when reduced to an existing DP-FTRL algorithm that uses an isotropic noise, our excess risk bound is sharper as a function of the eigenspectrum of the data covariance matrix and the ground truth model parameter. Furthermore, when unlabeled public data is available, we can design a better noise covariance matrix structure to improve the utility. For example, when the ground truth has a bounded $\ell_2$-norm, and the eigenspectrum decays polynomially (i.e., $\lambda_i = i^{-r}$ for $r > 1$), our method achieves $\widetilde{\mathcal{O}}(N^{-\frac{r}{1+2r}})$ and $\widetilde{\mathcal{O}}(N^{-\frac{r}{3+r} \wedge \frac{2r}{1+3r}})$ excess error for identity and specially designed covariance matrices, respectively. Notably, our method with a specially designed covariance matrix outperforms the one with an identity matrix when the eigenspectrum decays at least quadratically fast, i.e., $r \geq 2$. Our proposed method significantly improves upon existing differentially private methods for linear regression, which tend to scale with the problem dimension, leading to a vacuous guarantee in the over-parameterized regime.

## 1 INTRODUCTION

In recent years, machine learning has witnessed remarkable achievements across various domains, such as finance and health care. The rapid advancements in large language models (e.g., ChatGPT (Brown et al., 2020)) have further accelerated the use of machine learning techniques in our daily lives. While these techniques offer numerous benefits, there is an increasing concern regarding the privacy of sensitive personal information in the datasets used for training machine learning models (Fredrikson et al., 2015; Shokri et al., 2017). For instance, over-parameterized neural networks can memorize sensitive training data, posing significant privacy risks when deployed (Carlini et al., 2019; 2021). Extensive research has been conducted on privacy-preserving machine learning to mitigate these privacy concerns, primarily focusing on achieving differential privacy (DP) (Dwork et al., 2006). DP is a rigorous definition of privacy that protects each individual's privacy from adversaries who can access the models and potentially the rest of the data.

Although numerous differentially private machine learning methods have been established in the past decades (e.g., (Bassily et al., 2014; Wang et al., 2017; Bassily et al., 2019; Feldman et al., 2020)), it is common wisdom that these methods tend to perform worse as the model sizes increase. For instance, one commonly used approach for achieving differential privacy in empirical risk minimization (ERM) is DP-SGD (Bassily et al., 2014; Abadi et al., 2016), a private variant of stochastic gradient descent (SGD) that perturbs the gradient updates by adding isotropic Gaussian noise. However, the utility guarantee (e.g., empirical risk) of DP-SGD scales with the number of model parameters (Bassily et al., 2014) in the worst-case scenario due to the isotropic noise, leading to a vacuous learning guarantee when the number of model parameters is significantly larger than the number of training examples, which is often the case in modern machine learning tasks.

In recent years, there have been several efforts (Li et al., 2022b; Yu et al., 2022; De et al., 2022; Mehta et al., 2022; Li et al., 2022a; Zhou et al., 2021; Song et al., 2021; Kairouz et al., 2020) to address the challenges of applying differentially private machine learning methods to large models. For example, recent studies (Li et al., 2022b; Yu et al., 2022; De et al., 2022; Mehta et al., 2022)

demonstrate that leveraging DP-SGD for fine-tuning pre-trained large models can yield promising performance on downstream language and vision tasks while maintaining reasonable privacy guarantees. Nevertheless, there is still a lack of theoretical understanding regarding scenarios in which differentially private machine learning methods can effectively handle large models. To better understand this problem, we propose to study the seemingly simple yet highly challenging linear models in the over-parameterized regime.

More specifically, we consider the following private linear regression problem. Given a dataset $S = \{(\mathbf{x}_i, y_i)\}_{i=0}^{N-1}$, where each data point is drawn i.i.d. from some distribution $\mathcal{D}$, we want to find a model parameter $\widehat{\mathbf{w}}$ that can ensure differential privacy while achieving a small (population) excess risk: $L(\widehat{\mathbf{w}}) - L(\mathbf{w}^*)$, where

$$L(\mathbf{w}) := \mathbb{E}_{(\mathbf{x},y)\sim D}\big[\ell(\mathbf{w}; \mathbf{x}, y) := 1/2 \cdot (\mathbf{w}^\top \mathbf{x} - y)^2\big], \tag{1.1}$$

$\mathbf{x} \in \mathbb{R}^d$ is the feature, $y \in \mathbb{R}$ is the response, $\mathcal{D}$ is an unknown distribution over $\mathbf{x}$ and $y$, $\mathbf{w} \in \mathbb{R}^d$ is the model parameter, and $\mathbf{w}^*$ is the minimizer of $L(\mathbf{w})$. While differentially private linear regression has been extensively studied over the past decade (Vu & Slavkovic, 2009; Dwork et al., 2010; Dimitrakakis et al., 2014; Foulds et al., 2016; Minami et al., 2016; Wang, 2018; Sheffet, 2019; Cai et al., 2021; Varshney et al., 2022; Liu et al., 2023), most existing research focuses on the classical setting (i.e., $d < N$). These methods' utility (e.g., excess risk) increases with the problem dimension $d$, resulting in vacuous guarantees in the over-parameterized regime. For instance, state-of-the-art methods (Varshney et al., 2022; Liu et al., 2023) for differentially private linear regression provide an excess risk of $\widetilde{\mathcal{O}}\big(d/N + d^2/(\epsilon^2 N^2)\big)$, where $\epsilon$ is the privacy parameter, assuming the condition number of the data covariance matrix is a constant. Recently, a line of research (Kifer et al., 2012; Talwar et al., 2015; Wang & Gu, 2019; Cai et al., 2021; Asi et al., 2021) has studied private high-dimensional linear regression that achieves dimension-independent empirical risk. However, these methods often assume sparse model parameters and restricted strongly convex/smooth objective loss (Negahban et al., 2009; Loh & Wainwright, 2013). Therefore, a natural question we want to address in this paper is that

**Can one achieve a sharp, or even dimension independent, excess risk for differentially private linear regression in the over-parameterized regime under standard assumptions?**

Recently, another series of works (Dieuleveut & Bach, 2015; Berthier et al., 2020; Chen et al., 2020a; Zou et al., 2021b) have shown that stochastic gradient descent (SGD) for linear regression in the over-parameterized regime can achieve sharp excess risk that depends on the eigenspectrum of the data covariance matrix, rather than the problem dimension. Motivated by these findings, we aim to develop a private variant of SGD for linear regression that can yield similar sharp excess risk in the over-parameterized regime. However, several key challenges must be addressed to design such an algorithm and obtain strong theoretical guarantees. Firstly, ensuring differential privacy requires adding random noise to the training algorithm (Bassily et al., 2014). However, characterizing the effect of this random noise on the algorithm's convergence becomes significantly more challenging in the over-parameterized regime. Secondly, to achieve strong privacy and utility trade-offs, existing methods (Varshney et al., 2022; Liu et al., 2023) rely on large batch (or even full) gradients or privacy amplification techniques, which assume uniform sampling/random shuffling of the training data and impose stringent conditions on privacy parameters (Feldman et al., 2022; Mironov et al., 2019). Lastly, previous works typically propose adding isotropic noise to achieve differential privacy. However, in the over-parameterized regime, isotropic noise may further compromise the trade-off between privacy and utility.

**Contributions.** We develop a new private algorithm and establish corresponding analytical tools to tackle the challenges mentioned earlier and achieve a sharp excess risk for private over-parameterized linear regression. The main contributions of our work are summarized as follows.

- We propose a novel variant of the DP-FTRL algorithm (Kairouz et al., 2021) for the problem of private linear regression (see Section 3.1). The key innovation of our proposed method lies in using random noise with a general covariance matrix for differential privacy. By moving beyond the conventional use of isotropic noise, our approach enables us to achieve improved privacy and utility trade-offs in the over-parameterized regime.

- We develop new analytical tools to characterize the effect of the additive random noise (with a general covariance matrix $\boldsymbol{\Sigma}$) on the convergence of our algorithm, which can be of independent

interest. Equipped with these new tools, we prove that our method can achieve a sharp excess risk as a function of the eigenspectrum of the data covariance matrix and $\boldsymbol{\Sigma}$ (see Section 4). Our results can significantly outperform the state-of-the-art excess risk of $\widetilde{\mathcal{O}}\big(d/N + d^2/(\epsilon^2 N^2)\big)$ (Varshney et al., 2022), where $\epsilon$ is the privacy budget, for private linear regression when the dimension $d$ is much larger than the number of examples $N$. We further show that by adding random noise with a carefully designed covariance matrix $\boldsymbol{\Sigma}$ (see Section 3.2), our method can achieve improved privacy and utility trade-offs compared to using isotropic noise (i.e., $\boldsymbol{\Sigma} = \mathbf{I}$).

- We illustrate our results by considering specific eigenspectrum decays (see Section 4 and Table 1). For the polynomial decay (i.e., $\lambda_i = i^{-r}$ for $r > 1$), our method achieves $\widetilde{\mathcal{O}}(N^{-\frac{r}{1+2r}})$ and $\widetilde{\mathcal{O}}(N^{-\frac{r}{3+r} \wedge \frac{2r}{1+3r}})$ excess risk for identity and designed noise covariance matrices under the constant privacy budget ($\epsilon = O(1)$). Additionally, if we are willing to pay for a large privacy budget, i.e., $\epsilon = O\big(N^{\frac{r}{2+2r}}\big)$ and $\epsilon = O\big(N^{\frac{1}{1+r} \vee \frac{r-1}{2+2r}}\big)$ for identity and designed noise covariance matrices, our method recovers the non-private excess risk (Zou et al., 2021b) of $\widetilde{\mathcal{O}}(N^{-\frac{r}{1+r}})$. For the exponential eigenspectrum decay (i.e., $\lambda_i = e^{-i}$), our method achieves $\widetilde{\mathcal{O}}(N^{-\frac{1}{2}})$ and $\widetilde{\mathcal{O}}(N^{-\frac{2}{3}})$ excess risk for identity and designed noise covariance matrices under the constant privacy budget. If we are willing to pay for a large privacy budget, i.e., $\epsilon = O\big(N^{\frac{1}{2}}\big)$ for both noise covariance matrices, our method recovers the non-private excess risk (Zou et al., 2021b) of $\widetilde{\mathcal{O}}(N^{-1})$. Notably, our method with the designed noise covariance matrix achieves better privacy and utility trade-offs when the eigenspectrum decays at least quadratically fast, i.e., $r \geq 2$.

**Notation.** For $\mathbf{x} \in \mathbb{R}^d$, we use $\|\mathbf{x}\|_2$ to denote its $\ell_2$ norm. For a positive semidefinite matrix $\mathbf{A}$, we define $\|\mathbf{x}\|_{\mathbf{A}} = \mathbf{x}^\top \mathbf{A}\mathbf{x}$. For two quantities $a$ and $b$, we use $a \lesssim b$ if $a \leq C \cdot \text{polylog}(N) \cdot b$; we use $a \asymp b$ if $c/\text{polylog}(N) \cdot b \leq a \leq C \cdot \text{polylog}(N) \cdot b$, where $C$ and $c$ are absolute positive constants. $\mathcal{O}(\cdot)$ hides some constant parameters and $\widetilde{\mathcal{O}}(\cdot)$ further hides poly-logarithnmic terms. For any PSD matrix $\mathbf{A}$ with eigen-decomposition $\mathbf{A} = \sum_i \mu_i \mathbf{v}_i \mathbf{v}_i^\top$, where $\lambda_i$'s and $\mathbf{v}_i$'s are the eigenvalues of $\mathbf{A}$ in non-increasing order and the corresponding eigenvectors, let $\mathbf{A}_{k_1:k_2} := \sum_{k_1 < k \leq k_2} \mu_i \mathbf{v}_i \mathbf{v}_i^\top$.

## 1.1 ADDITIONAL RELATED WORK

Differentially private empirical risk (DP-ERM) minimization has been widely studied in the literature (e.g., (Chaudhuri et al., 2011; Song et al., 2013; Bassily et al., 2014; Wang et al., 2017; Bassily et al., 2019; Feldman et al., 2020; Asi et al., 2021)), where the goal is to solve the ERM problem while achieving the differential privacy for the minimizer. It has been shown that DP-(S)GD and its variants can achieve an excess risk of $\widetilde{\mathcal{O}}\big(1/N^{1/2} + d^{1/2}/(N\epsilon)\big)$ under $(\epsilon, \delta)$-DP (see Section 2) when the per example objective loss is convex and Lipschitz (Bassily et al., 2014). However, it is important to note that this result becomes vacuous for private over-parameterized linear regression, as $d$ often substantially exceeds $N$, and the objective loss is not Lipschitz unless the model parameters lie within a bounded domain.

In recent years, several attempts (Jain & Thakurta, 2014; Song et al., 2021; Kairouz et al., 2020; Zhou et al., 2021; Ma et al., 2022; Li et al., 2022a) have been made to achieve a tight excess risk in the over-parameterized regime for DP-ERM. For instance, Song et al. (2021) studied the private generalized linear models (GLMs). They proved that DP-GD could achieve a dimension independent excess risk of $\widetilde{\mathcal{O}}\big(1/N^{1/2} + \sqrt{\text{rank}}/(\epsilon N)\big)$ under $(\epsilon, \delta)$-DP, where $\text{rank}$ is the rank of the feature matrix and can be as large as $N$. However, their result requires the per-example objective loss to be convex and Lipschitz everywhere, limiting its applicability to the linear regression problem. Furthermore, DP-GD incurs significant computational complexity as it must compute the full gradient at each of the $n^2$ iterations. Subsequent works (Kairouz et al., 2020; Zhou et al., 2021) tried to extend this result to general convex/non-convex objective losses but relied on the assumption that gradients lie in a lower dimensional space. Ma et al. (2022) later showed that DP-SGD with a growing batch size could achieve an excess risk of $\widetilde{\mathcal{O}}\big(1/N^{1/2} + \text{trace}/(\epsilon N)\big)$ under $(\epsilon, \delta)$-DP, where $\text{trace}$ is the trace of the Hessian of the population loss. However, their result also assumes the per-example objective loss to be convex and Lipschitz everywhere. Li et al. (2022a) considered the case of a general convex objective with an $\ell_2$ norm regularizer. While they showed that DP-SGD with $\mathcal{O}(n^2)$ iterations can achieve a dimension independent excess risk, their result requires not only the objective loss to be Lipschitz but also imposes a stringent restricted Lipschitz continuity assumption.

## 2 PROBLEM SETTING AND PRELIMINARIES

We focus on the linear regression problem, where the population risk is defined in (1.1). We consider $y = \langle \mathbf{w}^*, \mathbf{x} \rangle + \zeta$, where $\mathbf{w}^*$ is the ground truth model parameter and $\zeta$ is the label noise that is independent of $\mathbf{x}$. Our goal is to learn a privacy-preserving (see Definition 2.4) model parameter $\widehat{\mathbf{w}}$ that minimize the population risk given a dataset $S = \left\{ (\mathbf{x}_i, y_i) \right\}_{i=0}^{N-1}$, where each data point $(\mathbf{x}_i, y_i)$ is drawn i.i.d. from $\mathcal{D}$. In this paper, we focus on the more challenging over-parameterized setting, i.e., $d > N$. We next introduce our assumptions on the data distribution and label noise.

**Assumption 2.1.** Suppose $\mathbf{x} = \mathbf{H}^{\frac{1}{2}}\mathbf{z}$, where $\mathbf{z}$ is a zero mean $\sigma_z$-sub-Gaussian random vector with an identity covariance matrix, i.e., for all vector $\mathbf{v}$, the following holds $\mathbb{E}\left[ \exp\left( \langle \mathbf{v}, \mathbf{z} \rangle^2 / (\sigma_z^2 \mathbb{E}[\langle \mathbf{v}, \mathbf{z} \rangle^2]) \right) \right] \leq 2$.

The sub-Gaussianity of the data has been considered in many previous works (Tsigler & Bartlett, 2020; Varshney et al., 2022; Liu et al., 2023) for (private) linear regression, which can imply the fourth-order momentum condition made in (Zou et al., 2021b; Velikanov et al., 2022), i.e., $\mathbb{E}_{\mathbf{x}}[\mathbf{x}\mathbf{x}^\top \mathbf{A}\mathbf{x}\mathbf{x}^\top] \leq 16\sigma_z^2 \cdot \mathrm{tr}(\mathbf{H}\mathbf{A})\mathbf{A}$ for any PSD matrix $\mathbf{A}$. In practice, most data follow the sub-Gaussian distribution, such as the data with a bounded norm. We will show in our later analysis that by imposing the following stronger assumption on the data distribution, we can design a specific noise covariance matrix to achieve improved privacy and utility trade-offs.

**Assumption 2.2.** Suppose $\mathbf{x} = \mathbf{H}^{\frac{1}{2}}\mathbf{z}$, where $\mathbf{z}$ is a zero mean random vector with independent sub-Gaussian coordinates with sub-Gaussian norm $\sigma_z$.

Assumption 2.3 further assumes that the coordinates of $\mathbf{z}$ are independent. This additional condition is beneficial in sharply characterizing the eigenspectrum of the gram matrix composed by a set of data points, which has been made in many prior works (Bartlett et al., 2020; Zou et al., 2021a; 2022). Regarding the label noise, we make the following assumption.

**Assumption 2.3.** We assume the noise $\zeta$ is zero mean with variance $\sigma_\zeta^2$. We further assume $\zeta$ is sub-Gaussian with sub-Gaussian norm $K\sigma_\zeta$ for some $K > 0$.

We next introduce the notions of differential privacy (Dwork et al., 2006) and Rényi Differential Privacy (RDP) (Mironov, 2017). In our privacy analysis, we use RDP and we state our results in terms of $(\epsilon, \delta)$-DP by converting the RDP guarantee to $(\epsilon, \delta)$-DP.

**Definition 2.4** ($(\epsilon, \delta)$-DP). A randomized mechanism $\mathcal{M}$ satisfies $(\epsilon, \delta)$-differential privacy if for adjacent datasets $S, S'$ differing by one element, and any output subset $O$, it holds that $\mathbb{P}[\mathcal{M}(S) \in O] \leq e^\epsilon \cdot \mathbb{P}[\mathcal{M}(S') \in O] + \delta$.

**Definition 2.5** (RDP). A randomized mechanism $\mathcal{M}$ satisfies $(\alpha, \rho)$-Rényi differential privacy with $\alpha > 1$ and $\rho > 0$ if for adjacent datasets $S, S' \in \mathcal{S}$ differing by one element, $D_\alpha\left( \mathcal{M}(S) \| \mathcal{M}(S') \right) := \log \mathbb{E}\left[ \left( \mathcal{M}(S) / \mathcal{M}(S') \right)^\alpha \right] \leq \rho$.

For a given function $q$, one can use the Gaussian mechanism $\mathcal{M}(S) = q(S) + \mathbf{z}$, where $\mathbf{z}$ is a random Gaussian vector, to achieve differential privacy. For example, by adding centered isotropic Gaussian noise, i.e., $\mathbf{z} \sim N(0, \sigma^2 \mathbf{I})$, one can achieve $(\sqrt{1.25\Delta(q)\log(2/\delta)}/\sigma, \delta)$-DP (Dwork et al., 2014) and $(\alpha, \alpha\Delta(q)^2/(2\sigma^2))$-RDP (Mironov, 2017), where $\sigma^2$ represents the noise magnitude. $\Delta(q)$ is the $\ell_2$-sensitivity of $q$, defined as $\Delta(q) = \sup_{S,S'} \|q(S) - q(S')\|_2$, where $S, S'$ are two adjacent datasets differing by one element. In this work, we propose adding non-isotropic noise, i.e., $\mathbf{z} \sim N(0, \sigma^2 \mathbf{\Sigma})$, to improve privacy and utility trade-offs in the over-parameterized linear regression. To this end, we derive the following result for the Gaussian mechanism with non-isotropic noise.

**Lemma 2.6.** Given a function $q$, the Gaussian mechanism $\mathcal{M} = q(S) + \mathbf{z}$, where $\mathbf{z} \sim N(0, \sigma^2 \mathbf{\Sigma})$ and $\mathbf{\Sigma}$ is positive definite, satisfies $(\alpha, \alpha\bar{\Delta}(q)^2/(2\sigma^2))$-RDP, where $\bar{\Delta}(q)^2 = \sup_{S,S'} \|q(S) - q(S')\|_{\mathbf{\Sigma}^{-1}}$ and $S, S'$ are two adjacent datasets differing by one element.

According to Lemma 2.6, when $\mathbf{\Sigma} = \mathbf{I}$, the above result recovers the Gaussian mechanism for RDP using isotropic noise (Mironov, 2017).

**Tree aggregation protocol.** We aim to achieve strong privacy and utility trade-offs without relying on large batch gradients or privacy amplification techniques. To this end, we propose to use the tree aggregation protocol (Dwork et al., 2010; Chan et al., 2011), which was originally designed for

---

**Algorithm 1:** DP-FTRL with Gaussian Noise $N(0, \mathbf{\Sigma})$ (DP-FTRL-$\mathbf{\Sigma}$)

---

1    **Input:** Training data $\{(\mathbf{x}_i, y_i)\}_{i=0}^{N-1}$, Estimating data $\{(\widetilde{\mathbf{x}}_i, \widetilde{y}_i)\}_{i=1}^m$, Covariance matrix $\mathbf{\Sigma}$,
     Noise multiplier $\tau$, Clipping list $\Psi = \{\}$, Parameters $\eta, C, \widetilde{\epsilon}, \widetilde{\delta}$

2    Initialize $\mathbf{w}_0 = \mathbf{0}$

3    **for** $t \in \{0, \dots, N-1\}$ **do**

4        Obtain $l_t = \text{RESIDUALEST}(\{(\widetilde{\mathbf{x}}_i, \widetilde{y}_i)\}_{i=1}^m, \mathbf{w}_t, \widetilde{\epsilon}, \widetilde{\delta})$ (Algorithm 2)

5        Set $\psi_t = C l_t$, and add $\psi_t$ to $\Psi$

6        Compute $\mathbf{g}_t = \text{CLIP}(\mathbf{x}_t \mathbf{x}_t^\top \mathbf{w}_t - \mathbf{x}_t y_t, \psi_t, \mathbf{\Sigma})$, where $\text{CLIP}(\boldsymbol{\nu}, \psi, \mathbf{\Sigma}) = \boldsymbol{\nu} \cdot \min\left\{1, \frac{\psi}{\|\boldsymbol{\nu}\|_{\mathbf{\Sigma}^{-1}}}\right\}$

7        Set the noise magnitude $\sigma^2 = \tau^2 \psi^2$, where $\psi = \max \Psi$

8        Send $(\mathbf{g}_t, \sigma^2 \mathbf{\Sigma})$ to the private tree aggregation protocol (Algorithm 3), and receive the
         private previous sum $\widetilde{\mathbf{g}}_{\leq t}$

9        Update $\mathbf{w}_{t+1} = \text{argmin}_{\mathbf{w} \in \mathbb{R}^d} \langle \widetilde{\mathbf{g}}_{\leq t}, \mathbf{w} \rangle + \frac{1}{2\eta} \|\mathbf{w}\|_2^2$

10    **Ouput:** $\overline{\mathbf{w}}_N = \frac{1}{N} \sum_{t=0}^{N-1} \mathbf{w}_t$

---

the partial sum problem. Consider the problem of privately releasing the partial sum $\sum_{i=1}^t \mathbf{v}_i$ for $t \in [N]$ given a stream of vectors $\mathbf{v}_1, \dots, \mathbf{v}_N$. In this protocol, a complete binary tree is constructed with leaf nodes as $\mathbf{v}_1, \dots, \mathbf{v}_N$. Each internal node in the tree stores the sum of all leaf nodes in its subtree. Therefore, each partial sum $\sum_{i=1}^t \mathbf{v}_i$ can be computed using at most $\lceil \log_2 t \rceil$ nodes in the tree. Since each $\mathbf{v}_i$ only affects at most $\bar{k} = \lceil \log_2 N \rceil + 1$ nodes, i.e., the nodes along the path from $\mathbf{v}_i$ to the root of the tree, the complete tree will be $(\epsilon, \delta)$-DP if we add random Gaussian noise $\mathbf{z} \sim N(0, \sigma^2 \mathbf{I})$ to each node, where $\sigma^2 = O(\psi^2 \bar{k} \log(1/\delta)/\epsilon^2)$ and $\psi$ is the $\ell_2$-norm upper bound of all $\mathbf{v}_i$'s. As a result, we will add at most $\lceil \log_2 t \rceil$ Gaussian noises to the partial sum $\sum_{i=1}^t \mathbf{v}_i$ to obtain its differentially private estimate. See more detailed discussions in Appendix A.

## 3   PRIVATE OVER-PARAMETERIZED LINEAR REGRESSION

In this section, we present our proposed algorithm, which only takes one pass over the training dataset and is able to achieve strong privacy and utility guarantees.

### 3.1   DIFFERENTIALLY PRIVATE ALGORITHM

Our proposed algorithm, i.e., DP-FTRL-$\mathbf{\Sigma}$, is illustrated in Algorithm 1, which is a variant of the DP-FTRL algorithm (Kairouz et al., 2021) with the key innovation of adding random noise with a general covariance matrix for differential privacy.

**Update rule.** The main idea of DP-FTRL-$\mathbf{\Sigma}$ is to use the follow-the-perturbed-leader (FTRL) with linearized losses (Hazan et al., 2016), where we seek to find the minimizer of the following regularized cumulative past losses at $t$-th iteration: $\mathbf{w}_{t+1} = \text{argmin}_{\mathbf{w} \in \mathbb{R}^d} \langle \sum_{j=0}^t \mathbf{g}_i, \mathbf{w} \rangle + \|\mathbf{w}\|_2^2/(2\eta)$ with $\sum_{j=0}^t \mathbf{g}_i$ as the sum of previous stochastic gradients (see line 9). Note that $\mathbf{w}_{t+1}$ shares the same update form with SGD when $\mathbf{w}_0 = \mathbf{0}$.

**Privacy mechanism.** To ensure the differential privacy of the learned model parameter, we propose to use the tree aggregation protocol (Dwork et al., 2010; Chan et al., 2011; Guha Thakurta & Smith, 2013; Kairouz et al., 2021) to obtain a private estimate of $\sum_{j=0}^t \mathbf{g}_i$ (see line 8). More specifically, at the beginning of training, we create a binary tree $\mathcal{T}$ of size $(2^{\lceil \log_2 N \rceil + 1} - 1)$ with $N$ leaves. At $t$-th iteration, the tree aggregation protocol receives the vector $\mathbf{g}_t$, computed using the data $(\mathbf{x}_t, y_t)$, then updates the binary tree $\mathcal{T}$ accordingly. Finally, it outputs $\widetilde{\mathbf{g}}_{\leq t}$, which is computed using the values stored in the current tree and perturbed by some random noise, as the private estimate of $\sum_{j=0}^t \mathbf{g}_i$. A more detailed description can be found in Appendix A. Compared to previous approaches (Varshney et al., 2022; Liu et al., 2023), the tree aggregation protocol has the advantage of providing strong privacy guarantees without requiring uniform sampling/random shuffling of the training data or using large batches in each update. For example, it is possible to apply our analytical tools to DP-Shuffled SGD (Varshney et al., 2022) to obtain dimensional independent utility guarantees. However, the random shuffling technique behind DP-Shuffled SGD will lead to a stringent conditions on privacy

parameter, i.e., $\epsilon \leq \sqrt{N}$ (Feldman et al., 2022; Varshney et al., 2022)), which is very restrictive in practice (one often considers a constant level privacy budget). In addition, this requirement will give us a much worse condition on $N$. At the same time, our method with the tree aggregation technique does not have such a requirement on $\epsilon$. Note that our method requires more memory usage to keep track of past stochastic gradients and noise vectors. Furthermore, we propose adding random noise with a general covariance matrix $\mathbf{\Sigma}$ and magnitude $\sigma^2$ (see line 8) inside the tree aggregation protocol. This enables us to achieve improved privacy and utility trade-offs.

**Adaptive clipping.** Gradient clipping (Abadi et al., 2016) (see line 6) is used to control the magnitude of the stochastic gradient and thus determine the appropriate amount of random noise for DP (see Lemma 2.6). The clipping parameter $\psi$ could significantly affect the algorithm's convergence (Chen et al., 2020b). For instance, if $\psi$ is set to be too small, the magnitude of the stochastic gradient is significantly reduced, resulting in slower convergence. Conversely, if $\psi$ is set to be too large, excessive noise is added (the noise magnitude scales with $\psi$) to achieve DP, potentially causing divergence even though the stochastic gradient remains unchanged. Ideally, we want to select $\psi$ adaptively as the magnitude of the current stochastic gradient, i.e., $\|\mathbf{x}_t\mathbf{x}_t^\top \mathbf{w}_t - \mathbf{x}_t y_t\|_{\mathbf{\Sigma}^{-1}}$, such that no clipping is required. To this end, we propose to estimate the magnitude of the stochastic gradient at $t$-th iteration using some estimating data $\{(\widetilde{\mathbf{x}}_i, \widetilde{y}_i)\}_{i=1}^m$ and set the clipping parameter $\psi_t$ accordingly. Note that we have $\|\mathbf{x}_t\mathbf{x}_t^\top \mathbf{w}_t - \mathbf{x}_t y_t\|_{\mathbf{\Sigma}^{-1}} \leq \|\mathbf{x}_t\|_{\mathbf{\Sigma}^{-1}} \cdot \ell^{1/2}(\mathbf{w}_t; \mathbf{x}_t, y_t)$. Thus, we set $\psi_t = C l_t$ (see line 5), where $C$ is the upper bound of $\|\mathbf{x}_i\|_{\mathbf{\Sigma}^{-1}}$ for all $\mathbf{x}_i$ in the training data and $l_t$ is an estimate of $\ell^{1/2}(\mathbf{w}_t; \mathbf{x}_t, y_t)$, which can be obtained by the residual estimator established in Liu et al. (2023). The key idea of the residual estimator, i.e., RESIDUALEST in line 4, is to use the (private) empirical variance estimator of the residual $\sum_{i=1}^m (\widetilde{y}_i - \widetilde{\mathbf{x}}_i^\top \mathbf{w}_t)^2/m$. A detailed description can be found in Appendix A. It is worth noting that when no additional estimating data is available, we can evenly split the training data and use one part as the estimating data. By doing so, all of our results will still hold, with the only modification being to change $N$ to $N/2$.

Note that when we choose $\mathbf{\Sigma} = \mathbf{I}$ and use a fixed clipping parameter $\psi$, Algorithm 1 reduces to the DP-FTRL algorithm (Kairouz et al., 2021). In the next section, we will provide a method to design the non-isotropic noise, which allows us to achieve better privacy and utility trade-offs.

## 3.2 DESIGN OF THE NOISE COVARIANCE

As mentioned previously, the additive noise, i.e., $\mathbf{z} \sim N(0, \sigma^2\mathbf{\Sigma})$ (see line 8 in Algorithm 1), in our method performs a trade-off between privacy and utility. Intuitively, stronger noise will lead to a better privacy guarantee but can hurt the convergence of the optimization algorithm. The standard choice of the noise covariance, i.e., $\mathbf{\Sigma} = \mathbf{I}$, can be understood as treating all coordinates equally. However, we notice that (1) this type of noise will incur large variance error in terms of the excess risk in the high-dimensional regime; (2) using the same Gaussian noise for all coordinates may be too conservative, as some coordinates may not have sufficient signals that need to be protected; and (3) the noise magnitude $\sigma^2$ will also be determined by the choice of $\mathbf{\Sigma}$ (see Lemma 2.6).

Motivated by this, we seek to explore a better design of $\mathbf{\Sigma}$ that can (1) successfully protect the gradient information and (2) achieve a faster convergence rate than when $\mathbf{\Sigma} = \mathbf{I}$. Note that for linear problems, the stochastic gradient for the data pair $(\mathbf{x}_i, y_i)$ is parallel to the feature $\mathbf{x}_i$, implying that the design of $\mathbf{\Sigma}$ should be related to the signal strengths of all coordinates of $\mathbf{x}_i$. More specifically, an improved design of $\mathbf{\Sigma}$ should concern the data covariance matrix $\mathbf{H}$: one may use stronger noise perturbation along the large-eigenvalue directions of $\mathbf{H}$. In contrast, the noise level can be relatively weaker along the small-eigenvalue directions of $\mathbf{H}$. Therefore, we consider the following design:

$$\mathbf{\Sigma} = \lambda\mathbf{I} + \mathbf{H}, \tag{3.1}$$

where $\lambda > 0$ is a user-defined constant. It can be seen that when $\lambda \to \infty$, we have $\mathbf{\Sigma} \to \lambda\mathbf{I}$, which is nearly the same as the identity design (ignoring the scaling parameter). Moreover, the noise magnitude $\sigma^2$ is also determined by $\mathbf{\Sigma}$, as discussed earlier. In our problem, $\sigma^2$ scales with $\mathrm{tr}(\mathbf{\Sigma}^{-1}\mathbf{H})$ (see Theorem 4.2) in order to achieve the desired privacy guarantee. If we choose $\mathbf{\Sigma} \to \mathbf{H}$ when $\lambda \to 0$, the noise magnitude $\sigma^2 \to d$, which explodes as $d \to \infty$. This implies that $\lambda$ should be a positive constant. Therefore, $\lambda$ is a knob to control the noise along different directions.

However, it is also worth noting that $\mathbf{H}$, i.e., the population covariance of the data, cannot be realized in practice. In this work, we consider the scenarios in which we are allowed to access a set of unlabeled public data $\{\widetilde{\mathbf{x}}_i\}_{i=1}^M$, which is sampled from the same distribution as the training data

points. In general, we consider the case that $M \geq N$. Therefore, we propose to use the unlabeled data first to generate an estimation of $\mathbf{H}$ and then design the noise covariance matrix as follows

$$\mathbf{\Sigma} = \left(\lambda \mathbf{I} + \widetilde{\mathbf{X}}\widetilde{\mathbf{X}}^{\top}\right)/M, \tag{3.2}$$

where $\widetilde{\mathbf{X}} = [\widetilde{\mathbf{x}}_1, \ldots, \widetilde{\mathbf{x}}_M] \in \mathbb{R}^{d \times M}$ and $\lambda > 0$ is a user-defined parameter. The idea of using public data in the DP methods has been widely used in the literature (Kairouz et al., 2020; Yu et al., 2022; Zhou et al., 2021). In addition, getting unlabeled public data is generally cheaper than labeled training data. Note that if we do not have a public dataset, we can use the isotropic noise in our method to still get a sharp excess risk bound (see Corollary 4.3).

## 4 MAIN RESULTS

In this section, we provide the privacy and utility guarantees of our proposed method.

**Theorem 4.1** (Privacy guarantee). If we set $\tau^2 = 4\bar{k}\left(\log(1/\delta) + \epsilon\right)/\epsilon^2$, $\widetilde{\epsilon} = \epsilon/\sqrt{8N\log(2/\delta)}$, $\widetilde{\delta} = \delta/(2N)$, where $\bar{k} = \lceil\log_2 N\rceil + 1$, then Algorithm 1 is $(\epsilon, \delta)$-DP.

The choice of $\widetilde{\epsilon}, \widetilde{\delta}$ in Theorem 4.1 is to ensure that RESIDUALEST (see Algorithm 2 in Appendix A) is $(\epsilon, \delta)$-DP when the estimating data $\{(\widetilde{\mathbf{x}}_i, \widetilde{y}_i)\}_{i=1}^m$ are not public available. Furthermore, our privacy guarantee does not rely on large batch gradients (Liu et al., 2023) or privacy amplification techniques (Varshney et al., 2022), which often impose stringent conditions on privacy parameters. For example, the DP-SHUFFLED SGD (Varshney et al., 2022) requires the privacy budget $\epsilon \leq 1/\sqrt{N}$ due to the random shuffling of the training data.

Next, we provide the utility guarantee, i.e., the excess risk, of our proposed method.

**Theorem 4.2** (Utility guarantee for general $\mathbf{\Sigma}$). Under the same conditions of $\tau^2$, $\widetilde{\epsilon}$, $\widetilde{\delta}$ as in Theorem 4.1, and suppose Assumptions 2.1, 2.3 hold. If we choose the clipping parameter $C = 6\sigma_z\sqrt{(1 + \sigma_z^2)\operatorname{tr}(\mathbf{\Sigma}^{-1}\mathbf{H})}\log N$, step size $\eta \leq 1/\left(16(1 + \sigma_z^4)\operatorname{tr}(\mathbf{H})\log_2 N\right)$, and the number of estimating data $m = \Omega(\log^2(N/\widetilde{\delta})/\widetilde{\epsilon})$, then with probability at least $1 - 1/N$, the output of Algorithm 1 satisfies

$$\mathbb{E}\left[L(\overline{\mathbf{w}}_N)\right] - L(\mathbf{w}^*) \leq \operatorname{err}_{\text{bias}} + \operatorname{err}_{\text{variance}},$$

where

$$\operatorname{err}_{\text{bias}} \lesssim \frac{\|\mathbf{w}_0 - \mathbf{w}^*\|_{\mathbf{H}_{0:k}^{-1}}^2}{\eta^2 N^2} + \|\mathbf{w}_0 - \mathbf{w}^*\|_{\mathbf{H}_{k:\infty}}^2 + \frac{\sigma_z^4\left(\|\mathbf{w}_0 - \mathbf{w}^*\|_{\mathbf{I}_{0:k}}^2 + N\eta\|\mathbf{w}_0 - \mathbf{w}^*\|_{\mathbf{H}_{k:\infty}}^2\right)}{N\eta(1 - \sigma_z^4\eta\operatorname{tr}(\mathbf{H}))} \cdot \Lambda;$$

$$\operatorname{err}_{\text{variance}} \lesssim \sigma_\zeta^2\sigma_z^4\left(1 + \eta\left(\sum_{i \leq k}\lambda_i + N\eta\sum_{i \geq k}\lambda_i^2\right)\right) \cdot \Lambda + \tau^2\sigma_z^4\eta\langle\mathbf{I}_{0:k} + N\eta\mathbf{H}_{k:\infty}, \bar{\mathbf{G}}_p\rangle \cdot \Lambda$$

$$+ \tau^2\langle\mathbf{H}_{0:k}^{-1} + N^2\eta^2\mathbf{H}_{k:\infty}, \bar{\mathbf{G}}_p\rangle \cdot \frac{1}{N}$$

with arbitrary $k \geq 1$, $\Lambda = k/N + N\eta^2\sum_{i>k}\lambda_i^2$, $\bar{\mathbf{G}}_p = \eta\sigma_z^4\operatorname{tr}(\mathbf{H}\mathbf{\Sigma}) \cdot \left(\mathbf{I}_{0:k} + N\eta\mathbf{H}_{k:\infty}\right) + \left(\mathbb{E}[\psi_{\max}^2 + \psi_{\max}^4] + 1\right) \cdot \mathbf{\Sigma}$, $\psi_{\max} = Cl$ and $l = \max\{l_0^2, l_1^2, \ldots, l_{N-1}^2\}$.

According to Theorem 4.2, the last two terms in $\operatorname{err}_{\text{variance}}$ correspond to the error introduced by the privacy mechanism. Note that in the non-private case, i.e., $\epsilon = \infty$, the term $\tau^2 = 4\bar{k}\left(\log(1/\delta) + \epsilon\right)/\epsilon^2$ becomes zero, and $\operatorname{err}_{\text{variance}}$ reduces to the non-private one (Zou et al., 2021b). Additionally, the error bound in Theorem 4.2 is expressed as a function with respect to the full eigenspectrum of $\mathbf{H}$ (i.e., $\lambda_i$'s), which does not explicitly depend on the problem dimension $d$. This result is therefore stronger than the existing dimension dependent utility bounds (Varshney et al., 2022; Liu et al., 2023). Note that $\psi_{\max} = Cl$ in $\bar{\mathbf{G}}_p$ determines the magnitude of the additive random noise to achieve DP. The term $C$ is dominated by $\operatorname{tr}(\mathbf{H}\mathbf{\Sigma}^{-1})$ and $l \lesssim \max\left\{\|\mathbf{w}_t - \mathbf{w}^*\|_{\mathbf{H}}^2 + \sigma_\zeta^2\right\}_{t=0}^{N-1}$.

Moreover, the error bound in Theorem 4.2, particularly the private error terms (e.g., the last two terms), strongly depends on the design of $\mathbf{\Sigma}$ (see the definition of $\bar{\mathbf{G}}_p$). Therefore, seeking a good design of $\mathbf{\Sigma}$ is important to achieve a better utility guarantee. In particular, we prefer $\mathbf{\Sigma}$ with a $\mathbf{H}$-like head space (in contrast to $\mathbf{I}$), so that the quantities $\langle\mathbf{I}_{0:k}, \mathbf{\Sigma}\rangle$ and $\langle\mathbf{H}_{0:k}^{-1}, \mathbf{\Sigma}\rangle$ can be well controlled. Additionally, $\mathbf{\Sigma}$ should also attain a relatively heavy tail so that the $\operatorname{tr}(\mathbf{H}\mathbf{\Sigma}^{-1})$ in $\psi_{\max}$ will not explode. This is consistent with our high-level idea of the design of $\mathbf{\Sigma}$ in (3.1).

Table 1: Excess risks for our method under different eigenspectrum decays. The results are presented under $(\epsilon, \delta)$-DP with constant $\epsilon$ and $\log(1/\delta)$, and parameters other than $N, d$ are considered to be constants. * The results are restated by using different stepsizes to achieve the optimal excess risks.

| Method | Excess Risk for Polynomial Decay: $\lambda_i = i^{-r}$ for $r > 1$ | Excess Risk for Exponential Decay: $\lambda_i = e^{-i}$ |
|---|---|---|
| NON-PRIVATE SGD* (Zou et al., 2021b) | $\widetilde{\mathcal{O}}\left(N^{-\frac{r}{1+r}}\right)$ | $\widetilde{\mathcal{O}}\left(N^{-1}\right)$ |
| DP-FTRL-$\mathbf{I}$ Corollary 4.4 | $\widetilde{\mathcal{O}}\left(N^{-\frac{r}{1+2r}}\right)$ | $\widetilde{\mathcal{O}}\left(N^{-1/2}\right)$ |
| DP-FTRL-$\mathbf{\Sigma}$ Corollary 4.6 | $\widetilde{\mathcal{O}}\left(N^{-\frac{r}{3+r} \wedge \frac{2r}{1+3r}}\right)$ | $\widetilde{\mathcal{O}}\left(N^{-2/3}\right)$ |

## 4.1 UTILITY AND PRIVACY TRADE-OFFS FOR IDENTITY NOISE COVARIANCE MATRIX

In this section, we present the utility guarantee of our method using the identity noise covariance matrix. To simplify the results, we assume that $\sigma_z$, $\lambda_1(\mathbf{H})$, and $\mathrm{tr}(\mathbf{H})$ are in the constant level.

**Corollary 4.3** (General utility guarantee for $\mathbf{\Sigma} = \mathbf{I}$). Under the same conditions of data, label noise, and parameters $\tau^2$, $\widetilde{\epsilon}$, $\widetilde{\delta}$, $\widetilde{m}$, $\eta$, $C$ as in Theorem 4.2. If we assume $\|\mathbf{w}_0 - \mathbf{w}^*\|_2 \leq B$, then with probability at least $1 - 1/N$, the output of Algorithm 1 satisfies

$$\mathbb{E}\left[L(\overline{\mathbf{w}}_N)\right] - L(\mathbf{w}^*) \lesssim \underbrace{\frac{B^2}{N\eta} + \left(\frac{\sigma_\zeta^2}{N} + \frac{B^2}{N^2\eta}\right) \cdot \left(k^* + N\eta \sum_{i > k^*} \lambda_i\right)}_{\text{Non-private error component}} + \underbrace{l^2\tau^2\eta \cdot \left(k^* + N\eta \sum_{i > k^*} \lambda_i\right)}_{\text{Private error component}},$$

where $k^* = \max\{k : \lambda_k \geq 1/(N\eta)\}$, and $l = \max\{l_0^2, l_1^2, \ldots, l_{N-1}^2\}$.

In Corollary 4.3, we decompose the excess risk into two components: the non-private error component and the private error component. The former can be further decomposed into the bias and variance errors (i.e., the first and second terms in the developed bound). Moreover, it can also be observed that the quantity $k^* + N\eta \sum_{i > k^*}$ is rather critical in the excess risk bound, which can be understood as the effective dimension of the problem, denoted by $\mathrm{EffectDim}$. Then, in order to achieve vanishing excess risk, we need to guarantee that $N\eta \gg 1$ and $\eta \cdot \mathrm{EffectDim} \ll 1$, which suggests that the effective dimension should satisfy $\mathrm{EffectDim} \ll N$.

To better illustrate the utility guarantee and identify the problem instances that the vanishing excess risk can be achieved, we consider some examples of eigenspectrums and present their corresponding guarantees as follows. The results presented consider parameters other than $N$, $d$, and $\epsilon$ as constants.

**Corollary 4.4** (Utility with $\mathbf{I}$ on specific distributions). Under the same conditions as Corollary 4.3.

1. If the eigenspectrum of $\mathbf{H}$ satisfies $\lambda_i = i^{-r}$ for some $r > 1$, then with probability at least $1 - 1/N$, we have $\mathbb{E}\left[L(\overline{\mathbf{w}}_N)\right] - L(\mathbf{w}^*) = \widetilde{\mathcal{O}}\left(N^{-\frac{r}{1+r}}\left(1 + \left(\epsilon^{-2}N^{\frac{r}{1+r}}\right)^{\frac{r}{1+2r}}\right)\right)$.

2. If the eigenspectrum of $\mathbf{H}$ satisfies $\lambda_i = e^{-i}$, then with probability at least $1 - 1/N$, the output of Algorithm 1 satisfies $\mathbb{E}\left[L(\overline{\mathbf{w}}_N)\right] - L(\mathbf{w}^*) = \widetilde{\mathcal{O}}\left(N^{-1}\left(1 + \left(\epsilon^{-2}N\right)^{\frac{1}{2}}\right)\right)$.

According to Corollary 4.4, our method with $\mathbf{\Sigma} = \mathbf{I}$ achieves a vanishing excess risk that is independent of the problem dimension when the eigenspectrum decays polynomially or exponentially. It's worth noting that the state-of-the-art methods (Varshney et al., 2022; Liu et al., 2023) for the private linear regression problem provide an excess risk of $\widetilde{\mathcal{O}}\left(d/N + d^2/(\epsilon^2 N^2)\right)$. However, this approach becomes invalid in the over-parameterized setting, as the dimension $d$ can be much larger than $N$. If we choose a constant privacy budget $\epsilon = O(1)$ for our method, we can achieve the excess risk of $\widetilde{\mathcal{O}}\left(N^{-r/(1+2r)}\right)$ and $\widetilde{\mathcal{O}}\left(N^{-1/2}\right)$ for polynomial and exponential decay, respectively. Furthermore, if we are willing to pay for the privacy budget $\epsilon = O\left(N^{r/(2+2r)}\right)$ and $\epsilon = O\left(N^{1/2}\right)$, our method can recover the non-private (Zou et al., 2021b) excess risk of $\widetilde{\mathcal{O}}\left(N^{-r/(1+r)}\right)$ and $\widetilde{\mathcal{O}}\left(N^{-1}\right)$ for polynomial and exponential decay, respectively.

### 4.2 UTILITY AND PRIVACY TRADE-OFFS FOR SPECIALLY DESIGNED COVARIANCE MATRIX

We provide the utility guarantee of our method with the noise covariance matrix in (3.2). We assume that $\sigma_z$, $\lambda_1(\mathbf{H})$, and $\text{tr}(\mathbf{H})$ are in the constant level in the following results.

**Corollary 4.5** (General utility guarantee for $\boldsymbol{\Sigma} = M^{-1}\big(\lambda\mathbf{I} + \widetilde{\mathbf{X}}\widetilde{\mathbf{X}}^\top\big)$)**.** Under the same conditions of parameters $\tau^2$, $\widetilde{\epsilon}$, $\widetilde{\delta}$, $\widetilde{m}$, $\eta$, $C$ as in Theorem 4.2, suppose Assumptions 2.2, 2.3 hold, $\|\mathbf{w}_0 - \mathbf{w}^*\|_2 \leq B$, and $M \geq N$. If we choose $\boldsymbol{\Sigma} = M^{-1}\big(\lambda\mathbf{I} + \widetilde{\mathbf{X}}\widetilde{\mathbf{X}}^\top\big)$ with $\lambda = M/(N\eta)$, then with probability at least $1 - 1/N$, the output of Algorithm 1 satisfies

$$\mathbb{E}\big[L(\overline{\mathbf{w}}_N)\big] - L(\mathbf{w}^*) \lesssim \underbrace{\frac{B^2}{N\eta} + \left(\frac{\sigma_\zeta^2}{N} + \frac{B^2}{N^2\eta}\right) \cdot \left(k^* + N\eta\sum_{i>k^*}\lambda_i\right)}_{\text{Non-private error component}}$$

$$\underbrace{+ \, l^2\tau^2 \cdot \left[\eta^2 \cdot \text{tr}(\mathbf{H}^2) \cdot \left(k^* + N\eta\sum_{i>k^*}\lambda_i\right) + \frac{1}{N} \cdot \left(k^* + N\eta\sum_{i>k^*}\lambda_i\right)^3\right]}_{\text{Private error component}},$$

where $k^* = \max\{k : \lambda_k \geq 1/(N\eta)\}$, and $l = \max\{l_0^2, l_1^2, \ldots, l_{N-1}^2\}$.

By combining Corollaries 4.3 and 4.5, it is clear that (1) the non-private error component will maintain the same as it does not depend on $\boldsymbol{\Sigma}$; (2) the private error components for $\boldsymbol{\Sigma} = \mathbf{I}$ and $\boldsymbol{\Sigma} = M^{-1}(\lambda\mathbf{I} + \widetilde{\mathbf{X}}\widetilde{\mathbf{X}}^\top)$ are roughly $\eta \cdot \text{EffectDim}$ and $\eta^2 \cdot \text{EffectDim} + (\text{EffectDim})^3/N$, respectively. Then, noting that we need to guarantee $N\eta \gg 1$ to achieve vanishing bias error, it can be seen that the private error component achieved by specially designed $\boldsymbol{\Sigma}$ outperforms that achieved by $\boldsymbol{\Sigma} = \mathbf{I}$ for the data distribution with small $\text{EffectDim}$ (i.e., the eigenspectrum has a fast decay).

To better present the advantage of the specially designed noise covariance over the identity one, we again consider some example data distributions and calculate their corresponding utility guarantees in the following corollary. We consider parameters other than $N$, $d$ and $\epsilon$ to be constants as before.

**Corollary 4.6** (Utility with $\boldsymbol{\Sigma} = M^{-1}\big(\lambda\mathbf{I} + \widetilde{\mathbf{X}}\widetilde{\mathbf{X}}^\top\big)$ on specific distributions)**.** Under the same conditions as Corollary 4.3.

1. If the eigenspectrum of $\mathbf{H}$ satisfies $\lambda_i = i^{-r}$ for some $r > 1$, then with probability at least $1 - 1/N$, we have $\mathbb{E}\big[L(\overline{\mathbf{w}}_N)\big] - L(\mathbf{w}^*) = \widetilde{\mathcal{O}}\Big(N^{-\frac{r}{1+r}}\Big[1 + \big(\epsilon^{-2}N^{\frac{2}{1+r}}\big)^{\frac{r}{3+r}} + \big(\epsilon^{-2}N^{\frac{r-1}{1+r}}\big)^{\frac{r}{1+3r}}\Big]\Big)$.

2. If the eigenspectrum of $\mathbf{H}$ satisfies $\lambda_i = e^{-i}$, then with probability at least $1 - 1/N$, the output of Algorithm 1 satisfies $\mathbb{E}\big[L(\overline{\mathbf{w}}_N)\big] - L(\mathbf{w}^*) = \widetilde{\mathcal{O}}\Big(N^{-1}\big(1 + \big(\epsilon^{-2}N\big)^{\frac{1}{3}}\big)\Big)$.

According to Corollary 4.6, our method with the covariance matrix in (3.2) can also achieve a vanishing empirical risk under the same eigenspectrum decay conditions as in Corollary 4.4. If we choose $\epsilon = O(1)$, we can achieve $\widetilde{\mathcal{O}}\big(N^{-\frac{r}{3+r} \wedge \frac{2r}{1+3r}}\big)$ and $\widetilde{\mathcal{O}}\big(N^{-2/3}\big)$ excess risk for polynomial and exponential decay. Therefore, our method with this designed $\boldsymbol{\Sigma}$ achieves a better excess risk than the one using $\mathbf{I}$ with the excess risk of $\widetilde{\mathcal{O}}\big(N^{-\frac{r}{1+2r}}\big)$ and $\widetilde{\mathcal{O}}\big(N^{-1/2}\big)$ when the eigenspectrum decays at least quadratically fast, i.e., $r \geq 2$. Moreover, if we are willing to pay for the privacy budget $\epsilon = O\big(N^{\frac{1}{1+r}} \vee N^{\frac{r-1}{2+2r}}\big)$ and $\epsilon = O(N^{1/2})$ for polynomial and exponential decay, our method can also recover the non-private excess risk as before. Table 1 summarizes these results for comparison.

## 5 CONCLUSION AND FUTURE WORK

We develop a new variant of DP-FTRL for private linear regression in the over-parameterized regime. The key innovation of our method is the utilization of non-isotropic noise, enabling us to obtain improved privacy and utility trade-offs. We prove that our method achieves a sharp excess risk, depending on the eigenspectrum of the data covariance matrix instead of the problem dimension. Specific examples of data distribution further validate the effectiveness of our method. As for future work, exploring different designs of $\boldsymbol{\Sigma}$ and studying how to overcome the limitation of using public data are both very interesting directions. Moreover, it is also interesting to extend our results to the kernel setting.

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

## A  ADDITIONAL ALGORITHMS

---

**Algorithm 2:** RESIDUALEST

---

1 **Input:** Data $\{(\widetilde{\mathbf{x}}_i, \widetilde{y}_i)\}_{i=1}^m$, model $\mathbf{w}$, parameters $\widetilde{\epsilon}, \widetilde{\delta}$
2 **Option 1:** $\{(\widetilde{\mathbf{x}}_i, \widetilde{y}_i)\}_{i=1}^m$ is a public dataset
3 $l = \sqrt{2 \sum_{i=1} r_i / m}$, where $r_i = (\widetilde{y}_i - \langle \mathbf{w}, \widetilde{\mathbf{x}}_i \rangle)^2$
4 **Option 2:** $\{(\widetilde{\mathbf{x}}_i, \widetilde{y}_i)\}_{i=1}^m$ is a private data
5 Obtain $\{r_i\}_{i=1}^m$ and split $\{r_i\}_{i=1}^m$ into $\widetilde{n}$ subsets of equal size, each with $|\mathcal{S}_j| = \widetilde{m}$ data for
   $j \in [\widetilde{n}]$, where $r_i = (\widetilde{y}_i - \langle \mathbf{w}, \widetilde{\mathbf{x}}_i \rangle)^2$, $\widetilde{n} = \lceil \beta_1 \log(N/\widetilde{\delta})/\widetilde{\epsilon} \rceil$, where $\beta_1$ is some large constant
6 Obtain $\{\bar{r}_j\}_{j=1}^{\widetilde{n}}$, where $\bar{r}_j = \sum_{i \in \mathcal{S}_j} r_i / \widetilde{m}$
7 Partition $[0, \infty)$ into geometrically increasing intervals, i.e.,
   $\Omega = \{\cdots, [2^{-1}, 1), [1, 2), [2, 2^2), \cdots\} \cup \{0\}$
8 Compute $\widehat{r}_k = \sum_{j=1}^{\widetilde{n}} I(\bar{r}_j \in B_k)/\widetilde{n}$, where $B_k$ is the interval in $\Omega$
9 **if** $\widehat{r}_k \in (0, 2\log(1/\widetilde{\delta})/(\widetilde{\epsilon}\widetilde{n}) + (1/\widetilde{n}))$ **then**
10 $\quad \widetilde{r}_k = \widehat{r}_k + z_k$, where $z_k \sim \text{Lap}\big(0, 2/(\widetilde{\epsilon}\widetilde{n})\big)$
11 **else**
12 $\quad \widetilde{r}_k = 0$
13 Let $[l_1, l_2]$ be the nonempty interval containing the largest $\widetilde{r}_k$ and set $l = \sqrt{2l_2}$
14 **Ouput:** $l$

---

In Algorithm 1, we propose to use the residual estimator established in Liu et al. (Liu et al., 2023) to estimate $\ell^{1/2}(\mathbf{w}_t; \mathbf{x}_t, y_t)$. The detailed description of this method is in Algorithm 2. The theoretical guarantees of Algorithm 2 can be found in Lemma B.3.

---

**Algorithm 3:** Private Tree Aggregation Protocol

---

1 **Input:** Vectors $\mathbf{g}_0, \ldots, \mathbf{g}_{N-1}$ (in an online sequence), Noise covariance $\sigma^2 \mathbf{\Sigma}$
2 **Initialization:** Create a binary tree $\mathcal{T}$ of size $2^{\lceil \log_2 N \rceil + 1} - 1$ with leaves $m_1, m_2, \ldots, m_N$
3 **Online phase:** At each iteration $t$, execute lines 3 to 17
4 Accept $\mathbf{g}_t$ from the data stream
5 Let $\mathbf{p} = \{n_1, n_2, \ldots, n_k\}$ be a set of nodes from the root of $\mathcal{T}$ to the $t$-th leaf, where $k$ is the
   depth of $\mathcal{T}$, $n_1$ is the root, and $n_k$ is the leaf $m_t$
6 **Tree update:** lines 4 to 8
7 Let $\mathbf{p}_j = \{n_j, n_{j+1}, \ldots, n_k\}$, where $n_j$ is the last node in $\mathbf{p}$ that is a left child in $\mathcal{T}$
8 **for** $i \in [k]$ **do**
9 $\quad \boldsymbol{\nu}_i = \boldsymbol{\nu}_i + \mathbf{g}_t$, where $\boldsymbol{\nu}_i$ is the associated value of node $n_i$
10 $\quad$ **if** $n_i \in \mathbf{p}_j$ **then**
11 $\quad\quad \boldsymbol{\nu}_i = \boldsymbol{\nu}_i + \mathbf{z}$, where $\mathbf{z} \sim N(0, \sigma^2 \mathbf{\Sigma})$
12 **Obtain private sum:** lines 10 to 13
13 Initialize $\widetilde{\mathbf{g}}_{\leq t} = \mathbf{0}$ and let $\{b_1, \ldots, b_k\}$ be the $k$ bit binary representation of $t$
14 **for** $i \in [k]$ **do**
15 $\quad$ **if** $b_i = 1$ **then**
16 $\quad\quad \widetilde{\mathbf{g}}_{\leq t} = \widetilde{\mathbf{g}}_{\leq t} + \boldsymbol{b}_i$, where $\boldsymbol{b}_i = \boldsymbol{\nu}_i$ if $n_i$ is the left child, otherwise $\boldsymbol{b}_i$ is the value
   associated with the left sibling of $n_i$
17 **Output:** $\widetilde{\mathbf{g}}_{\leq t}$

---

In Algorithm 1, we propose to use the tree aggregation protocol to achieve differential privacy. More specifically, the tree aggregation protocol we used (Algorithm 3) is motivated by Guha Thakurta & Smith (2013); Kairouz et al. (2021). We create a binary tree (see initialization in Algorithm 3) $\mathcal{T}$ with size $2^{\lceil \log_2 N \rceil + 1} - 1$ and $N$ leaves (denoted by $m_1, \ldots, m_N$) and initializes the values associated with each node to the zero vector (see Figure 1). At $t$-th iteration, the tree aggregation protocol will first accept a new vector $\mathbf{g}$. Then, it adds the newly received vector $\mathbf{g}$ to all nodes along the path to

the root of $\mathcal{T}$ starting from the $t$-th leaf $m_t$, and adding Gaussian noise to the nodes along this path from $m_t$ to the first left child (see tree update in Algorithm 3). Finally, it retrieves the noisy sum $\widetilde{\mathbf{g}}$, which serves as the private estimate of the sum of all the previously received vectors, using the values stored in $\mathcal{T}$ (see obtain private sum in Algorithm 3).

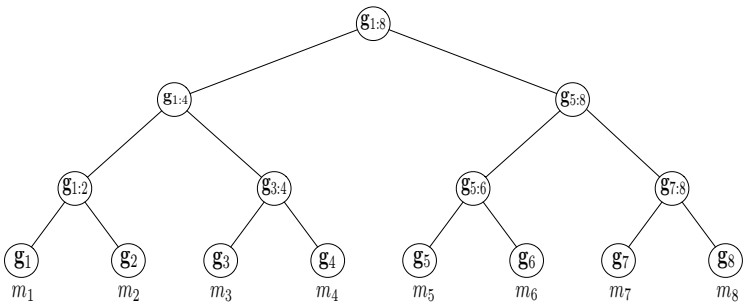

Figure 1: Illustration of a binary tree with $8$ leaf nodes in the tree aggregation protocol. $m_i$ denotes the $i$-th leaf nodes with associated value $\mathbf{g}_i$. $\mathbf{g}_{k:l} = \sum_{j=k}^{l} \mathbf{g}_j$ denotes the value associated with the internal node in the tree.

## B  PROOFS

### B.1  PROOFS OF PRIVACY GUARANTEES

In our analysis, we use the following lemma (Mironov, 2017) to convert the RDP guarantee to the $(\epsilon, \delta)$-DP.

**Lemma B.1.** If a randomized mechanism $\mathcal{M} : \mathcal{S}^n \to \mathcal{R}$ satisfies $(\alpha, \rho)$-RDP, then $\mathcal{M}$ satisfies $(\rho + \log(1/\delta)/(\alpha - 1), \delta)$-DP for all $\delta \in (0, 1)$.

Now we are ready to provide the privacy guarantees of Algorithm 1.

*Proof of Theorem 4.1.* To establish the privacy guarantees of Algorithm 1, we need to prove that $\{\mathbf{w}_t\}_{t \in [N]}$ and $\{l_t\}_{t=0}^{N-1}$ are private. Since the training dataset $\{(\mathbf{x}_i, y_i)\}_{i=0}^{N-1}$ and estimating data $\{(\widetilde{\mathbf{x}}_i, \widetilde{y}_i)\}_{i=1}^{m}$ are disjoint, we only need to provide the privacy guarantees of $\{\mathbf{w}_t\}_{t \in [N]}$ and $\{l_t\}_{t=0}^{N-1}$ separately.

We first show that the complete binary tree $\mathcal{T}$ is private since each $\widetilde{\mathbf{g}}_{\leq t}$ for $t \in [N]$ is a combination of at most $k = \lceil \log_2 N \rceil + 1$ nodes in $\mathcal{T}$. We will then show that Algorithm 2 is also differentially private. Finally, we can prove that Algorithm 1 is private due to the parallel composition of differential privacy.

Let $\{\mathbf{v}_j\}_{j=1}^{K}$ and $\{\mathbf{v}_j'\}_{j=1}^{K}$ be the values of the nodes in the post-order tree traversal generated by adding Gaussian noises (line 11 in Algorithm 3) based on the neighboring datasets $S$ and $S'$ with $i$-th data (will be stored in leaf node $m_i$ in $\mathcal{T}$) is different. Let $\{v_1, v_2, \ldots, v_K\}$ be any set of values of nodes in the post-order tree traversal of the binary tree $\mathcal{T}$. We have the joint density of $\mathbf{v}_1, \ldots, \mathbf{v}_K$ as

$$p\big(\mathbf{v}_1 = v_1, \ldots, \mathbf{v}_K = v_K\big)$$
$$= \prod_{j=1}^{K} p\big(\mathbf{v}_j = v_j | \mathbf{v}_1 = v_1, \ldots, \mathbf{v}_{j-1} = v_{j-1}\big)$$
$$= \prod_{j \in \mathbf{p}_i} p\big(\mathbf{v}_j = v_j | \mathbf{v}_1 = v_1, \ldots, \mathbf{v}_{j-1} = v_{j-1}\big) \cdot \prod_{j \notin \mathbf{p}_i} p\big(\mathbf{v}_j = v_j | \mathbf{v}_1 = v_1, \ldots, \mathbf{v}_{j-1} = v_{j-1}\big),$$

where $\mathbf{p}_i$ is the path from leaf node $m_i$ to the root of $\mathcal{T}$. Similarly, we can get the joint density of $\mathbf{v}_1', \ldots, \mathbf{v}_K'$ as

$$p\big(\mathbf{v}_1' = v_1, \ldots, \mathbf{v}_K' = v_K\big)$$

$$= \prod_{j \in \mathbf{p}_i} p\big(\mathbf{v}'_j = v_j | \mathbf{v}_1 = v_1, \ldots, \mathbf{v}'_{j-1} = v_{j-1}\big) \cdot \prod_{j \notin \mathbf{p}_i} p\big(\mathbf{v}'_j = v_j | \mathbf{v}_1 = v_1, \ldots, \mathbf{v}'_{j-1} = v_{j-1}\big).$$

Therefore, let $\mathcal{M}(S)$ and $\mathcal{M}(S')$ be the distributions of the output values of the nodes on the neighboring datasets $S$ and $S'$, we have (with a little bit of abuse of notations)

$$\begin{aligned} D_\alpha\big(\mathcal{M}(S)||\mathcal{M}(S')\big) = \frac{1}{\alpha - 1} \log \int_{\mathcal{V}} \prod_{j \in \mathbf{p}_i} & p\big(\mathbf{v}_j = v_j | \mathbf{v}_1 = v_1, \ldots, \mathbf{v}_{j-1} = v_{j-1}\big)^{1-\alpha} \\ & \times \prod_{j \notin \mathbf{p}_i} p\big(\mathbf{v}_j = v_j | \mathbf{v}_1 = v_1, \ldots, \mathbf{v}_{j-1} = v_{j-1}\big)^{1-\alpha} \\ & \times \prod_{j \in \mathbf{p}_i} p\big(\mathbf{v}'_j = v_j | \mathbf{v}'_1 = v_1, \ldots, \mathbf{v}'_{j-1} = v_{j-1}\big)^{\alpha} \\ & \times \prod_{j \notin \mathbf{p}_i} p\big(\mathbf{v}'_j = v_j | \mathbf{v}'_1 = v_1, \ldots, \mathbf{v}'_{j-1} = v_{j-1}\big)^{\alpha} dv. \end{aligned}$$

Since for $j \notin \mathbf{p}_i$, we have

$$p\big(\mathbf{v}_j = v_j | \mathbf{v}_1 = v_1, \ldots, \mathbf{v}_{j-1} = v_{j-1}\big) = p\big(\mathbf{v}'_j = v_j | \mathbf{v}'_1 = v_1, \ldots, \mathbf{v}'_{j-1} = v_{j-1}\big),$$

we can integrate over $v_j$ for $j \notin \mathbf{p}_i$ to obtain

$$\begin{aligned} D_\alpha\big(\mathcal{M}(S)||\mathcal{M}(S')\big) = \frac{1}{\alpha - 1} \log \int_{\mathcal{V}} \prod_{j \in \mathbf{p}_i} & p\big(\mathbf{v}_j = v_j | \mathbf{v}_1 = v_1, \ldots, \mathbf{v}_{j-1} = v_{j-1}\big)^{1-\alpha} \\ & \times \prod_{j \in \mathbf{p}_i} p\big(\mathbf{v}'_j = v_j | \mathbf{v}'_1 = v_1, \ldots, \mathbf{v}'_{j-1} = v_{j-1}\big)^{\alpha} dv. \end{aligned}$$

(B.1)

Notice that each node along the path $\mathbf{p}_i$ is a summation query and will only be affected by its leaf nodes. Therefore, the sensitivity of each node along this path is upper bounded by $2\psi$ where $\psi = \max\{\psi_1, \ldots, \psi_j\}$ and $\psi_1, \ldots, \psi_j$ is the norm bound of its leaf nodes. As a result, according to Lemma 2.6, line 11 in Algorithm 3 can ensure that each node along path $\mathbf{p}_i$ is $(\alpha, \alpha/\tau^2)$-RDP for all $i \in [N]$ (recall the definition of $\sigma^2 = \tau^2\psi^2$ in Algorithm 1). Therefore, for each integral in (B.1), we have

$$\begin{aligned} \int_{\mathcal{V}_j} & p\big(\mathbf{v}_j = v_j | \mathbf{v}_1 = v_1, \ldots, \mathbf{v}_{j-1} = v_{j-1}\big)^{1-\alpha} p\big(\mathbf{v}'_j = v_j | \mathbf{v}'_1 = v_1, \ldots, \mathbf{v}'_{j-1} = v_{j-1}\big)^{\alpha} dv_j \\ & \leq \exp\big(\alpha(\alpha - 1)/(2\tau^2)\big). \end{aligned}$$

Plugging the above result into (B.1), we can obtain

$$D_\alpha\big(\mathcal{M}(S)||\mathcal{M}(S')\big) \leq \frac{1}{\alpha - 1} \log\bigg[ \prod_{j \in \mathbf{p}_i} \exp(\alpha(\alpha - 1)/2) \bigg] \leq \frac{k\alpha}{2\tau^2},$$

where $k = \lceil \log_2 N \rceil + 1$. Thus, we show that the tree aggregation protocol is $(\alpha, \alpha k/\tau^2)$-RDP. Therefore, $\overline{\mathbf{w}}_N$ is $(\alpha, \alpha\rho)$-RDP, where $\rho = k/\tau^2$.

Next, we translate the RDP to DP. According to Lemma B.1, $\overline{\mathbf{w}}_N$ is $(\epsilon, \delta)$-DP with $\epsilon = \alpha\rho + \log(1/\delta)/(\alpha - 1)$. Therefore, we can choose $\alpha = 1 + \sqrt{\log(1/\delta)/\rho}$ to get the smallest $\epsilon = \rho + 2\sqrt{\log(1/\delta)\rho}$. Thus, we can obtain $\rho = \big(\sqrt{\log(1/\delta) + \epsilon} - \sqrt{\log(1/\delta)}\big)^2 = k/\tau^2$. Therefore, let

$$\tau^2 = \frac{k}{\rho} \leq \frac{4k\big(\log(1/\delta) + \epsilon\big)}{\epsilon^2},$$

the tree aggregation protocol is $(\epsilon, \delta)$-DP.

Next, we show that $\{l_t\}_{t=0}^{N-1}$ is differentially private. According to Lemma 2.3 in Karwa & Vadhan (2017), if we choose $\widetilde{\epsilon} = \epsilon/\sqrt{8N\log(2/\delta)}$ and $\widetilde{\delta} = \delta/(2N)$, then Algorithm 2 is $(\epsilon/\sqrt{8N\log(2/\delta)}, \delta/(2N))$-DP. Therefore, by the advanced composition result (Kairouz et al., 2015), we have that $\{l_t\}_{t=0}^{N-1}$ is $(\epsilon, \delta)$-DP. $\qquad \square$

## B.2 Proofs of Utility Guarantees

### B.2.1 Upper bounds on the stochastic gradient

To establish the utility guarantee of Algorithm 1, we will first show that the event $E = \{\|\mathbf{x}_t\mathbf{x}_t^\top\mathbf{w}_t - \mathbf{x}_ty_t\|_{\mathbf{\Sigma}^{-1}} \leq \psi_t, \forall 0 \leq t \leq N-1\}$ holds with high probability, where $\psi_t = Cl_t$, $l_t$ is the output of Algorithm 2, and $C = 3\sqrt{2\gamma_1\gamma_4\operatorname{tr}(\mathbf{\Sigma}^{-1}\mathbf{H})}\log N$. Therefore, under the event $E$, we do not need to perform any clipping procedure, and line 5 in Algorithm 1 reduces to $\mathbf{g}_t = \mathbf{x}_t\mathbf{x}_t^\top\mathbf{w}_t - \mathbf{x}_ty_t$. Therefore, we can directly use the learning guarantee results derived in the next subsection, i.e., section B.2.2. To prove the event $E$ holds, we need following lemmas

**Lemma B.2.** Under Assumption 2.2, we have

- (*Bounded norm*) There exist positive constants $\gamma_1 = 2\sigma_z^2$, such that for any PSD matrix $\mathbf{\Sigma}$, the following holds with probability at least $1 - c_1\beta$

$$\left\|\mathbf{\Sigma}^{-\frac{1}{2}}\mathbf{x}\right\|_2^2 \leq \gamma_1\operatorname{tr}(\mathbf{\Sigma}^{-1}\mathbf{H})\log(1/\beta),$$

  where $c_1$ is an absolute constant.

- (*Bounded product*) There exist positive constants $\gamma_2 = \sigma_z^2$, such that for any fixed vector $\mathbf{v}$, with probability at least $1 - c_2\beta$

$$\langle\mathbf{x}, \mathbf{v}\rangle^2 \leq \gamma_2\mathbf{v}^\top\mathbf{H}\mathbf{v}\log(1/\beta),$$

  where $c_2$ is an absolute constant.

In addition, under Assumption 2.3, we have

- (*Bounded noise*) There exists positive constant $\gamma_3 = \sigma_\zeta^2$, such that the following holds with probability at least $1 - c_3\beta$

$$\zeta^2 \leq \gamma_3\log(1/\beta),$$

  where $c_3$ is an absolute constant.

**Lemma B.3.** For Algorithm 2 with option 1, if we have $m = \Omega(\log N)$, then we have $l^2 \geq \|\mathbf{w}^* - \mathbf{w}\|_{\mathbf{H}}^2 + \sigma_\zeta^2$ holds with probability at least $1 - 1/N^{10}$. Algorithm 2 with option 2 is $(\widetilde{\epsilon}, \widetilde{\delta})$-DP. In addition, if we have $m = \Omega(\beta_1\log(\beta_2/\widetilde{\delta})\log N/\widetilde{\epsilon})$, then with probability at least $1 - 1/\beta_2$, $\|\mathbf{w}^* - \mathbf{w}\|_{\mathbf{H}}^2 + \sigma_\zeta^2 \leq l^2 \leq 2(\|\mathbf{w}^* - \mathbf{w}\|_{\mathbf{H}}^2 + \sigma_\zeta^2)$.

Now we are ready to prove the results. We have

$$\|\mathbf{x}_t\mathbf{x}_t^\top\mathbf{w}_t - \mathbf{x}_ty_t\|_{\mathbf{\Sigma}^{-1}} \leq \|\mathbf{x}_t\|_{\mathbf{\Sigma}^{-1}} \cdot \ell^{1/2}(\mathbf{w}_t; \mathbf{x}_i, y_i).$$

By Lemma B.2, we have $\|\mathbf{x}_t\|_{\mathbf{\Sigma}^{-1}}^2 \leq \gamma_1\operatorname{tr}(\mathbf{\Sigma}^{-1}\mathbf{H})\log(1/\beta)$ holds with high probability. In addition, we have

$$\ell^{1/2}(\mathbf{w}_t; \mathbf{x}_t, y_t) = \mathbf{x}_t^\top(\mathbf{w}_t - \mathbf{w}^*) + \zeta_t \leq \sqrt{2(\mathbf{x}_t^\top(\mathbf{w}_t - \mathbf{w}^*))^2 + 2\zeta_t^2}.$$

According to Lemma B.2, we have with high probability

$$(\mathbf{x}_t^\top(\mathbf{w}_t - \mathbf{w}^*))^2 \leq \gamma_2\log(1/\beta)\|\mathbf{w}_t - \mathbf{w}^*\|_{\mathbf{H}}^2.$$

Additionally, due to Lemma B.2, we have with high probability

$$\zeta_t^2 \leq \gamma_3\log(1/\beta).$$

Therefore, we have

$$\ell^{1/2}(\mathbf{w}_t; \mathbf{x}_t, y_t) \leq \sqrt{2(\mathbf{x}_t^\top(\mathbf{w}_t - \mathbf{w}^*))^2 + 2\zeta_t^2} \leq \sqrt{2\gamma_4\log(1/\beta)(\|\mathbf{w}_t - \mathbf{w}^*\|_{\mathbf{H}}^2 + \sigma_\zeta^2)},$$

where $\gamma_4 = \max\{\gamma_2, 1\}$. Next, we need to estimate the upper bound of $\|\mathbf{w}_t - \mathbf{w}^*\|_{\mathbf{H}}^2 + \sigma_\zeta^2$ at each iteration. To this end, we propose to use Algorithm 2. Thus, according to Lemma B.3, we have with high probability $\|\mathbf{w}_t - \mathbf{w}^*\|_{\mathbf{H}}^2 + \sigma_\zeta^2 \leq l_t^2$. As a result, we have

$$
\begin{aligned}
\|\mathbf{x}_t\mathbf{x}_t^\top\mathbf{w}_t - \mathbf{x}_ty_t\|_{\mathbf{\Sigma}^{-1}}^2 &\leq \|\mathbf{x}_t\|_{\mathbf{\Sigma}^{-1}}^2 \cdot \ell(\mathbf{w}_t; \mathbf{x}_i, y_i) \\
&\leq \gamma_1\operatorname{tr}(\mathbf{\Sigma}^{-1}\mathbf{H})\log(1/\beta) \cdot 2\gamma_4\log(1/\beta)(\|\mathbf{w}_t - \mathbf{w}^*\|_{\mathbf{H}}^2 + \sigma_\zeta^2) \\
&\leq 2l_t^2\gamma_1\gamma_4\log^2(1/\beta)\operatorname{tr}(\mathbf{\Sigma}^{-1}\mathbf{H}).
\end{aligned}
$$

Thus, if we choose $C^2 = 2\gamma_1\gamma_4\log^2(1/\beta)\operatorname{tr}(\mathbf{\Sigma}^{-1}\mathbf{H})$, set $\beta = 1/N^3$, by union bound, the event $E$ holds with probability at least $1 - 1/N$.

### B.2.2 LEARNING GUARANTEES

**Connection between DP-SGD and DP-FTRL.** Next, we will establish the connection between Algorithm 1 and DP-SGD, and thus use the technique developed in Zou et al. (Zou et al., 2021b). According to line 9 in Algorithm 1, we have

$$\mathbf{w}_{t+1} = \operatorname*{argmin}_{\mathbf{w}} \langle \widetilde{\mathbf{g}}_{\leq t}, \mathbf{w} \rangle + \frac{1}{2\eta} \|\mathbf{w}\|_2^2, \tag{B.2}$$

where $\widetilde{\mathbf{g}}_{\leq t}$ is the differentially private estimate of $\sum_{i=0}^{t} \nabla \ell(\mathbf{w}_i; \mathbf{x}_i, y_i)$ generated by the tree aggregation protocol (see Algorithm 3). It has been observed (Kairouz et al., 2021) that $\widetilde{\mathbf{g}}_{\leq t} = \sum_{i=0}^{t} \nabla \ell(\mathbf{w}_i; \mathbf{x}_i, y_i) + \boldsymbol{\nu}_t$, where $\boldsymbol{\nu}_t$ is a combination of at most $O(\lceil \log_2 t \rceil)$ independent random Gaussian vectors. Plugging this form in to (B.2), we have

$$\mathbf{w}_{t+1} = \operatorname*{argmin}_{\mathbf{w}} \sum_{i=0}^{t} \langle \nabla \ell(\mathbf{w}_i; \mathbf{x}_i, y_i), \mathbf{w} \rangle + \langle \boldsymbol{\nu}_t, \mathbf{w} \rangle + \frac{1}{2\eta} \|\mathbf{w}\|_2^2. \tag{B.3}$$

Since $\mathbf{w}_0 = \mathbf{0}$, we can obtain

$$\mathbf{w}_{t+1} = \mathbf{w}_t - \eta \cdot [\nabla \ell(\mathbf{w}_t; \mathbf{x}_i, y_i) + \boldsymbol{\xi}_t], \tag{B.4}$$

where $\boldsymbol{\xi}_t = \boldsymbol{\nu}_t - \boldsymbol{\nu}_{t-1}, \boldsymbol{\nu}_{-1} = \mathbf{0}$. According to (B.4), we have the update rule of DP-FTRL as

$$\mathbf{w}_{t+1} = \mathbf{w}_t - \eta \cdot [\nabla \ell(\mathbf{w}_t; \mathbf{x}_i, y_i) + \boldsymbol{\xi}_t] = \mathbf{w}_t - \eta \cdot [\mathbf{x}_i \mathbf{x}_i^\top \mathbf{w}_t - \mathbf{x}_i y_i + \boldsymbol{\xi}_t]$$

where $\boldsymbol{\xi}_t = \boldsymbol{\nu}_t - \boldsymbol{\nu}_{t-1}, \boldsymbol{\nu}_{-1} = \mathbf{0}$. In addition, we have $y = \langle \mathbf{w}^*, \mathbf{x} \rangle + \epsilon$. As a result, we can perform a similar bias-variance decomposition in Zou et al. (Zou et al., 2021b). Note that we will focus on the outer product of the error vector $\mathbf{w}_t - \mathbf{w}^*$:

$$\mathrm{Err}_t := \mathbb{E}_{\mathrm{Alg}} \big[ (\mathbf{w}_t - \mathbf{w}^*)(\mathbf{w}_t - \mathbf{w}^*)^\top \big].$$

Besides, we also introduce the following tensor operations on the matrix space that will be repeatedly applied in the theoretical analysis:

**Definition B.4.** Given stepsize $\eta \lesssim 1/\operatorname{tr}(\mathbf{H})$, we define

$$\mathcal{I} = \mathbf{I} \otimes \mathbf{I}, \ \mathcal{M} = \mathbb{E}[\mathbf{x}^{\otimes 4}], \ \widetilde{\mathcal{M}} = \mathbf{H} \otimes \mathbf{H}.$$

Besides, given any PSD matrix $\mathbf{A}$, we define the operators $\mathcal{T}$ and $\widetilde{\mathcal{T}}$ that satisfies

$$(\mathcal{I} - \eta \mathcal{T}) \circ \mathbf{A} = \mathbb{E}[(\mathbf{I} - \eta \mathbf{x} \mathbf{x}^\top) \mathbf{A} (\mathbf{I} - \eta \mathbf{x} \mathbf{x}^\top)], \ (\mathcal{I} - \eta \widetilde{\mathcal{T}}) \circ \mathbf{A} = \mathbb{E}[(\mathbf{I} - \eta \mathbf{H}) \mathbf{A} (\mathbf{I} - \eta \mathbf{H})].$$

Then it can be also verified that

$$(\mathcal{I} - \eta \mathcal{T}) \circ \mathbf{A} = (\mathcal{I} - \eta \widetilde{\mathcal{T}}) \circ \mathbf{A} + (\mathcal{M} - \widetilde{\mathcal{M}}) \circ \mathbf{A}.$$

Based on the above tensor operations, the following theorem provides the bias-variance decomposition of $\mathrm{Err}_t$ as well as their updated forms.

**Theorem B.5.** Under Assumptions 2.1 and 2.3, let $\beta = 16\sigma_z^2$, the error covariance $\mathrm{Err}_t$ can be decomposed as follows,

$$\mathrm{Err}_t = \mathbf{B}_t + \mathbf{C}_t,$$

where

$$\mathbf{B}_t = (\mathcal{I} - \eta \mathcal{T}) \circ \mathbf{B}_{t-1}, \quad \mathbf{B}_0 = (\mathbf{w}_0 - \mathbf{w}^*)(\mathbf{w}_0 - \mathbf{w}^*)^\top;$$
$$\mathbf{C}_t = (\mathcal{I} - \eta \mathcal{T}) \circ \mathbf{C}_{t-1} + \eta^2 \mathbf{G}_t, \quad \mathbf{C}_0 = \mathbf{0}, \ \mathbf{G}_t \preceq \sigma_\zeta^2 \mathbf{H} + O\big(\mathbb{E}\big[\psi(t)^2 + \psi(t)^4\big]\tau^2 \log_2 N\big)\boldsymbol{\Sigma} + \mathbf{D},$$

where

$$\mathbf{D} \preceq O\big(\tau^2 \log_2 N\big) \bigg\{ \bigg[ \sum_{j \in \mathcal{S}(t)} (\mathcal{I} - \eta \widetilde{\mathcal{T}})^{t-j+1} \bigg] \circ \boldsymbol{\Sigma} + O(\eta \beta \log_2 N) \cdot \operatorname{tr}(\mathbf{H}\boldsymbol{\Sigma}) \cdot \big(\mathbf{I} - (\mathbf{I} - \eta \mathbf{H})^N\big) \bigg\},$$

$\mathcal{S}(t)$ is a index set with $|\mathcal{S}(t)| \leq \lceil \log_2 N \rceil + 1$.

Next, we will establish the guarantee on the excess risk $\mathbb{E}[L_D(\bar{\mathbf{w}})] - L_D(\mathbf{w}^*)$. In particular, this will be performed based on the update forms of bias covariance $\mathbf{B}_t$ and variance covariance $\mathbf{C}_t$ in Theorem B.5. First, note that we consider the iterate average as the algorithm output, the following lemma provides the bound on the excess risk:

**Lemma B.6** (Lemma B.3 in Zou et al. (Zou et al., 2021b)). Let $\bar{\mathbf{w}}_N = 1/N \sum_{t=0}^{N-1} \mathbf{w}_t$, then if the stepsize satisfies $\eta \leq 1/\lambda_1(\mathbf{H})$, we have

$$\mathbb{E}[L_D(\bar{\mathbf{w}}_N)] - L_D(\mathbf{w}^*) \leq \underbrace{\frac{1}{N^2} \sum_{t=0}^{N-1} \sum_{k=t}^{N-1} \langle (\mathbf{I} - \gamma\mathbf{H})^{k-t}\mathbf{H}, \mathbf{B}_t \rangle}_{\text{bias}} + \underbrace{\frac{1}{N^2} \sum_{t=0}^{N-1} \sum_{k=t}^{N-1} \langle (\mathbf{I} - \gamma\mathbf{H})^{k-t}\mathbf{H}, \mathbf{C}_t \rangle}_{\text{variance}}.$$

**Upper bound for the variance error.** The key to proving the upper bound of the variance error is to establish a sharp characterization of $\mathbf{C}_t$. Recall the update rule of $\mathbf{C}_t$:

$$\mathbf{C}_t = (\mathcal{I} - \eta\mathcal{T}) \circ \mathbf{C}_{t-1} + \eta^2 \mathbf{G}_t, \quad \mathbf{C}_0 = \mathbf{0}.$$

Then, we would like to remark that although the above update rule is similar to that of the standard SGD, the proof technique in Zou et al. (Zou et al., 2021b) cannot be applied. In particular, Zou et al. (Zou et al., 2021b) first proves an upper bound for $\mathbf{C}_\infty$ and uses this quantity to control the update of $\mathbf{C}_t$. However, when introducing the additive noise, the matrix $\mathbf{G}_t$ will no longer be well aligned with $\mathbf{H}$ so that $\mathbf{C}_\infty$ will explode (e.g., becoming $\mathbf{H}^{-1}\mathbf{\Sigma}$). To this end, rather than leveraging $\mathbf{C}_\infty$ as a reference to control the dynamics of $\mathbf{C}_t$, we consider to use a sharper reference $\mathbf{C}_N$. The following lemma gives a "rough" bound on $\mathbf{C}_N$.

**Lemma B.7.** Let $\mathbf{G}$ be an union upper bound on $\mathbf{G}_0, \ldots, \mathbf{G}_{N-1}$, then under Assumption 2.1, if the step size satisfies $\eta \lesssim 1/(\operatorname{tr}(\mathbf{H}) \log N)$, let $\beta = 16\sigma_z^4$, it holds that

$$\mathbf{C}_0 \preceq \cdots \preceq \mathbf{C}_N \preceq \frac{\beta\eta^2}{1 - \beta\eta \operatorname{tr}(\mathbf{H})} \cdot \langle \mathbf{I} - (\mathbf{I} - \eta\mathbf{H})^N, \mathbf{G} \rangle \cdot (\mathbf{I} - (\mathbf{I} - \eta\mathbf{H})^N)$$

$$+ \eta^2 \cdot \sum_{t=0}^{N-1} (\mathcal{I} - \eta\widetilde{\mathcal{T}})^t \circ \mathbf{G}.$$

Given this rough bound of $\mathbf{C}_t$ for all $t \leq N$, we can then get the following bound on the variance error.

**Lemma B.8.** If the step size satisfies $\eta \lesssim 1/(\operatorname{tr}(\mathbf{H}) \log N)$, then let $\beta = 16\sigma_z^2$, the following holds for any $k \geq 1$,

$$\text{variance} \lesssim \frac{\beta\eta}{N} \cdot \langle \mathbf{I}_{0:k} + N\eta\mathbf{H}_{k:\infty}, \bar{\mathbf{G}} \rangle \cdot \Big( k + N^2\eta^2 \sum_{i>k} \lambda_i^2 \Big) + \frac{1}{N} \cdot \langle \mathbf{H}_{0:k}^{-1} + N^2\eta^2 \mathbf{H}_{k:\infty}, \bar{\mathbf{G}} \rangle,$$

where

$$\bar{\mathbf{G}} = \sigma_\zeta^2 \cdot \mathbf{H} + \tau^2\beta \cdot (\log_2 N)^2 \eta \operatorname{tr}(\mathbf{H}\mathbf{\Sigma}) \cdot (\mathbf{I}_{0:k} + N\eta\mathbf{H}_{k:\infty})$$

$$+ \big( \mathbb{E}[\psi(N)^2 + \psi(N)^4] + 1 \big) \cdot \tau^2 \cdot \log_2 N \cdot \mathbf{\Sigma}.$$

**Upper bound for the bias error.** By Lemma B.11 in Zou et al. (Zou et al., 2021b), the upper bound on the bias error can be directly obtained, which is summarized in the following lemma.

**Lemma B.9.** If the step size satisfies $\eta \leq 1/\lambda_1$, then it holds that for any $k \geq 1$,

$$\text{bias} \leq \frac{1}{\eta^2 N^2} \cdot \|\mathbf{w}_0 - \mathbf{w}^*\|_{\mathbf{H}_{0:k}^{-1}}^2 + \|\mathbf{w}_0 - \mathbf{w}^*\|_{\mathbf{H}_{k:\infty}}^2$$

$$+ \frac{2\beta \big( \|\mathbf{w}_0 - \mathbf{w}^*\|_{\mathbf{I}_{0:k}}^2 + N\eta\|\mathbf{w}_0 - \mathbf{w}^*\|_{\mathbf{H}_{k:\infty}}^2 \big)}{N\eta(1 - \beta\eta \operatorname{tr}(\mathbf{H}))} \cdot \Big( \frac{k}{N} + N\eta^2 \sum_{i>k} \lambda_i^2 \Big). \tag{B.5}$$

### B.3 PROOFS FOR COROLLARIES

In this section, we provide proofs of our corollaries.

### B.3.1 Proof of Corollary 4.3

*Proof of Corollary 4.3.* In addition, according to Lemma B.8, we have

$$\text{variance} \leq \frac{\beta\eta}{N} \cdot \langle \mathbf{I}_{0:k} + N\eta\mathbf{H}_{k:\infty}, \, \bar{\mathbf{G}} \rangle \cdot (k + N^2\eta^2 \sum_{i>k} \lambda_i^2) + \frac{1}{N} \cdot \langle \mathbf{H}_{0:k}^{-1} + N^2\eta^2\mathbf{H}_{k:\infty}, \, \bar{\mathbf{G}} \rangle,$$

where $k \geq 1$, and

$$\bar{\mathbf{G}} = \sigma_\zeta^2 \cdot \mathbf{H} + \tau^2\beta \cdot (\log_2 N)^2 \eta \operatorname{tr}(\mathbf{H}\boldsymbol{\Sigma}) \cdot (\mathbf{I}_{0:k} + N\eta\mathbf{H}_{k:\infty})$$
$$+ \left( \mathbb{E}\left[ \psi(N)^2 + \psi(N)^4 \right] + 1 \right) \cdot \tau^2 \cdot \log_2 N \cdot \boldsymbol{\Sigma}.$$

and $\psi(N)^2 = 18l\gamma_1\gamma_4 \log^2(N) \operatorname{tr}(\boldsymbol{\Sigma}^{-1}\mathbf{H})$, $l = \max\{l_0^2, l_1^2, \ldots, l_{N-1}^2\}$. Therefore, we can get (ignore some constant parameters)

$$\bar{\mathbf{G}} \preceq O(\text{polylog}(N)) \cdot \left[ \sigma_\zeta^2\mathbf{H} + \tau^2\eta \operatorname{tr}(\mathbf{H}\boldsymbol{\Sigma})\mathbf{I} + l^2\tau^2 \operatorname{tr}(\boldsymbol{\Sigma}^{-1}\mathbf{H})^2\boldsymbol{\Sigma} \right]$$
$$\preceq O(\text{polylog}(N)) \cdot \left[ \sigma_\zeta^2\mathbf{H} + \tau^2\mathbf{I} + l^2\tau^2 \operatorname{tr}(\boldsymbol{\Sigma}^{-1}\mathbf{H})^2\boldsymbol{\Sigma} \right],$$

where the second line comes from $\eta \leq 1/(\beta \operatorname{tr}(\mathbf{H}\boldsymbol{\Sigma}))$. As a result, we have

$$\text{variance} \lesssim \frac{\beta\eta}{N} \cdot \langle \mathbf{I}_{0:k} + N\eta\mathbf{H}_{k:\infty}, \, \sigma_\zeta^2\mathbf{H} \rangle \cdot (k + N^2\eta^2 \sum_{i>k} \lambda_i^2) + \frac{1}{N} \cdot \langle \mathbf{H}_{0:k}^{-1} + N^2\eta^2\mathbf{H}_{k:\infty}, \, \sigma_\zeta^2\mathbf{H} \rangle$$

$$+ \frac{\beta\eta}{N} \cdot \langle \mathbf{I}_{0:k} + N\eta\mathbf{H}_{k:\infty}, \, \tau^2\mathbf{I} + l^2\tau^2 \operatorname{tr}(\boldsymbol{\Sigma}^{-1}\mathbf{H})^2\boldsymbol{\Sigma} \rangle \cdot (k + N^2\eta^2 \sum_{i>k} \lambda_i^2)$$

$$+ \frac{1}{N} \cdot \langle \mathbf{H}_{0:k}^{-1} + N^2\eta^2\mathbf{H}_{k:\infty}, \, \tau^2\mathbf{I} + l^2\tau^2 \operatorname{tr}(\boldsymbol{\Sigma}^{-1}\mathbf{H})^2\boldsymbol{\Sigma} \rangle. \tag{B.6}$$

Note that the first line on the right hand side corresponds to the nonprivate variance and the remaining part corresponds to the error introduced by the private mechanism. Note that we have $\tau^2 \leq \frac{4\left( \lceil \log_2 N \rceil + 1 \right)\left( \log(1/\delta) + \epsilon \right)}{\epsilon^2}$. When $\epsilon$ goes to infinity, which means we do not have any privacy guarantee, then the variance error reduces to the non-private one. In the following discussion, we assume $\beta = 16\sigma_z^2 = O(1)$, $\operatorname{tr}(\mathbf{H}) \leq \rho_1$ and $\lambda_1 \leq 1$, where $\rho_1 = \Theta(1)$, and we mainly focus on the error introduced by the private mechanism, denoted by as it will dominate the non-private part (note that $\mathbf{H} \preceq \mathbf{I}$).

First, we consider the nonprivate part. According to (B.6), noting that $\beta$ is in the constant order, we have

$$\text{variance}_{\text{np}} \lesssim \frac{\eta}{N} \cdot \langle \mathbf{I}_{0:k} + N\eta\mathbf{H}_{k:\infty}, \, \sigma_\zeta^2\mathbf{H} \rangle \cdot (k + N^2\eta^2 \sum_{i>k} \lambda_i^2) + \frac{1}{N} \cdot \langle \mathbf{H}_{0:k}^{-1} + N^2\eta^2\mathbf{H}_{k:\infty}, \, \sigma_\zeta^2\mathbf{H} \rangle. \tag{B.7}$$

Then we set $k^* = \max\{k : \lambda_k \geq 1/(N\eta)\}$, we have $\mathbf{I}_{0:k} + N\eta\mathbf{H}_{k:\infty} \preceq \mathbf{I}$ and then

$$\text{variance}_{\text{np}} \lesssim \frac{\sigma_\zeta^2\eta}{N} \cdot \operatorname{tr}(\mathbf{H}) \cdot (k^* + N^2\eta^2 \sum_{i>k^*} \lambda_i^2) + \frac{\sigma_\zeta^2}{N} \cdot (k^* + N^2\eta^2 \sum_{i>k^*} \lambda_i^2)$$

$$\lesssim \frac{\sigma_\zeta^2}{N} \cdot (k^* + N^2\eta^2 \sum_{i>k^*} \lambda_i^2),$$

where we use the fact that $\eta \operatorname{tr}(\mathbf{H}) \leq 1$ in the last inequality.

In addition, according to (B.5) and the assumption that $\|\mathbf{w}_0 - \mathbf{w}^*\|_2 \leq B$, it holds that

$$\text{bias} \lesssim \frac{B^2}{\eta^2 N^2 \lambda_{k^*}} + B^2\lambda_{k^*+1} + \frac{2\left( B^2 + N\eta\lambda_{k^*+1}B^2 \right)}{N\eta(1 - \eta \operatorname{tr}(\mathbf{H}))} \cdot \left( \frac{k^*}{N} + N\eta^2 \sum_{i>k^*} \lambda_i^2 \right)$$

$$\lesssim \frac{B^2}{N\eta} + \frac{B^2}{N^2\eta} \cdot \left( k^* + N^2\eta^2 \sum_{i>k^*} \lambda_i^2 \right)$$

$$\lesssim \frac{B^2}{N\eta} + \frac{B^2}{N^2\eta} \cdot \left(k^* + N\eta \sum_{i>k^*} \lambda_i\right), \tag{B.8}$$

where the last line is due to the fact that $\lambda_i \leq 1/(N\eta)$ for all $i > k^*$.

Combining (B.7) and (B.8), we have

$$\text{err}_{\text{np}} \lesssim \frac{\sigma_\zeta^2}{N} \cdot (k^* + N^2\eta^2 \sum_{i>k^*} \lambda_i^2) + \frac{B^2}{N\eta} + \frac{B^2}{N^2\eta} \cdot \left(k^* + N\eta \sum_{i>k^*} \lambda_i\right) \tag{B.9}$$

$$\eqsim \frac{B^2}{N\eta} + \left(\frac{\sigma_\zeta^2}{N} + \frac{B^2}{N^2\eta}\right) \cdot \left(k^* + N\eta \sum_{i>k^*} \lambda_i\right). \tag{B.10}$$

On the other hand, according to (B.6), we have

$$\text{variance}_{\text{p}} \lesssim \frac{l^2\tau^2}{N} \cdot \left[\eta \cdot \left[k^* + N\eta \operatorname{tr}(\mathbf{H}_{k^*:\infty})\right] \cdot \left(k^* + N^2\eta^2 \sum_{i>k^*} \lambda_i^2\right)\right.$$

$$\left. + \left[\operatorname{tr}(\mathbf{H}_{0:k^*}^{-1}) + N^2\eta^2 \operatorname{tr}(\mathbf{H}_{k^*:\infty})\right]\right]$$

$$\lesssim \frac{l^2\tau^2}{N} \cdot \left[\eta \cdot \left(k^* + N\eta \sum_{i>k^*} \lambda_i\right)^2 + N\eta\left(k^* + N\eta \sum_{i>k^*} \lambda_i\right)\right], \tag{B.11}$$

where the second inequality holds since $\operatorname{tr}(\mathbf{H}_{0:k^*}^{-1}) = \sum_{i=1}^{k^*} \lambda_i^{-1} \leq N\eta \cdot k^*$. Further note that we have assumed $\eta \leq 1/\operatorname{tr}(\mathbf{H})$, thus

$$k^* + N\eta \sum_{i>k^*} \lambda_i \leq N\eta \sum_i \lambda_i \leq N. \tag{B.12}$$

Therefore, (B.11) becomes

$$\text{variance}_{\text{p}} \lesssim l^2\tau^2\eta \cdot \left(k^* + N\eta \sum_{i>k^*} \lambda_i\right).$$

Combine the above result with (B.11), we can get that

$$\text{err}_{\text{p}} \lesssim \text{err}_{\text{np}} + \text{variance}_{\text{p}}$$

$$\lesssim \underbrace{\frac{B^2}{N\eta} + \left(\frac{\sigma_\zeta^2}{N} + \frac{B^2}{N^2\eta}\right) \cdot \left(k^* + N\eta \sum_{i>k^*} \lambda_i\right)}_{\text{Non-private error component}} + \underbrace{l^2\tau^2\eta \cdot \left(k^* + N\eta \sum_{i>k^*} \lambda_i\right)}_{\text{Private error component}}.$$

This completes the proof. $\qquad\square$

### B.3.2 PROOF OF COROLLARY 4.4

*Proof of Corollary 4.4.* If $\lambda_i = i^{-r}$, we can immediately get that $k^* = (N\eta)^{\frac{1}{r}}$ and

$$k^* + N\eta \sum_{i>k^*} \lambda_i \eqsim (N\eta)^{\frac{1}{r}} + N\eta \cdot (N\eta)^{\frac{1-r}{r}} \eqsim (N\eta)^{\frac{1}{r}}.$$

Therefore, we can get

$$\text{err}_{\text{p}} \lesssim \left(\frac{\sigma_\zeta^2}{N} + \frac{B^2}{N^2\eta}\right) \cdot (N\eta)^{\frac{1}{r}} + \frac{B^2}{N\eta} + l^2\tau^2\eta \cdot (N\eta)^{\frac{1}{r}}.$$

Note that we can choose the following step size

$$\eta \eqsim \min\left\{N^{-\frac{1}{1+r}}, N^{-\frac{1+r}{1+2r}} \tau^{-\frac{2r}{1+2r}}\right\},$$

we can get

$$\mathrm{err_p} \lesssim (\sigma_\zeta^2 + B^2) \cdot N^{-\frac{r}{1+r}} + (B^2 + l^2) \cdot (\tau^{-2}N)^{-\frac{r}{1+2r}}.$$

Therefore, we have

$$\mathrm{err_p} = \widetilde{\mathcal{O}}\Big(N^{-\frac{r}{1+r}} + \tau^{\frac{2r}{1+2r}} N^{-\frac{r}{1+2r}}\Big)$$
$$= \widetilde{\mathcal{O}}\Big(N^{-\frac{r}{1+r}} \cdot \Big(1 + \big(\tau^2 N^{\frac{r}{1+r}}\big)^{\frac{r}{1+2r}}\Big)\Big).$$

Note that $\tau^2 = \widetilde{\mathcal{O}}(1/\epsilon^2)$. Therefore, to achive the nonprivate rate, we need to pay for the extra privacy at the order of $\epsilon = \widetilde{\mathcal{O}}\Big(N^{\frac{r}{2+2r}}\Big)$.

If $\lambda_i = e^{-i}$, we can get $k^* = \log(N\eta)$ and then

$$k^* + N\eta \sum_{i > k^*} \lambda_i \asymp \log(N\eta) + N\eta \cdot (N\eta)^{-1} \asymp \log(N\eta).$$

Then, the excess risk bound becomes

$$\mathrm{err_p} \lesssim \left(\frac{\sigma_\zeta^2}{N} + \frac{B^2}{N^2\eta}\right) \cdot \log(N\eta) + \frac{B^2}{N\eta} + l^2\tau^2\eta \cdot \log(N\eta)$$
$$\lesssim \frac{\sigma_\zeta^2}{N} + \frac{B^2}{N\eta} + l^2\tau^2\eta.$$

Then we can choose the following step size

$$\eta \asymp \min\big\{1, (N\tau^2)^{-1/2}\big\},$$

and obtain

$$\mathrm{err_p} \lesssim \frac{1}{N} + (\tau^{-2}N)^{-1/2} = \widetilde{\mathcal{O}}\Big(N^{-1}\big(1 + (\epsilon^{-2}N)^{1/2}\big)\Big),$$

where we use the fact that $\tau^2 = \widetilde{\mathcal{O}}(1/\epsilon)$ in the last equality. This completes the proof. $\qquad\square$

### B.3.3 PROOF OF COROLLARY 4.5

*Proof of Corollary 4.5.* In this case, we consider the following design of $\mathbf{\Sigma}$:

$$\mathbf{\Sigma} = M^{-1}(\widetilde{\mathbf{X}}\widetilde{\mathbf{X}}^\top + \lambda\mathbf{I}).$$

Then we first give the following two useful lemmas.

**Lemma B.10.** Under Assumption 2.2, let denote $\widehat{k} := \min_k\{k : \lambda_{k+1}M \le \lambda + \sum_{i>k}\lambda_i\}$ and set $\mathbf{\Sigma} = M^{-1}(\lambda\mathbf{I} + \widetilde{\mathbf{X}}\widetilde{\mathbf{X}}^\top)$ for some positive constant $\lambda > 0$, then with probability at least $1 - \exp(-\Omega(M))$,

$$\mathrm{tr}(\mathbf{\Sigma}^\dagger\mathbf{H}) \lesssim \left(\widehat{k} \cdot \frac{\lambda + \sum_{j>\widehat{k}}\lambda_j}{\lambda} + \sum_{i>\widehat{k}} \frac{M\lambda_i}{\lambda}\right).$$

**Lemma B.11.** Let $\mathbf{H} = \sum_i \lambda_i\mathbf{v}_i\mathbf{v}_i^\top$ be the eigen-decomposition of $\mathbf{H}$, then for any PSD matrix $\mathbf{A}$ that can be decomposed as $\mathbf{A} = \sum_i \mu_i\mathbf{v}_i\mathbf{v}_i^\top$, we have with probability at least $1 - \exp(-\Omega(M^{1/2}))$,

$$\mathrm{tr}(\mathbf{A}\mathbf{\Sigma}) \lesssim \frac{\lambda}{M}\mathrm{tr}(\mathbf{A}) + \mathrm{tr}(\mathbf{A}\mathbf{H}).$$

By Lemma B.8, we have

$$\bar{\mathbf{G}} \le O\big(\mathrm{polylog}(N)\big) \cdot \big[\sigma_\zeta^2\mathbf{H} + \tau^2\eta\,\mathrm{tr}(\mathbf{H}\mathbf{\Sigma}) \cdot (\mathbf{I}_{0:k} + N\eta\mathbf{H}_{k:\infty}) + l^2\tau^2\,\mathrm{tr}(\mathbf{\Sigma}^{-1}\mathbf{H})^2\mathbf{\Sigma}\big].$$

Note that the design of the noise covariance matrix $\mathbf{\Sigma}$ does not affect the non-private error component, we only need to consider the private error component. In particular, we have

$\text{variance}_{\text{p}}$

$$\lesssim \frac{l^2\tau^2}{N} \cdot \text{tr}(\mathbf{H\Sigma}) \cdot \left[\eta^2 \cdot \langle \mathbf{I}_{0:k} + N\eta\mathbf{H}_{k:\infty}, \mathbf{I}_{0:k} + N\eta\mathbf{H}_{k:\infty} \rangle \cdot \left(k + N^2\eta^2 \sum_{i>k} \lambda_i^2\right)\right.$$

$$\left. + \eta \cdot \langle \mathbf{H}_{0:k}^{-1} + N^2\eta^2\mathbf{H}_{k:\infty}, \mathbf{I}_{0:k} + N\eta\mathbf{H}_{k:\infty} \rangle \right]$$

$$+ \frac{l^2\tau^2}{N} \cdot \text{tr}(\mathbf{\Sigma}^{-1}\mathbf{H})^2 \cdot \left[\eta \cdot \langle \mathbf{I}_{0:k} + N\eta\mathbf{H}_{k:\infty}, \mathbf{\Sigma} \rangle \cdot \left(k + N^2\eta^2 \sum_{i>k} \lambda_i^2\right) + \langle \mathbf{H}_{0:k}^{-1} + N^2\eta^2\mathbf{H}_{k:\infty}, \mathbf{\Sigma} \rangle \right]$$

$$\eqsim \frac{l^2\tau^2}{N} \cdot \text{tr}(\mathbf{H\Sigma}) \cdot \underbrace{\left[\eta^2 \cdot \left(k + N^2\eta^2 \sum_{i>k} \lambda_i^2\right)^2 + \eta \cdot \left(\sum_{i\leq k} \lambda_i^{-1} + N^3\eta^3 \sum_{i>k} \lambda_i^2\right)\right]}_{I_1}$$

$$+ \frac{l^2\tau^2}{N} \cdot \text{tr}(\mathbf{\Sigma}^{-1}\mathbf{H})^2 \cdot \underbrace{\left[\eta \cdot \langle \mathbf{I}_{0:k} + N\eta\mathbf{H}_{k:\infty}, \mathbf{\Sigma} \rangle \cdot \left(k + N^2\eta^2 \sum_{i>k} \lambda_i^2\right) + \langle \mathbf{H}_{0:k}^{-1} + N^2\eta^2\mathbf{H}_{k:\infty}, \mathbf{\Sigma} \rangle \right]}_{I_2}.$$

Regarding $I_1$, we can again set $k^* = \max\{k : \lambda_k \geq 1/(N\eta)\}$, then

$$I_1 \leq \eta^2 \cdot \left(k^* + N\eta \sum_{i>k^*} \lambda_i\right)^2 + N\eta^2 \cdot \left(k^* + N\eta \sum_{i>k^*} \lambda_i\right) \lesssim N\eta^2 \left(k^* + N\eta \sum_{i>k^*} \lambda_i\right),$$

where the second inequality is by (B.12). Regarding $I_2$, two quantities demand to be characterized: $\langle \mathbf{I}_{0:k} + N\eta\mathbf{H}_{k:\infty}, \mathbf{\Sigma} \rangle$ and $\langle \mathbf{H}_{0:k}^{-1} + N^2\eta^2\mathbf{H}_{k:\infty}, \mathbf{\Sigma} \rangle$. Then by Lemma B.11, we have with probability at least $1 - \exp(-\Omega(M^{1/2}))$, it holds that

$$\langle \mathbf{I}_{0:k} + N\eta\mathbf{H}_{k:\infty}, \mathbf{\Sigma} \rangle \lesssim \frac{\lambda}{M} \text{tr}(\mathbf{I}_{0:k} + N\eta\mathbf{H}_{k:\infty}) + \langle \mathbf{I}_{0:k} + N\eta\mathbf{H}_{k:\infty}, \mathbf{H} \rangle$$

$$= \frac{\lambda}{M} \cdot \left(k + N\eta \sum_{i>k} \lambda_i\right) + \sum_{i\leq k} \lambda_i + N\eta \sum_{i>k} \lambda_i^2.$$

Besides, we have

$$\langle \mathbf{H}_{0:k}^{-1} + N^2\eta^2\mathbf{H}_{k:\infty}, \mathbf{\Sigma} \rangle \lesssim \frac{\lambda}{M} \text{tr}(\mathbf{H}_{0:k}^{-1} + N^2\eta^2\mathbf{H}_{k:\infty}) + \langle \mathbf{H}_{0:k}^{-1} + N^2\eta^2\mathbf{H}_{k:\infty}, \mathbf{H} \rangle$$

$$= \frac{\lambda}{M} \cdot \left(\sum_{i\leq k} \lambda_i^{-1} + N^2\eta^2 \sum_{i>k} \lambda_i\right) + k + N^2\eta^2 \sum_{i>k} \lambda_i^2.$$

Then, letting $\lambda = \frac{M}{N\eta}$ and setting $k = k^*$, using the fact that $N\eta \sum_{i>k^*} \lambda^2 \leq N\eta\lambda_{k^*+1} \sum_{i>k^*} \lambda_i \geq \sum_{i>k^*} \lambda_i$, we further get that

$$\langle \mathbf{I}_{0:k} + N\eta\mathbf{H}_{k:\infty}, \mathbf{\Sigma} \rangle \lesssim \frac{1}{N\eta} \cdot \left(k + N\eta \sum_{i>k^*} \lambda_i\right) + \text{tr}(\mathbf{H});$$

$$\langle \mathbf{H}_{0:k}^{-1} + N^2\eta^2\mathbf{H}_{k:\infty}, \mathbf{\Sigma} \rangle \lesssim k + N\eta \sum_{i>k^*} .$$

Therefore, we can finally get the following bound for $I_2$,

$$I_2 \lesssim \frac{1}{N} \cdot \left(k^* + N\eta \sum_{i>k^*} \lambda_i\right)^2 + (\eta \, \text{tr}(\mathbf{H}) + 1) \cdot \left(k^* + N\eta \sum_{i>k^*} \lambda_i\right).$$

Applying the fact that $\eta \leq \text{tr}(\mathbf{H})$ and (B.12), we have

$$I_2 \lesssim k^* + N\eta \sum_{i>k^*} \lambda_i.$$

Combining the results for $I_1$ and $I_2$, we can finally get the following upper bound for $\text{variance}_\text{p}$:

$$\text{variance}_\text{p} \lesssim l^2\tau^2\eta^2 \cdot \text{tr}(\mathbf{H\Sigma}) \cdot \left(k^* + N\eta \sum_{i>k^*} \lambda_i\right) + \frac{l^2\tau^2}{N} \cdot \text{tr}(\mathbf{\Sigma}^{-1}\mathbf{H})^2 \cdot \left(k^* + N\eta \sum_{i>k^*} \lambda_i\right).$$

Furthermore, by the assumption that $M \geq N$ and using our choice $\lambda = M/(N\eta) \geq 1/\eta \geq \text{tr}(\mathbf{H})$, we can get the following by Lemma B.10

$$\text{tr}(\mathbf{\Sigma}^2\mathbf{H}) \lesssim \left(\widehat{k} \cdot \frac{\lambda + \text{tr}(\mathbf{H})}{\lambda} + \sum_{i>\widehat{k}} \frac{M\lambda_i}{\lambda}\right) \lesssim \widehat{k} + N\eta \sum_{i>\widehat{k}} \lambda_i \lesssim k^* + N\eta \sum_{i>k^*} \lambda_i,$$

where $\widehat{k} := \min_k\{k : \lambda_{k+1}M \leq \lambda + \sum_{i>k} \lambda_i\}$ and the last inequality holds since $\lambda_i \approx 1/N\eta$ for all $i \in [k^*, \widehat{k}]$ (or $i \in [\widehat{k}, k^*]$). Besides, we also get the following by Lemma B.11: with probability at least $1 - \exp(-\Omega(M^{1/2}))$, it holds that $\text{tr}(\mathbf{H\Sigma}) \lesssim 1/(N\eta)\text{tr}(\mathbf{H}) + \text{tr}(\mathbf{H}^2) \lesssim \text{tr}(\mathbf{H}^2)$. Therefore, we can finally get the following bound for $\text{variance}_\text{p}$:

$$\text{variance}_\text{p} \lesssim l^2\tau^2 \cdot \left[\eta^2 \cdot \text{tr}(\mathbf{H}^2) \cdot \left(k^* + N\eta \sum_{i>k^*} \lambda_i\right) + \frac{1}{N} \cdot \left(k^* + N\eta \sum_{i>k^*} \lambda_i\right)^3\right].$$

Combining with the non-private error results in (B.9), we can finally obtain

$$\text{err}_\text{p} \lesssim \text{err}_\text{np} + \text{variance}_\text{p}$$

$$\lesssim \underbrace{\frac{B^2}{N\eta} + \left(\frac{\sigma_\zeta^2}{N} + \frac{B^2}{N^2\eta}\right) \cdot \left(k^* + N\eta \sum_{i>k^*} \lambda_i\right)}_{\text{Non-private error component}}$$

$$+ \underbrace{l^2\tau^2 \cdot \left[\eta^2 \cdot \text{tr}(\mathbf{H}^2) \cdot \left(k^* + N\eta \sum_{i>k^*} \lambda_i\right) + \frac{1}{N} \cdot \left(k^* + N\eta \sum_{i>k^*} \lambda_i\right)^3\right]}_{\text{Private error component}}.$$

$\square$

### B.3.4 PROOF OF COROLLARY 4.6

*Proof of Corollary 4.6.* If $\lambda_i = i^{-r}$, we can immediately get $k^* = (N\eta)^{\frac{1}{r}}$ and

$$k^* + N\eta \sum_{i>k^*} \lambda_i \approx (N\eta)^{\frac{1}{r}} + N\eta \cdot (N\eta)^{\frac{1-r}{r}} \approx (N\eta)^{\frac{1}{r}}.$$

Therefore, we can get

$$\text{err}_\text{p} \lesssim \left(\frac{\sigma_\zeta^2}{N} + \frac{B^2}{N^2\eta}\right) \cdot (N\eta)^{\frac{1}{r}} + \frac{B^2}{N\eta} + l^2\tau^2 \cdot \left[\eta^2 \cdot \text{tr}(\mathbf{H}^2) \cdot (N\eta)^{\frac{1}{r}} + \frac{1}{N} \cdot (N\eta)^{\frac{1}{r}}\right].$$

Then, assuming $\sigma_\zeta^2$, $B^2$, $l^2$, and $\text{tr}(\mathbf{H}^2)$ are constants, we further obtain

$$\text{err}_\text{p} \lesssim \frac{1}{N\eta} + \frac{(N\eta)^{\frac{1}{r}}}{N} + \tau^2 \cdot \left[\eta^2 \cdot (N\eta)^{\frac{1}{r}} + \frac{1}{N} \cdot (N\eta)^{\frac{3}{r}}\right].$$

Therefore, we can pick

$$\eta \approx \min\left\{N^{-\frac{1}{1+r}}, \tau^{-\frac{2r}{r+3}}N^{-\frac{3}{r+3}}, \tau^{-\frac{2r}{1+3r}}N^{-\frac{1+r}{1+3r}}\right\},$$

and obtain the following bound of $\text{err}_\text{p}$,

$$\text{err}_\text{p} \lesssim N^{-\frac{r}{1+r}} + (\tau^{-2}N)^{-\frac{r}{3+r}} + (\tau^{-1}N)^{-\frac{2r}{1+3r}}$$

$$= \widetilde{\mathcal{O}}\left(N^{-\frac{r}{1+r}}\left[1 + \left(\epsilon^{-2}N^{\frac{2}{1+r}}\right)^{\frac{r}{3+r}} + \left(\epsilon^{-2}N^{\frac{r-1}{1+r}}\right)^{\frac{r}{1+3r}}\right]\right),$$

where we use the fact that $\tau^2 = \widetilde{\mathcal{O}}(\epsilon^{-2})$.

If $\lambda_i = e^{-i}$, then we can get $k^* + N\eta \sum_{i>k^*} \lambda_i \approx \log(N)$, then

$$\mathrm{err_p} \lesssim \frac{1}{N\eta} + \tau^2 \cdot \left(\eta^2 + \frac{1}{N}\right).$$

Picking the stepsize

$$\eta \approx \min\left\{1, (N\tau^2)^{-1/3}, \frac{1}{N^2\tau^2}\right\},$$

applying the fact that $\tau^2 = \widetilde{\mathcal{O}}(\epsilon^{-2})$, we can then get

$$\mathrm{err_p} \lesssim \frac{1}{N} + (\tau^{-2}N^2)^{-1/3} + (N\tau^{-2})^{-1} = \widetilde{\mathcal{O}}\left(N^{-1}\left(1 + (\epsilon^{-2}N)^{\frac{1}{3}}\right)\right).$$

$\square$

### B.4 PROOFS OF THEOREM B.5

In this section, we provide the proof of Theorem B.5.

**Properties of the additive noises $\boldsymbol{\xi}_t$.** Recall that we have

$$\mathbf{w}_{t+1} = \mathbf{w}_t - \eta \cdot [\mathbf{x}_i \mathbf{x}_i^\top \mathbf{w}_t - \mathbf{x}_i y_i + \boldsymbol{\xi}_t] \tag{B.13}$$

where $\boldsymbol{\xi}_t = \boldsymbol{\nu}_t - \boldsymbol{\nu}_{t-1}$, $\boldsymbol{\nu}_{-1} = \mathbf{0}$. According to Algorithm 3, we have that $\boldsymbol{\xi}_t$ is a combination of at most $\lceil \log_2 t + 1 \rceil$ random Gaussian vectors, i.e., $\boldsymbol{\xi}_t = \sum_{k \in \mathcal{K}(t)} \psi(k)\mathbf{z}_k$, where $\mathcal{K}(t)$ is an index set and $\mathcal{K}(t) = \{2^{Q-1}, 2^{Q-1} + 2^{Q-2}, \ldots, 2^{Q-1} + 2^{Q-2} + \cdots + 2^0\}$ when $t = 2^Q$ with $Q \in \mathbb{N}$. When $t \in (2^{Q-1}, 2^Q)$, $\mathcal{K}(t)$ consists of at most $(\lceil \log_2 t \rceil + 1)$ terms staring from $2^{Q-1}$ to $t$. $\psi(k) = \max\{\psi_1, \ldots, \psi_k\}$, and $\mathbf{z}_k \sim N(0, \tau^2 \boldsymbol{\Sigma})$. The reason that we use the indexed $\mathbf{z}_k$ is that $\boldsymbol{\xi}_t$ and $\boldsymbol{\xi}_j$ may have the same $\mathbf{z}_k$, and the index will help us in the analysis. In addition, consider $\boldsymbol{\xi}_t$ and $\boldsymbol{\xi}_j$, where $j < t$ and $j \notin \mathcal{K}(t)$, we have

$$\mathbb{E}[\boldsymbol{\xi}_j \boldsymbol{\xi}_t^\top] = 0, \tag{B.14}$$

where the equality is due to the fact that only $\boldsymbol{\xi}_k$ will depend on $\mathbf{z}_k$, where $k \in \mathcal{K}(t)$.

**Bias-variance decomposition.** According to (B.13) and $y_i = \mathbf{x}_i^\top \mathbf{w}^* + \zeta_i$, we have

$$\mathbf{w}_{t+1} - \mathbf{w}^* = (\mathbf{I} - \eta\mathbf{x}_t\mathbf{x}_t^\top)(\mathbf{w}_t - \mathbf{w}^*) + \eta\mathbf{x}_t\zeta_t - \eta\boldsymbol{\xi}_t.$$

Therefore, we can obtain

$$\mathbb{E}[(\mathbf{w}_{t+1} - \mathbf{w}^*)(\mathbf{w}_{t+1} - \mathbf{w}^*)^\top]$$
$$= \mathbb{E}[(\mathbf{I} - \eta\mathbf{x}_t\mathbf{x}_t^\top)(\mathbf{w}_t - \mathbf{w}^*)(\mathbf{w}_t - \mathbf{w}^*)^\top(\mathbf{I} - \eta\mathbf{x}_t\mathbf{x}_t^\top)] + \eta^2\mathbb{E}[\zeta_t^2\mathbf{x}_t\mathbf{x}_t^\top]$$
$$- \eta\mathbb{E}[(\mathbf{I} - \eta\mathbf{x}_t\mathbf{x}_t^\top)\mathbf{w}_t\boldsymbol{\xi}_t^\top] - \eta\mathbb{E}[\boldsymbol{\xi}_t\mathbf{w}_t^\top(\mathbf{I} - \eta\mathbf{x}_t\mathbf{x}_t^\top)] + \eta^2\mathbb{E}[\boldsymbol{\xi}_t\boldsymbol{\xi}_t^\top],$$

where the first line on R.H.S. is the original bias-variance decomposition, and the second line on R.H.S. is the extra variance error introduced by the random noise generated by the private mechanism. We next consider the term $\mathbb{E}[(\mathbf{I} - \eta\mathbf{x}_t\mathbf{x}_t^\top)\mathbf{w}_t\boldsymbol{\xi}_t^\top]$, we have

$$\mathbf{w}_{t+1} = (\mathbf{I} - \eta\mathbf{x}_t\mathbf{x}_t^\top)\mathbf{w}_t + \eta\mathbf{x}_t y_t + \eta\boldsymbol{\xi}_t = \sum_{j=1}^t (\mathbf{A}_t\mathbf{A}_{t-1}\cdots\mathbf{A}_j(\eta\mathbf{x}_{j-1}y_{j-1} + \eta\boldsymbol{\xi}_{j-1})) + \eta\mathbf{x}_t y_t + \eta\boldsymbol{\xi}_t,$$

where $\mathbf{A}_t = \mathbf{I} - \eta\mathbf{x}_t\mathbf{x}_t^\top$. Therefore, we can obtain

$$\mathbf{w}_t\boldsymbol{\xi}_t^\top = \sum_{j=1}^{t-1} (\mathbf{A}_{t-1}\mathbf{A}_{t-2}\cdots\mathbf{A}_j(\eta\mathbf{x}_{j-1}y_{j-1} + \eta\boldsymbol{\xi}_{j-1}))\boldsymbol{\xi}_t^\top + \eta\mathbf{x}_{t-1}y_{t-1}\boldsymbol{\xi}_t^\top + \eta\boldsymbol{\xi}_{t-1}\boldsymbol{\xi}_t^\top.$$

According to (B.14), $\mathbb{E}\big[\boldsymbol{\xi}_j\boldsymbol{\xi}_t^\top\big] \neq 0$ when $j \in \mathcal{K}(t)$ and $|\mathcal{K}(t)| = \lceil \log_2 t \rceil + 1 = k(t)$. As a result, we have

$$\mathbb{E}\big[(\mathbf{I} - \eta\mathbf{x}_t\mathbf{x}_t^\top)\mathbf{w}_t\boldsymbol{\xi}_t^\top\big] = \mathbb{E}\big[\eta\mathbf{A}_t \sum_{j\in\mathcal{K}(t)} (\mathbf{A}_{t-1}\mathbf{A}_{t-2}\cdots\mathbf{A}_j\boldsymbol{\xi}_{j-1})\boldsymbol{\xi}_t^\top\big] + \mathbb{E}\big[\eta\mathbf{A}_t\boldsymbol{\xi}_{t-1}\boldsymbol{\xi}_t^\top\big]$$
$$+ \mathbb{E}\big[\eta\mathbf{A}_t\mathbf{x}_{t-1}y_{t-1}\boldsymbol{\xi}_t^\top\big].$$

Note that we have

$$\mathbb{E}\big[\eta\mathbf{A}_t\mathbf{x}_{t-1}y_{t-1}\boldsymbol{\xi}_t^\top\big] = \mathbb{E}\big[\eta\mathbf{A}_t\mathbf{x}_{t-1}y_{t-1}\Big(\sum_{k'\in\mathcal{K}(t)}\psi(k')\mathbf{z}_{k'}\Big)^\top\big] = 0,$$

where the last equality is due to the fact that $\{\mathbf{z}_i\}_{i=1}^t$ are zero mean and are independent of other rvs. In addition, for $j \in \mathcal{K}(t)$, we have

$$\boldsymbol{\xi}_{j-1}\boldsymbol{\xi}_t^\top = \Big(\sum_{k\in\mathcal{K}(j-1)}\psi(k)\mathbf{z}_k\Big)\Big(\sum_{k'\in\mathcal{K}(t)}\psi(k')\mathbf{z}_{k'}\Big)^\top,$$

Therefore, we can obtain

$$\mathbb{E}\big[\mathbf{A}_t\mathbf{A}_{t-1}\mathbf{A}_{t-2}\cdots\mathbf{A}_j\boldsymbol{\xi}_{j-1}\boldsymbol{\xi}_t^\top\big] = \mathbb{E}\big[\mathbf{A}_{t:j}\Big(\sum_{k\in\mathcal{K}(j-1)}\psi(k)\mathbf{z}_k\Big)\Big(\sum_{k'\in\mathcal{K}(t)}\psi(k')\mathbf{z}_{k'}\Big)^\top\big]$$
$$= \mathbb{E}\big[\mathbf{A}_{t:j}\Big(\sum_{k_1\in\mathcal{K}(j-1,t)}\psi(k_1)^2\mathbf{z}_{k_1}\mathbf{z}_{k_1}^\top + \sum_{k\neq k\prime}\psi(k)\psi(k')\mathbf{z}_k\mathbf{z}_{k'}^\top\Big)\big]$$
$$= \mathbb{E}\big[\mathbf{A}_{t:j}\Big(\sum_{k_1\in\mathcal{K}(j-1,t)}\psi(k_1)^2\mathbf{z}_{k_1}\mathbf{z}_{k_1}^\top\Big)\big]$$
$$= \tau^2 \sum_{k_1\in\mathcal{K}(j-1,t)}\mathbb{E}\big[\psi(k_1)^2\mathbf{A}_{t:j}\boldsymbol{\Sigma}\big],$$

where $\mathbf{A}_{t:j} = \mathbf{A}_t\cdots\mathbf{A}_j$, $\mathcal{K}(j-1,t) = \mathcal{K}(j-1)\cap\mathcal{K}(t)$, and the third and last lines are due to the fact that $\{\mathbf{z}_i\}_{i=1}^t$ are independent of $\{\psi_i\}_{i=1}^t$ and $\{\mathbf{A}_i\}_{i=1}^t$. Therefore, we can get

$$\mathbb{E}\big[\eta\mathbf{A}_t \sum_{j\in\mathcal{K}(t)} (\mathbf{A}_{t-1}\mathbf{A}_{t-2}\cdots\mathbf{A}_j\boldsymbol{\xi}_{j-1})\boldsymbol{\xi}_t^\top\big] = \sum_{j\in\mathcal{K}(t)}\sum_{k_1\in\mathcal{K}(j-1,t)}\eta\tau^2\mathbb{E}\big[\psi(k_1)^2\mathbf{A}_{t:j}\boldsymbol{\Sigma}\big].$$

Following the same proof, we can obtain that

$$\mathbb{E}\big[\eta\mathbf{A}_t\boldsymbol{\xi}_{t-1}\boldsymbol{\xi}_t^\top\big] = \sum_{k_1\in\mathcal{K}(t-1,t)}\eta\tau^2\mathbb{E}\big[\psi(k_1)^2\mathbf{A}_t\boldsymbol{\Sigma}\big]. \tag{B.15}$$

Thus, we have

$$-\eta\mathbb{E}\big[(\mathbf{I} - \eta\mathbf{x}_t\mathbf{x}_t^\top)\mathbf{w}_t\boldsymbol{\xi}_t^\top\big] - \eta\mathbb{E}\big[\boldsymbol{\xi}_t\mathbf{w}_t^\top(\mathbf{I} - \eta\mathbf{x}_t\mathbf{x}_t^\top)\big]$$
$$= -\eta^2\tau^2 \sum_{j\in\mathcal{K}(t)}\sum_{k_1\in\mathcal{K}(j-1,t)}\Big(\mathbb{E}\big[\psi(k_1)^2\mathbf{A}_{t:j}\boldsymbol{\Sigma}\big] + \mathbb{E}\big[\psi(k_1)^2\boldsymbol{\Sigma}\mathbf{A}_{j:t}\big]\Big)$$
$$\preceq \eta^2\tau^2 k(N) \sum_{j\in\mathcal{K}(t)}\mathbb{E}\big[\mathbf{A}_t\mathbf{A}_{t-1}\cdots\mathbf{A}_j\boldsymbol{\Sigma}\mathbf{A}_j\cdots\mathbf{A}_{t-1}\mathbf{A}_t\big]$$
$$+ \eta^2\tau^2\mathbb{E}\big[\psi(t)^4\big]k(N)\boldsymbol{\Sigma}.$$

As a result, we can obtain

$$\mathbb{E}\big[(\mathbf{w}_{t+1} - \mathbf{w}^*)(\mathbf{w}_{t+1} - \mathbf{w}^*)^\top\big]$$
$$\preceq \mathbb{E}\big[(\mathbf{I} - \eta\mathbf{x}_t\mathbf{x}_t^\top)(\mathbf{w}_t - \mathbf{w}^*)(\mathbf{w}_t - \mathbf{w}^*)^\top(\mathbf{I} - \eta\mathbf{x}_t\mathbf{x}_t^\top)\big] + \eta^2\mathbb{E}\big[\zeta_t^2\mathbf{x}_t\mathbf{x}_t^\top\big]$$
$$+ \eta^2\tau^2 k(N) \sum_{j\in\mathcal{K}(t)}\mathbb{E}\big[\mathbf{A}_t\cdots\mathbf{A}_j\boldsymbol{\Sigma}\mathbf{A}_j\cdots\mathbf{A}_t\big] + \eta^2\tau^2(\mathbb{E}\big[\psi(t)^2 + \psi(t)^4\big])k(N)\boldsymbol{\Sigma},$$
$$\tag{B.16}$$

where the upper bound of $\eta^2\mathbb{E}\big[\boldsymbol{\xi}_t\boldsymbol{\xi}_t^\top\big]$ is due to the similar proof from (B.15).

Now the remaining proof can be conducted based on the following lemma.

**Lemma B.12.** If the stepsize satisfies $\eta \leq 1/(4\beta \operatorname{tr}(\mathbf{H})\log N)$, then let $\mathcal{S}(t)$ be a set of iterate indices that satisfy $j \leq t \ \forall j \in \mathcal{S}(t)$, we have

$$\sum_{j\in\mathcal{S}(t)} \mathbb{E}\big[\mathbf{A}_t \cdots \mathbf{A}_j \mathbf{\Sigma} \mathbf{A}_j \cdots \mathbf{A}_t\big] \preceq \bigg[ \sum_{j\in\mathcal{S}(t)} (\mathcal{I} - \eta\widetilde{\mathcal{T}})^{t-j+1} \bigg] \circ \mathbf{\Sigma}$$
$$+ O(\eta\beta|\mathcal{S}(t)|) \cdot \operatorname{tr}(\mathbf{H}\mathbf{\Sigma}) \cdot \big(\mathbf{I} - (\mathbf{I} - \eta\mathbf{H})^N\big).$$

*Proof of Theorem B.5.* According to Lemma B.12 and use the fact that $|\mathcal{K}(t)| = O(\log_2 N)$, $k(t) \leq k(N) = O(\log_2 N)$, we can obtain the following by (B.16): for all $t \leq N$

$$\mathbb{E}\big[(\mathbf{w}_{t+1} - \mathbf{w}^*)(\mathbf{w}_{t+1} - \mathbf{w}^*)^\top\big]$$
$$\preceq \mathbb{E}\big[\big(\mathbf{I} - \eta\mathbf{x}_t\mathbf{x}_t^\top\big)(\mathbf{w}_t - \mathbf{w}^*)(\mathbf{w}_t - \mathbf{w}^*)^\top\big(\mathbf{I} - \eta\mathbf{x}_t\mathbf{x}_t^\top\big)\big] + \eta^2\mathbb{E}\big[\zeta_t^2\mathbf{x}_t\mathbf{x}_t^\top\big]$$
$$+ \eta^2\tau^2\big(\mathbb{E}\big[\psi(t)^2 + \psi(t)^4\big]\big)k(N)\mathbf{\Sigma}$$
$$+ O\big(\eta^2\tau^2\log_2 N\big) \cdot \bigg\{\bigg[ \sum_{j\in\mathcal{S}(t)} (\mathcal{I} - \eta\widetilde{\mathcal{T}})^{t-j+1} \bigg] \circ \mathbf{\Sigma} + O(\eta\beta\log_2 N) \cdot \operatorname{tr}(\mathbf{H}\mathbf{\Sigma}) \cdot \big(\mathbf{I} - (\mathbf{I} - \eta\mathbf{H})^N\big)\bigg\},$$

where $\mathcal{S}(t)$ is a set of iterate indices satisfying $|\mathcal{S}(t)| = O(\log_2 N)$. Then accordingly, we have the new bias-variance decomposition: considering the bias and variance-covariance matrices $\mathbf{B}_t$ and $\mathbf{C}_t$ that satisfy $\mathbb{E}\big[(\mathbf{w}_t - \mathbf{w}^*)(\mathbf{w}_t - \mathbf{w}^*)^\top\big] = \mathbf{B}_t + \mathbf{C}_t$, we have

$$\mathbf{B}_{t+1} = \mathbb{E}\big[\big(\mathbf{I} - \eta\mathbf{x}_t\mathbf{x}_t^\top\big)\mathbf{B}_t\big(\mathbf{I} - \eta\mathbf{x}_t\mathbf{x}_t^\top\big)\big], \quad \mathbf{B}_0 = (\mathbf{w}_0 - \mathbf{w}^*)(\mathbf{w}_0 - \mathbf{w}^*)^\top,$$

and

$$\mathbf{C}_{t+1} \preceq \mathbb{E}\big[\big(\mathbf{I} - \eta\mathbf{x}_t\mathbf{x}_t^\top\big)\mathbf{C}_t\big(\mathbf{I} - \eta\mathbf{x}_t\mathbf{x}_t^\top\big)\big] + \eta^2\mathbb{E}\big[\zeta_t^2\mathbf{x}_t\mathbf{x}_t^\top\big] + \eta^2\tau^2\big(\mathbb{E}\big[\psi(t)^2 + \psi(t)^4\big]\big)k(N)\mathbf{\Sigma}$$
$$+ \eta^2\mathbf{D}$$
$$\mathbf{C}_0 = \mathbf{0},$$

where

$$\mathbf{D} \preceq O\big(\tau^2\log_2 N\big)\bigg\{\bigg[ \sum_{\tau\in\mathcal{S}(t)} (\mathcal{I} - \eta\widetilde{\mathcal{T}})^{t-\tau+1} \bigg] \circ \mathbf{\Sigma} + O(\eta\beta\log_2 N) \cdot \operatorname{tr}(\mathbf{H}\mathbf{\Sigma}) \cdot \big(\mathbf{I} - (\mathbf{I} - \eta\mathbf{H})^N\big)\bigg\}.$$

This completes the proof. $\qquad\square$

## B.5 PROOFS OF ADDITIONAL LEMMAS

### B.5.1 PROOF OF LEMMA 2.6

*Proof of Lemma 2.6.* By the definition of Rényi divergence, we have

$$D_\alpha\big(N(0, \mathbf{\Sigma})||N(\boldsymbol{\mu}, \mathbf{\Sigma})\big)$$
$$= \frac{1}{\alpha - 1}\log\int_{\mathbb{R}^d} \frac{1}{\sqrt{(2\pi)^k|\mathbf{\Sigma}|}} \cdot \exp\big[-\big(\alpha\mathbf{z}\mathbf{\Sigma}^\dagger\mathbf{z} + (\alpha - 1)(\mathbf{z} - \boldsymbol{\mu})^\top\mathbf{\Sigma}^\dagger(\mathbf{z} - \boldsymbol{\mu})\big)/2\big]$$
$$= \frac{1}{\alpha - 1}\log\int_{\mathbb{R}^d} \frac{1}{\sqrt{(2\pi)^k|\mathbf{\Sigma}|}} \cdot \exp\big[-\big(\mathbf{z}^\top\mathbf{\Sigma}^\dagger\mathbf{z} + 2(1 - \alpha\boldsymbol{\mu}^\top\mathbf{\Sigma}^\dagger\mathbf{z} + (\alpha - 1)\boldsymbol{\mu}^\top\mathbf{\Sigma}^\dagger\boldsymbol{\mu})\big)/2\big]$$
$$= \frac{1}{\alpha - 1}\log\big\{\exp\big[\alpha(\alpha - 1)\boldsymbol{\mu}^\top\mathbf{\Sigma}^\dagger\boldsymbol{\mu}\big]/2\big\}$$
$$= \frac{\alpha}{2} \cdot \|\boldsymbol{\mu}\|_{\mathbf{\Sigma}^\dagger}^2.$$

$$\square$$

### B.5.2 PROOF OF LEMMA B.2

*Proof of Lemma B.2.* For *Bounded norm*, we use the following lemma.

**Lemma B.13.** Suppose $\mathbf{z}$ is a zero mean, sub-Gaussian random vector with sub-Gaussian norm $\sigma_z$. Let $\mathbf{B}$ be an $m \times n$ matrix, then for any $t \geq 0$, we have

$$\mathbb{P}\big(\|\mathbf{B}\mathbf{z}\|_2^2 \geq \sigma_z^2 \|\mathbf{B}\|_F^2 + t\big) \leq \exp\Big(-\frac{ct}{\sigma_z^2 \|\mathbf{B}\|_2^2}\Big),$$

where $c$ is an absolute constant.

According to Lemma B.13, we can choose $\mathbf{B} = \boldsymbol{\Sigma}^{-\frac{1}{2}}\mathbf{H}^{\frac{1}{2}}$ and $t = \sigma_z^2 \|\mathbf{B}\|_2^2 \log(1/\beta)$, we can obtain that with probability at least $1 - c'\beta$

$$\big\|\boldsymbol{\Sigma}^{-\frac{1}{2}}\mathbf{x}\big\|_2^2 \leq 2\sigma_z^2 \operatorname{tr}(\boldsymbol{\Sigma}^{-1}\mathbf{H}) \log(1/\beta),$$

where $c' \geq \beta^{c-1}$.

For *Bounded product*, using Markov inequality, we have

$$\mathbb{P}\big(\langle \mathbf{x}, \mathbf{v}\rangle^2 \geq t^2\big) = \mathbb{P}\big(\langle \mathbf{z}, \mathbf{H}^{\frac{1}{2}}\mathbf{v}\rangle^2 \geq t^2\big)$$

$$= \mathbb{P}\Big(e^{\frac{\langle \mathbf{z}, \widetilde{\mathbf{v}}\rangle^2}{\sigma_z^2 \mathbb{E}\langle \mathbf{z}, \widetilde{\mathbf{v}}\rangle^2}} \geq e^{\frac{t^2}{\sigma_z^2 \mathbb{E}\langle \mathbf{z}, \widetilde{\mathbf{v}}\rangle^2}}\Big)$$

$$\leq e^{-\frac{t^2}{\sigma_z^2 \mathbb{E}\langle \mathbf{z}, \widetilde{\mathbf{v}}\rangle^2}} \cdot \mathbb{E}\Big(e^{\frac{\langle \mathbf{z}, \widetilde{\mathbf{v}}\rangle^2}{\sigma_z^2 \mathbb{E}\langle \mathbf{z}, \widetilde{\mathbf{v}}\rangle^2}}\Big)$$

$$\leq 2e^{-\frac{t^2}{\sigma_z^2 \mathbb{E}\langle \mathbf{z}, \widetilde{\mathbf{v}}\rangle^2}},$$

where $\widetilde{\mathbf{v}} = \mathbf{H}^{\frac{1}{2}}\mathbf{v}$ and the last inequality is due to Assumption 2.2. By choosing $t^2 = \sigma_z^2 \mathbb{E}\langle \mathbf{z}, \widetilde{\mathbf{v}}\rangle^2 \log(1/\beta)$, we can derive the result.

For *Bounded noise*, according to Assumption 2.3, we can directly prove it by the definition of sub-Gaussian random variable with $\gamma_3 = \sigma_\zeta^2$.

$\square$

### B.5.3 PROOF OF LEMMA B.3

*Proof of Lemma B.3.* Therefore, we propose to estimate the upper bound of $\|\mathbf{w}_t - \mathbf{w}^*\|_H^2 + \sigma_\zeta^2$ at each iteration. To this end, we can use: (1) public data to estimate it; or (2) The $(\epsilon_0, \delta_0)$-DP algorithm, i.e., the stability-based histogram (Bun et al., 2016; Karwa & Vadhan, 2017; Liu et al., 2023), on half of the data to estimate it. Here $\epsilon_0 = \epsilon/4\sqrt{T \log(1/\delta_0)}$ and $\delta_0 = \delta/(2T)$ (due to the composition rule, we need to make sure each iteration our method is private, and by composition, we have for $T$ iteration, our method is $(\epsilon, \delta)$-DP). For both of the methods, the key idea is to make use of

For option 1, we have $\{(\widetilde{\mathbf{x}}_i, \widetilde{y}_i)\}_{i=1}^m$ is public dataset. Recall that we have $\{r_i\}_{i=1}^m$ with $r_i = (\widetilde{y}_i - \langle \mathbf{w}_t, \widetilde{\mathbf{x}}_i\rangle)^2$. Thus, we have

$$r_i = \widetilde{\zeta}_i^2 + 2\widetilde{\zeta}_i(\mathbf{w}^* - \mathbf{w}_t)^\top \widetilde{\mathbf{x}}_i + \big(\widetilde{\mathbf{x}}_i^\top (\mathbf{w}^* - \mathbf{w}_t)\big)^2.$$

In addition, by Bernstein's inequality, we have

$$\Big|\frac{1}{m}\sum_{i=1}^m \langle \widetilde{\mathbf{x}}_i, \mathbf{w}^* - \mathbf{w}_t\rangle^2 - \|\mathbf{w}^* - \mathbf{w}_t\|_{\mathbf{H}}^2\Big| \leq C_1 \|\mathbf{w}^* - \mathbf{w}_t\|_{\mathbf{H}}^2,$$

$$\Big|\frac{1}{m}\sum_{i=1}^m \widetilde{\zeta}_i^2 - \sigma_\zeta^2\Big| \leq C_2 \sigma_\zeta^2,$$

$$\Big|\frac{1}{m}\sum_{i=1}^m \widetilde{\zeta}_i \langle \widetilde{\mathbf{x}}_i, \mathbf{w}^* - \mathbf{w}_t\rangle\Big| \leq C_3 \sigma_\zeta \|\mathbf{w}^* - \mathbf{w}_t\|_{\mathbf{H}},$$

where $C_1, C_2, C_3$ are much less than 1 due to large number of $m \geq O(\log N)$. Then, by triangle inequality, we have

$$\Big|\frac{1}{m}\sum_{i=1}^m r_i - \big(\|\mathbf{w}^* - \mathbf{w}_t\|_{\mathbf{H}}^2 + \sigma_\zeta^2\big)\Big| \leq C_4\big(\|\mathbf{w}^* - \mathbf{w}_t\|_{\mathbf{H}}^2 + \sigma_\zeta^2\big), \tag{B.17}$$

where $C_4 \leq 1/2$ given $m \geq O(\log N)$. Thus, if we have public data, we can use $2\sum_{i=1}^{m} r_i/m$ to estimate the upper bound of $\|\mathbf{w}^* - \mathbf{w}_t\|_{\mathbf{H}}^2 + \sigma_\zeta^2$.

For option 2, we have $\{(\widetilde{\mathbf{x}}_i, \widetilde{y}_i)\}_{i=1}^{m}$ is a private dataset. Recall that we split the data $\{r_i\}_{i=1}^{m}$ with $r_i = (\widetilde{y}_i - \langle \mathbf{w}_t, \widetilde{\mathbf{x}}_i \rangle)^2$ into $\widetilde{n}$ subsets of equal size, each with $|\mathcal{S}_j| = \widetilde{m}$ data for $j \in [\widetilde{n}]$. According to (Karwa & Vadhan, 2017), if $\widetilde{\delta} \leq 1/\widetilde{n}$, and we have

$$\widetilde{n} \geq \frac{8}{\widetilde{\epsilon}\beta} \log\left(\frac{4}{\widetilde{\delta}\alpha}\right),$$

we can obtain

$$\mathbb{P}\big(|\widetilde{r}_k - \widehat{r}_k| \geq \beta\big) \leq \alpha.$$

Therefore, if $\widetilde{m}$ is large enough, i.e., $\widetilde{m} \geq O(\log N)$, according to (B.17), we have

$$\big|\bar{r}_k - \big(\|\mathbf{w}^* - \mathbf{w}_t\|_{\mathbf{H}}^2 + \sigma_\zeta^2\big)\big| \leq C_4\big(\|\mathbf{w}^* - \mathbf{w}_t\|_{\mathbf{H}}^2 + \sigma_\zeta^2\big).$$

Thus, if we choose $C_4 \leq 1/4$ (by having large enough $\widetilde{m}$), we can show that all $\bar{r}_k$'s lie in two consecutive bins. According to the private algorithm, we can see that all $\widetilde{r}_k$'s lie in the same two consecutive bins. Let $[l_1, l_2]$ be the nonempty bins containing the most $\widetilde{r}_k$'s. We can ensure that $2l_2 \geq \|\mathbf{w}^* - \mathbf{w}_t\|_{\mathbf{H}}^2 + \sigma_\zeta^2$. Therefore, we can use $2l_2$ to estimate the upper bound of $\|\mathbf{w}^* - \mathbf{w}_t\|_{\mathbf{H}}^2 + \sigma_\zeta^2$. In this case, we require

$$\widetilde{n} \geq \frac{8}{\widetilde{\epsilon}\beta} \log\left(\frac{4}{\widetilde{\delta}\alpha}\right) = \frac{16\sqrt{2N\log(2/\delta)}}{\epsilon\beta} \log\left(\frac{8N}{\delta\alpha}\right) \text{ and } \widetilde{m} \geq O(\log N),$$

where we plugging $\widetilde{\epsilon} = \epsilon/\sqrt{8N\log(2/\delta)}$ and $\widetilde{\delta} = \delta/(2N)$. $\qquad \square$

### B.5.4 PROOF OF LEMMA B.7

We first provide the following useful lemma.

**Lemma B.14.** Under Assumption 2.1, for every $\mathbf{A} \succeq \mathbf{0}$, let $\beta = 16\sigma_z^4$, it holds that

$$(\mathcal{M} - \widetilde{\mathcal{M}}) \circ \mathcal{T}^{-1} \circ \mathbf{A} \preceq \mathcal{M} \circ \mathcal{T}^{-1} \circ \mathbf{A} \preceq \frac{\beta \operatorname{tr}(\mathbf{A})}{1 - \beta\gamma \operatorname{tr}(\mathbf{H})} \cdot \mathbf{H} \preceq \beta \operatorname{tr}(\mathbf{A})\mathbf{H}.$$

*Proof.* It can be shown by the same methods as in (Zou et al., 2021b; Jain et al., 2017b;a). $\qquad \square$

*Proof of Lemma B.7.* Based on the update rule of $\mathbf{C}_t$, we can further obtain

$$\mathbf{C}_N = \eta^2 \sum_{t=0}^{N-1} (\mathcal{I} - \eta\mathcal{T})^t \circ \mathbf{G}_t.$$

Then, it is clear that if $\eta \leq 1/\lambda_1$, we have $\mathcal{I} - \eta\mathcal{T}$ is a PSD mapping so that

$$\mathbf{C}_0 \preceq \mathbf{C}_1 \preceq \cdots \preceq \mathbf{C}_N.$$

Then denote $\mathbf{G}$ as an union upper bound on $\mathbf{G}_0, \ldots, \mathbf{G}_{N-1}$, then due to the fact that $(\mathcal{I} - \eta\mathcal{T})$ is a PSD mapping, we further have

$$\mathbf{C}_N \preceq \eta^2 \sum_{t=0}^{N-1} (\mathcal{I} - \eta\mathcal{T})^t \circ \mathbf{G} = \eta \cdot \mathcal{T}^{-1} \circ (\mathcal{I} - (\mathcal{I} - \eta\mathcal{T})^N) \circ \mathbf{G}.$$

Note that $\widetilde{\mathcal{T}} - \mathcal{T}, \mathcal{I} - \eta\widetilde{\mathcal{T}}$ and $\mathcal{I} - \eta\mathcal{T}$ is a PSD mapping, we have

$$(\mathcal{I} - \eta\widetilde{\mathcal{T}})^N \circ \mathbf{G} \preceq (\mathcal{I} - \eta\widetilde{\mathcal{T}})^{N-1} \circ (\mathcal{I} - \eta\mathcal{T}) \circ \mathbf{G} \preceq \cdots \preceq (\mathcal{I} - \eta\mathcal{T})^N \circ \mathbf{G}.$$

Then using the fact that $\mathcal{T}^{-1}$ is a PSD mapping, we further thave

$$\mathbf{C}_N \preceq \eta \cdot \mathcal{T}^{-1} \circ (\mathcal{I} - (\mathcal{I} - \eta\mathcal{T})^N) \circ \mathbf{G}.$$

Then we can apply Lemma B.14 and obtain

$$(\mathcal{M} - \widetilde{\mathcal{M}}) \circ \mathbf{C}_N \preceq \eta \cdot (\mathcal{M} - \widetilde{\mathcal{M}}) \circ \mathcal{T}^{-1} \circ (\mathcal{I} - (\mathcal{I} - \eta \widetilde{\mathcal{T}})^N) \circ \mathbf{G}$$

$$\preceq \frac{\beta \eta}{1 - \beta \eta \operatorname{tr}(\mathbf{H})} \cdot \operatorname{tr}\left((\mathcal{I} - (\mathcal{I} - \eta \widetilde{T})^N) \circ \mathbf{G}\right) \cdot \mathbf{H}$$

$$\preceq \frac{\beta \eta}{1 - \beta \eta \operatorname{tr}(\mathbf{H})} \cdot \langle \mathbf{I} - (\mathbf{I} - \eta \mathbf{H})^N, \mathbf{G} \rangle \cdot \mathbf{H}.$$

Then, given the above rough bound on $\mathbf{C}_N$, we can further obtain a refined version as follows:

$$\mathbf{C}_{t+1} = (\mathcal{I} - \eta \widetilde{\mathcal{T}}) \circ \mathbf{C}_t + \eta^2 \cdot (\mathcal{M} - \widetilde{\mathcal{M}}) \circ \mathbf{C}_t + \eta^2 \mathbf{G}_t$$

$$\preceq (\mathcal{I} - \eta \widetilde{\mathcal{T}}) \circ \mathbf{C}_t + \eta^2 \cdot (\mathcal{M} - \widetilde{\mathcal{M}}) \circ \mathbf{C}_N + \eta^2 \mathbf{G}.$$

Then, it follows that

$$\mathbf{C}_N \preceq \eta^2 \cdot \sum_{t=0}^{N-1} (\mathcal{I} - \eta \widetilde{\mathcal{T}})^{N-1-t} \circ (\mathcal{M} - \widetilde{\mathcal{M}}) \circ \mathbf{C}_N + \eta^2 \cdot \sum_{t=0}^{N-1} (\mathcal{I} - \eta \widetilde{\mathcal{T}})^{N-1-t} \circ \mathbf{G}$$

$$\preceq \frac{\beta \eta^3}{1 - \beta \eta \operatorname{tr}(\mathbf{H})} \cdot \langle \mathbf{I} - (\mathbf{I} - \eta \mathbf{H})^N, \mathbf{G} \rangle \cdot \sum_{t=0}^{N-1} (\mathcal{I} - \eta \widetilde{\mathcal{T}})^{N-1-t} \circ \mathbf{H} + \eta^2 \cdot \sum_{t=0}^{N-1} (\mathcal{I} - \eta \widetilde{\mathcal{T}})^{N-1-t} \circ \mathbf{G}$$

$$= \frac{\beta \eta^2}{1 - \beta \eta \operatorname{tr}(\mathbf{H})} \cdot \langle \mathbf{I} - (\mathbf{I} - \eta \mathbf{H})^N, \mathbf{G} \rangle \cdot \left(\mathbf{I} - (\mathbf{I} - \eta \mathbf{H})^N\right) + \eta^2 \cdot \sum_{t=0}^{N-1} (\mathcal{I} - \eta \widetilde{\mathcal{T}})^t \circ \mathbf{G}.$$

This completes the proof.

$\square$

### B.5.5  PROOF OF LEMMA B.8

*Proof of Lemma B.8.* By Lemma B.6, we have

$$\operatorname{variance} = \frac{1}{N^2} \sum_{t=0}^{N-1} \sum_{k=t}^{N-1} \langle (\mathbf{I} - \eta \mathbf{H})^{k-t} \mathbf{H}, \mathbf{C}_t \rangle$$

$$= \frac{1}{N^2 \eta} \sum_{t=0}^{N-1} \langle \mathbf{I} - (\mathbf{I} - \eta \mathbf{H})^{N-t}, \mathbf{C}_t \rangle$$

$$\leq \frac{1}{N^2 \eta} \langle \mathbf{I} - (\mathbf{I} - \eta \mathbf{H})^N, \sum_{t=0}^{N-1} \mathbf{C}_t \rangle$$

$$\leq \frac{1}{N \eta} \langle \mathbf{I} - (\mathbf{I} - \eta \mathbf{H})^N, \mathbf{C}_N \rangle,$$

where the first inequality is due to the fact that $\mathbf{C}_t \preceq \mathbf{C}_N$ for all $t \leq N$ (see Lemma B.7). Then applying Lemma B.7, we have

$$\operatorname{variance} \leq \frac{1}{N \eta} \cdot \langle \mathbf{I} - (\mathbf{I} - \eta \mathbf{H})^N, \mathbf{C}_N \rangle$$

$$\leq \frac{2 \beta \eta}{N} \cdot \langle \mathbf{I} - (\mathbf{I} - \eta \mathbf{H})^N, \mathbf{G} \rangle \cdot \langle \mathbf{I} - (\mathbf{I} - \eta \mathbf{H})^N, \mathbf{I} - (\mathbf{I} - \eta \mathbf{H})^N \rangle$$

$$\quad + \frac{\eta}{N} \cdot \langle \mathbf{I} - (\mathbf{I} - \eta \mathbf{H})^N, \sum_{t=0}^{N-1} (\mathcal{I} - \eta \widetilde{\mathcal{T}})^t \circ \mathbf{G} \rangle$$

$$= \frac{2 \beta \eta}{N} \cdot \langle \mathbf{I} - (\mathbf{I} - \eta \mathbf{H})^N, \mathbf{G} \rangle \cdot \langle \mathbf{I} - (\mathbf{I} - \eta \mathbf{H})^N, \mathbf{I} - (\mathbf{I} - \eta \mathbf{H})^N \rangle$$

$$\quad + \frac{\eta}{N} \cdot \langle \sum_{t=0}^{N-1} (\mathcal{I} - \eta \widetilde{\mathcal{T}})^t \circ \left(\mathbf{I} - (\mathbf{I} - \eta \mathbf{H})^N\right), \mathbf{G} \rangle$$

$$\leq \frac{2\beta\eta}{N} \cdot \langle \mathbf{I} - (\mathbf{I} - \eta\mathbf{H})^N, \mathbf{G} \rangle \cdot \langle \mathbf{I} - (\mathbf{I} - \eta\mathbf{H})^N, \mathbf{I} - (\mathbf{I} - \eta\mathbf{H})^N \rangle$$
$$+ \frac{1}{N} \cdot \langle (\mathbf{I} - (\mathbf{I} - \eta\mathbf{H})^N)^2 \mathbf{H}^{-1}, \mathbf{G} \rangle. \tag{B.18}$$

Further by Theorem B.5, we can get for any $t \leq N$,

$$\mathbf{G}_t \preceq \sigma_\zeta^2 \mathbf{H} + O\big(\mathbb{E}\big[\psi(t)^2 + \psi(t)^4\big]\tau^2 \log_2 N\big)\mathbf{\Sigma}$$
$$+ O\big(\tau^2 \log_2 N\big)\bigg\{ \bigg[ \sum_{\tau \in \bar{\mathcal{K}}(t)} (\mathcal{I} - \eta\widetilde{\mathcal{T}})^{t-\tau+1} \bigg] \circ \mathbf{\Sigma} + O(\eta\beta \log_2 N) \cdot \text{tr}(\mathbf{H}\mathbf{\Sigma}) \cdot \big(\mathbf{I} - (\mathbf{I} - \eta\mathbf{H})^N\big) \bigg\}.$$

Noting that $\psi(t) \leq \psi(N)$ for all $t \leq N$ and $(\mathcal{I} - \eta\widetilde{\mathcal{T}}) \circ \mathbf{A} \preceq \mathbf{A}$ for any PSD matrix $\mathbf{A}$ that commutes with $\mathbf{H}$. Therefore, we can further get that

$$\langle \mathbf{A}, \mathbf{G} \rangle \lesssim \sigma_\zeta^2 \cdot \langle \mathbf{A}, \mathbf{H} \rangle + \tau^2\beta \cdot (\log_2 N)^2 \eta \cdot \text{tr}(\mathbf{H}\mathbf{\Sigma}) \cdot \langle \mathbf{A}, (\mathbf{I} - (\mathbf{I} - \eta\mathbf{H})^N) \rangle$$
$$+ \big(\mathbb{E}\big[\psi(N)^2 + \psi(N)^4\big] + 1\big) \cdot \tau^2 \cdot \log_2 N \cdot \langle \mathbf{A}, \mathbf{\Sigma} \rangle,$$

where $\mathbf{A}$ can be any PSD matrix that commutes with $\mathbf{H}$. Moreover, noting that for any $x \in (0, 1/\eta)$, we have $1 - (1 - \eta x)^N \leq \min\{1, N\eta x\}$, then

$$\mathbf{I} - (\mathbf{I} - \eta\mathbf{H})^N \preceq \mathbf{I}_{0:k} + N\eta\mathbf{H}_{k:\infty}$$

for any $k \geq 0$. Therefore, we can define a new matrix as follows:

$$\bar{\mathbf{G}} = \sigma_\zeta^2 \cdot \mathbf{H} + \tau^2\beta \cdot (\log_2 N)^2 \eta \, \text{tr}(\mathbf{H}\mathbf{\Sigma}) \cdot \big(\mathbf{I}_{0:k} + N\eta\mathbf{H}_{k:\infty}\big)$$
$$+ \big(\mathbb{E}\big[\psi(N)^2 + \psi(N)^4\big] + 1\big) \cdot \tau^2 \cdot \log_2 N \cdot \mathbf{\Sigma}.$$

Therefore, putting the above results into (B.18), we obtain

$$\text{variance} \lesssim \frac{\beta\eta}{N} \cdot \langle \mathbf{I}_{0:k} + N\eta\mathbf{H}_{k:\infty}, \bar{\mathbf{G}} \rangle \cdot \big(k + N^2\eta^2 \sum_{i>k} \lambda_i^2\big) + \frac{1}{N} \cdot \langle \mathbf{H}_{0:k}^{-1} + N^2\eta^2\mathbf{H}_{k:\infty}, \bar{\mathbf{G}} \rangle.$$

This completes the proof. $\qquad\square$

### B.5.6 PROOF OF LEMMA B.12

The following lemma is useful in the subsequent proof.

**Lemma B.15** (Lemma C.4 in Wu et al. (Wu et al., 2022)). *For a sequence of PSD matrices $\{\mathbf{M}_t\}_{t=0\ldots N}$ that satisfy $\mathbf{M}_t = (\mathcal{I} - \eta\mathcal{T}) \circ \mathbf{M}_{t-1}$ and $\mathbf{M}_0 = \mathbf{\Sigma}$, then if $\eta \leq 1/(4\beta\,\text{tr}(\mathbf{H})\log N)$, it holds that for any $t \in [0, N]$*

$$\text{tr}(\mathbf{H}\mathbf{M}_t) \leq 4 \cdot \left\langle \frac{1}{\eta N}\mathbf{I}_{0:k} + \mathbf{H}_{k:\infty}, \mathbf{\Sigma} \right\rangle,$$

*where $k \in [N]$ can be arbitrarily chosen.*

*Proof of Lemma B.12.* Note that $\mathbf{\Sigma}$ is independent of $\mathbf{A}_t, \ldots, \mathbf{A}_j$, therefore when taking the expectation of over $\mathbf{A}_t, \ldots, \mathbf{A}_j$, then can be treated equally. In particular, for any $t$ and $j$, we have

$$\mathbb{E}\big[\mathbf{A}_t \cdots \mathbf{A}_j \mathbf{\Sigma} \mathbf{A}_j \cdots \mathbf{A}_t\big] = (\mathcal{I} - \eta\mathcal{T})^{t-j+1} \circ \mathbf{\Sigma}.$$

Then define $\mathbf{M}_s = (\mathcal{I} - \eta\mathcal{T})^s \circ \mathbf{\Sigma}$ and $\mathbf{M}_0 = \mathbf{\Sigma}$, we can get the following:

$$\mathbf{M}_{s+1} = (\mathcal{I} - \eta\mathcal{T}) \circ \mathbf{M}_s \preceq (\mathcal{I} - \eta\widetilde{\mathcal{T}}) \circ \mathbf{M}_s + \beta\eta^2 \cdot \text{tr}(\mathbf{H}\mathbf{M}_s) \cdot \mathbf{H}. \tag{B.19}$$

Then by Lemma B.15 and set $k = 0$, we can immediately obtain that

$$\text{tr}(\mathbf{H}\mathbf{M}_s) \leq 4 \cdot \text{tr}(\mathbf{H}\mathbf{\Sigma}).$$

Plugging the above inequality into (B.19), we have

$$\mathbf{M}_s \preceq (\mathcal{I} - \eta\widetilde{\mathcal{T}}) \circ \mathbf{M}_{s-1} + 4\beta\eta^2 \cdot \text{tr}(\mathbf{H}\mathbf{\Sigma}) \cdot \mathbf{H}$$

$$= (\mathcal{I} - \eta\widetilde{\mathcal{T}})^s \circ \mathbf{\Sigma} + 4\beta\eta^2 \cdot \mathrm{tr}(\mathbf{H}\mathbf{\Sigma}) \cdot \sum_{r=0}^{s-1} (\mathcal{I} - \eta\widetilde{T})^r \circ \mathbf{H}$$

$$\preceq (\mathcal{I} - \eta\widetilde{\mathcal{T}})^s \circ \mathbf{\Sigma} + 4\beta\eta^2 \cdot \mathrm{tr}(\mathbf{H}\mathbf{\Sigma}) \cdot \sum_{r=0}^{N-1} (\mathcal{I} - \eta\widetilde{T})^r \circ \mathbf{H}$$

$$\preceq (\mathcal{I} - \eta\widetilde{\mathcal{T}})^s \circ \mathbf{\Sigma} + 4\beta\eta \cdot \mathrm{tr}(\mathbf{H}\mathbf{\Sigma}) \cdot \left(\mathbf{I} - (\mathbf{I} - \eta\mathbf{H})^N\right)$$

Then, it suffices to prove the upper bound for $\sum_{j\in\mathcal{S}(t)} \mathbf{M}_{t-j+1}$. In particular, applying the above result leads to

$$\sum_{j\in\mathcal{S}(t)} \mathbf{M}_{t-j+1} \preceq \sum_{j\in\mathcal{S}(t)} (\mathcal{I} - \eta\widetilde{\mathcal{T}})^{t-j+1} \circ \mathbf{\Sigma} + 4\beta\eta \cdot \mathrm{tr}(\mathbf{H}\mathbf{\Sigma}) \cdot \left(\mathbf{I} - (\mathbf{I} - \eta\mathbf{H})^N\right) \cdot |\mathcal{S}(t)|$$

$$\preceq \left[\sum_{j\in\mathcal{S}(t)} (\mathcal{I} - \eta\widetilde{\mathcal{T}})^{t-j+1}\right] \circ \mathbf{\Sigma} + O(\eta\beta|\mathcal{S}(t)|) \cdot \mathrm{tr}(\mathbf{H}\mathbf{\Sigma}) \cdot \left(\mathbf{I} - (\mathbf{I} - \eta\mathbf{H})^N\right).$$

This completes the proof.

$\square$

### B.5.7 PROOF OF LEMMA B.10

*Proof of Lemma B.10.* Note that $\mathbf{\Sigma} = M^{-1}(\lambda\mathbf{I} + \widetilde{\mathbf{X}}\widetilde{\mathbf{X}}^\top)$, by the Woodbury identity (Golub & Van Loan, 2013), we have

$$(\lambda\mathbf{I} + \widetilde{\mathbf{X}}\widetilde{\mathbf{X}}^\top)^\dagger = \lambda^{-1}\mathbf{I} - \lambda^{-2}\widetilde{\mathbf{X}}(\mathbf{I} + \lambda^{-1}\widetilde{\mathbf{X}}^\top\widetilde{\mathbf{X}})^{-1}\widetilde{\mathbf{X}}^\top$$

Therefore, we can further obtain

$$\mathrm{tr}(\mathbf{\Sigma}^\dagger\mathbf{H}) = M \cdot \mathrm{tr}\left(\lambda^{-1}\mathbf{H} - \lambda^{-2}\mathbf{H}\widetilde{\mathbf{X}}(\mathbf{I} + \lambda^{-1}\widetilde{\mathbf{X}}^\top\widetilde{\mathbf{X}})^{-1}\widetilde{\mathbf{X}}^\top\right)$$
$$= M \cdot \lambda^{-1} \mathrm{tr}(\mathbf{H}) - M\lambda^{-2} \mathrm{tr}\left(\widetilde{\mathbf{X}}^\top\mathbf{H}\widetilde{\mathbf{X}}(\mathbf{I} + \lambda^{-1}\widetilde{\mathbf{X}}^\top\widetilde{\mathbf{X}})^{-1}\right). \quad \text{(B.20)}$$

Then by Assumption 2.2, let $\mathbf{H} = \sum_i \lambda_i\mathbf{v}_i\mathbf{v}_i^\top$, where $\lambda_i$ and $\mathbf{v}_i$ denote the $i$-th largest eigenvalue of $\mathbf{H}$ and its corresponding eigenvector, we can further define $\mathbf{z}_i = \widetilde{\mathbf{X}}^\top\mathbf{v}_i/\lambda_i^{1/2} \in \mathbb{R}^M$. Then $\mathbf{z}_i$'s will be independent $\sigma_z$-subGaussian random vectors and the following identities hold:

$$\widetilde{\mathbf{X}}^\top\widetilde{\mathbf{X}} = \sum_i \lambda_i\mathbf{z}_i\mathbf{z}_i^\top, \quad \widetilde{\mathbf{X}}^\top\mathbf{H}\widetilde{\mathbf{X}} = \sum_i \lambda_i^2\mathbf{z}_i\mathbf{z}_i^\top.$$

Then, we can further obtain that

$$\mathrm{tr}\left(\widetilde{\mathbf{X}}^\top\mathbf{H}\widetilde{\mathbf{X}}(\mathbf{I} + \lambda^{-1}\widetilde{\mathbf{X}}^\top\widetilde{\mathbf{X}})^{-1}\right) = \mathrm{tr}\left(\left(\sum_i \lambda_i^2\mathbf{z}_i\mathbf{z}_i^\top\right)\left(\mathbf{I} + \lambda^{-1}\sum_i \lambda_i\mathbf{z}_i\mathbf{z}_i^\top\right)^{-1}\right)$$
$$= \sum_i \lambda_i^2 \cdot \mathbf{z}_i^\top\left(\mathbf{I} + \lambda^{-1}\sum_i \lambda_i\mathbf{z}_i\mathbf{z}_i^\top\right)^{-1}\mathbf{z}_i.$$

By This further implies that

$$\mathrm{tr}(\mathbf{\Sigma}^\dagger\mathbf{H}) = M \cdot \left(\lambda^{-1}\sum_i \lambda_i - \lambda^{-2}\sum_i \lambda_i^2 \cdot \mathbf{z}_i^\top\left(\mathbf{I} + \lambda^{-1}\sum_i \lambda_i\mathbf{z}_i\mathbf{z}_i^\top\right)^{-1}\mathbf{z}_i\right)$$
$$= M\lambda^{-1} \cdot \sum_i \left\{\lambda_i - \lambda^{-1}\lambda_i^2 \cdot \mathbf{z}_i^\top\left(\mathbf{I} + \lambda^{-1}\sum_i \lambda_i\mathbf{z}_i\mathbf{z}_i^\top\right)^{-1}\mathbf{z}_i\right\}. \quad \text{(B.21)}$$

By Sherman-Morrison formula, denote $\mathbf{A} = \mathbf{I} + \lambda^{-1}\sum_i \lambda_i\mathbf{z}_i\mathbf{z}_i^\top$ and $\mathbf{A}_{-i} = \mathbf{I} + \lambda^{-1}\sum_{j\neq i} \lambda_j\mathbf{z}_j\mathbf{z}_j^\top$, we have

$$\mathbf{z}_i^\top\left(\mathbf{I} + \lambda^{-1}\sum_i \lambda_i\mathbf{z}_i\mathbf{z}_i^\top\right)^{-1}\mathbf{z}_i = \mathbf{z}_i^\top\left(\mathbf{A}_{-i}^{-1} - \frac{\lambda^{-1}\lambda_i\mathbf{A}_{-i}^{-1}\mathbf{z}_i\mathbf{z}_i^\top\mathbf{A}_{-i}^{-1}}{1 + \lambda^{-1}\lambda_i\mathbf{z}_i^\top\mathbf{A}_{-i}^{-1}\mathbf{z}_i}\right)\mathbf{z}_i$$

$$= \mathbf{z}_i^\top \mathbf{A}_{-i}^{-1} \mathbf{z}_i - \frac{\lambda^{-1} \lambda_i \left( \mathbf{z}_i^\top \mathbf{A}_{-i}^{-1} \mathbf{z}_i \right)^2}{1 + \lambda^{-1} \lambda_i \mathbf{z}_i^\top \mathbf{A}_{-i}^{-1} \mathbf{z}_i}$$

$$= \frac{\mathbf{z}_i^\top \mathbf{A}_{-i}^{-1} \mathbf{z}_i}{1 + \lambda^{-1} \lambda_i \mathbf{z}_i^\top \mathbf{A}_{-i}^{-1} \mathbf{z}_i}.$$

Moreover, we have

$$\lambda_i - \lambda^{-1} \lambda_i^2 \cdot \mathbf{z}_i^\top \left( \mathbf{I} + \lambda^{-1} \sum_i \lambda_i \mathbf{z}_i \mathbf{z}_i^\top \right)^{-1} \mathbf{z}_i = \lambda_i - \frac{\lambda^{-1} \lambda_i^2 \mathbf{z}_i^\top \mathbf{A}_{-i}^{-1} \mathbf{z}_i}{1 + \lambda^{-1} \lambda_i \mathbf{z}_i^\top \mathbf{A}_{-i}^{-1} \mathbf{z}_i}$$

$$= \frac{\lambda_i}{1 + \lambda^{-1} \lambda_i \mathbf{z}_i^\top \mathbf{A}_{-i}^{-1} \mathbf{z}_i}.$$

Putting the above results into (B.21) leads to

$$\mathrm{tr}(\mathbf{\Sigma}^\dagger \mathbf{H}) = M \lambda^{-1} \cdot \sum_i \frac{\lambda_i}{1 + \lambda^{-1} \lambda_i \mathbf{z}_i^\top \mathbf{A}_{-i}^{-1} \mathbf{z}_i} = M \cdot \sum_i \frac{\lambda_i}{\lambda + \lambda_i \mathbf{z}_i^\top \mathbf{A}_{-i}^{-1} \mathbf{z}_i}.$$

We then provide the following lemma that is adapted from the proof of Lemma 7 in Tsigler & Bartlett (2020).

**Lemma B.16.** Under Assumption 2.2, denote $\widehat{k} := \min_k \{ k : \lambda_{k+1} M \leq \lambda + \sum_{i>k} \lambda_i \}$, then with probability at least $1 - \exp(-\Omega(n))$, it holds that

$$\mathbf{z}_i^\top \mathbf{A}_{-i}^{-1} \mathbf{z}_i \geq C \cdot \frac{\lambda M}{\lambda + \sum_{j > \widehat{k}} \lambda_j},$$

where $C$ is an absolute positive constant.

Based on the above lemma, we can obtain

$$\mathrm{tr}(\mathbf{\Sigma}^\dagger \mathbf{H}) = M \cdot \sum_i \frac{\lambda_i}{\lambda + \lambda_i \mathbf{z}_i^\top \mathbf{A}_{-i}^{-1} \mathbf{z}_i} \leq M \cdot \sum_i \frac{\lambda_i}{\lambda + C \cdot \frac{M \lambda_i \lambda}{\lambda + \sum_{j > \widehat{k}} \lambda_j}}.$$

Note that $\lambda_i M \geq \lambda + \sum_{j > \widehat{k}} \lambda_j$ for all $i \leq \widehat{k}$, then we have

$$\mathrm{tr}(\mathbf{\Sigma}^\dagger \mathbf{H}) \leq C' \cdot M \cdot \left( \sum_{i \leq \widehat{k}} \frac{\lambda + \sum_{j > \widehat{k}} \lambda_j}{M \lambda} + \sum_{i > \widehat{k}} \frac{\lambda_i}{\lambda} \right) = C' \cdot \left( \widehat{k} \cdot \frac{\lambda + \sum_{j > \widehat{k}} \lambda_j}{\lambda} + \sum_{i > \widehat{k}} \frac{M \lambda_i}{\lambda} \right).$$

This completes the proof. $\qquad \square$

### B.5.8 PROOF OF LEMMA B.11

*Proof of Lemma B.11.* Recall that $\mathbf{\Sigma} = M^{-1}(\lambda \mathbf{I} + \widetilde{\mathbf{X}} \widetilde{\mathbf{X}}^\top)$, we have

$$\mathrm{tr}(\mathbf{A} \mathbf{\Sigma}) = \frac{\lambda}{M} \mathrm{tr}(\mathbf{A}) + \frac{1}{M} \mathrm{tr}(\widetilde{\mathbf{X}}^\top \mathbf{A} \widetilde{\mathbf{X}}).$$

Similar to the proof of Lemma B.10, defining $\mathbf{z}_i = \widetilde{\mathbf{X}} \mathbf{v}_i / \lambda_i^{1/2}$, we can get

$$\mathrm{tr}(\widetilde{\mathbf{X}}^\top \mathbf{A} \widetilde{\mathbf{X}}) = \mathrm{tr}\left( \sum_i \lambda_i \mu_i \mathbf{z}_i \mathbf{z}_i^\top \right) = \sum_i \lambda_i \mu_i \cdot \| \mathbf{z}_i \|_2^2. \tag{B.22}$$

Then noting that $\mathbf{z}_i$ is a sub-Gaussian random vector with identity covariance matrix, we can further get that with probability at $1 - \exp(-\Omega(M^{1/2}))$,

$$\mathrm{tr}(\widetilde{\mathbf{X}}^\top \mathbf{A} \widetilde{\mathbf{X}}) \leq C \cdot M \sum_i \lambda_i \mu_i = C \cdot M \, \mathrm{tr}(\mathbf{A} \mathbf{H}), \tag{B.23}$$

for some absolute constant $C > 1$. Putting (B.23) into (B.22), we obtain

$$\mathrm{tr}(\mathbf{A} \mathbf{\Sigma}) \leq C \cdot \left( \frac{\lambda}{M} \mathrm{tr}(\mathbf{A}) + \mathrm{tr}(\mathbf{A} \mathbf{H}) \right).$$

This completes the proof.

$\qquad \square$

