# OpenReview forum: "Private Overparameterized Linear Regression without Suffering in High Dimensions"
_ICLR.cc/2024/Conference — Submitted to ICLR 2024_

### Official Review · Reviewer_2YVT · 2023-10-31

**Soundness:** 4 excellent
**Presentation:** 4 excellent
**Contribution:** 3 good
**Rating:** 6
**Confidence:** 3

**Summary:**

The paper proposes a certain data-adaptive DP training method for the linear regression. It improves upon the recent utility bounds by Varshney et al. (COLT 2022) by getting rid of the explicit dependency on the feature space dimension. This is obtained by utilizing the covariance matrix $H$ of the features in several ways: by adding non-isotropic noise with a covariance that is of the form $\lambda I + H$ where $ \lambda > 0$ and by using an adaptive clipping that also takes into account $H$ and the magnitudes of the residuals $x^T \theta_k - y$, where $\theta_k$ denotes the model parameters at iteration $k$. Also, instead of using the shuffling mechanism as in (Varshney et al., 2022), the paper uses the DP-FTRL tree aggregation mechanism. By using the non-isotropic noise with the data-covariance matrix, the method can be thought of as a certain second-order version of DP-SGD (or DP-FTRL) for the linear regression. The covariance matrix $H$ is assumed to be obtained from public data, i.e., no private algorithm for its estimation is given, and the dimension-independency is obtained using an analysis where the trace of $H$ (i.e. the sum of the singular values) appears (by using a norm weighted with $H$ in the analysis, as far as I see). The analysis gives the dimension-independency even in case $H$ is not used for the noise addition and clipping .

**Strengths:**

- The paper is really well written, and seems to address an important problem: based on my knowledge and looking at the references, this way of taking into account the spectrum of the feature covariances has not been addressed before in this detail and accuracy. Even if the analysis is limited to the linear regression, I get the impression this is an important contribution.

- Generality: the results could be used for various models of data. It seems that the techniques pave the way for follow up work, where one could try to get rid of the need for public data, for example.

**Weaknesses:**

- The assumption that we know the covariance of the features $H$ or that we can somehow get an approximation from the public data is quite strong. I think it would make the paper much stronger if there was some version that makes a DP approximation of $H$.

- I think one argument against the baseline method of Varshney et al. (2022) is weak: namely that you can get rid of the restriction on $\varepsilon$ by using DP-FTRL instead of the shuffling approach. This argument is mentioned few times. As far as I see, the condition on $\varepsilon$ for the shuffling approach of Varshney et al. comes from the analysis of Feldman et al. (2022) for shufflers, where the $(\varepsilon,\delta)$-bounds are obtained by bounding the $(\varepsilon,\delta)$-distance between certain binomial distributions. And the condition on $\varepsilon$ then comes from the analysis that they use to obtain convenient analytical bounds. I believe one could have for the shuffler $(\varepsilon,\delta)$-bounds with arbitrarily small $\varepsilon$'s, just the way you have DP bounds for DP-FTRL. My point is that this restriction is not inherent limitation of the shuffler approach, just of the particular analytical bounds, and it would be possible to get bounds with small $\varepsilon$'s.

**Questions:**

Do you think the current techniques could be used in case where you release $H$ privately?

Small remarks:

- There are typos in summation indices in several places, e.g. p. 3:
$$
\sum_{k_1 < k \leq k_2} \mu_i v_i v_i^T
$$
and
$$
\sum_{j=0}^T g_i
$$
in several places at least on p. 5.

- In the condition on the fourth-order momentum on p.4, shouldn't there be $ \preccurlyeq $ (or something similar) instead of $ \leq $ ?

- Beginning of P. 6: the condition $\varepsilon \leq \sqrt{N}$ must be a typo. It does not seem restrictive nor to correspond to results of Feldman et al.

- Check the bibliography. Example: you give the Arxiv versions of Varshney et al. (2022) and Karwa and Vadhan (2018) as references. They have appeared in COLT and ITCS, respectively.

---

> ### Author Response · Authors · 2023-11-19
>
> We thank the reviewer for their detailed comments, and we address the reviewer’s concerns as follows.
>
> >The assumption that we know the covariance of the features $\mathbf{H}$ or that we can somehow get an approximation from the public data is quite strong. I think it would make the paper much stronger if there was some version that makes a DP approximation of $\mathbf{H}$.
>
> Note that we do not assume that our algorithm knows the covariance of the features $\mathbf{H}$.
> In addition, if we do not have a public dataset, we can use the isotropic noise $\mathbf{\Sigma}=\mathbf{I}$ in our method to still get excess risk bounds (see Corollary 4.3, Corollary 4.4) that depend on the eigenspectrum of the data covariance instead of the problem dimension as those in Varshney et al. (2022), Liu et al. (2023). The idea of using public data in the DP methods has been widely used in the DP literature, e.g., Kairouz et al. (2020), Yu et al, (2022), Zhou et al. (2021). Moreover, getting unlabeled public data is generally cheaper than labeled training data. It is a good idea to develop a DP approximate $\mathbf{H}$, and we leave it as our future work.
>
> >Do you think the current techniques could be used in case where you release H privately?
>
> Yes, because our developed analytical framework can handle the additive noise with general covariance matrices. Therefore, we can directly plug the DP approximation of $\mathbf{H}$ into our framework to get the result.
>
> >you can get rid of the restriction on $\epsilon$, this restriction is not inherent limitation of the shuffler approach
>
> Firstly, we would like to clarify the restriction of $\epsilon$ in Varshney et al. (2022). On page 6, it should be $\epsilon\leq 1/\sqrt{N}$ instead of $\epsilon\leq \sqrt{N}$. Therefore, the  $\epsilon$ in Varshney et al. (2022) is very restrictive since they can only choose a very small $\epsilon$. Secondly, we believe that this restriction is essential when applying the shuffling result (Feldman et al. (2022)). The reason that they have this restriction is as follows. In their proofs, they first show that each step in their method is $(\epsilon_0,\delta_0)$-DP. Then they use the shuffling amplification result to show that their method is $(\epsilon,\delta)$-DP with $\epsilon=\epsilon_0/\sqrt{N}$. Note that this shuffling amplification result requires $\epsilon_0\leq 1$, and thus they have the restriction $\epsilon\leq 1/\sqrt{N}$. The good thing about this shuffling amplification result is that one only needs to add random noise with the variance at the order of $1/\epsilon_0^2=1/(\epsilon^2 N)$ at each step to achieve $(\epsilon,\delta)$-DP. As a result, one can significantly improve the utility guarantee due to the $1/N$ factor in variance for the random noise. In order to get rid of this restriction, one needs to choose a large $\epsilon_0$. However, according to Feldman et al. (2022) (comments under Theorem 3.1 and also in Appendix B), when $\epsilon_0$ is larger than the order of $\log N$, there is no amplification effect, resulting in $\epsilon=\epsilon_0$. As a result, one needs to add random noise with the variance at the order of $1/\epsilon_0^2=1/\epsilon^2$ at each iteration to achieve $(\epsilon,\delta)$-DP. This leads to a substantially weaker utility guarantee, as the beneficial $1/N$ factor in the noise’s variance no longer applies. Therefore, we believe this restriction is essential in the shuffling result to achieve a strong utility guarantee since it does not allow local privacy parameter $\epsilon_0$ to be large.
>
> >Typos and bibliography issue
>
> Thanks for pointing out those typos and the bibliography issue, we will fix them in the revision.

---

> ### Comment · Reviewer_2YVT · 2023-11-22
>
> Thank you for the clarification on the shuffling result. (By the way, I think you mean Feldman et al. (2020) instead of Feldman et al. (2022) here). I had a look at their Appendix B you mentioned and that indeed seems to be the case, up to log factors. I think you are neglecting here the log factors which in this case can be quite considerable, compared e.g. to $\sqrt{n}$ (those log factors may include $e^{\epsilon_0}$). Still, I would reformulate that part, and emphasise that amplification vs. non-amplification since I think you could take larger epsilons for it, as it reads currently you get the impression there is some hard limitation (ultimately it is just the analytical approximations limitation).
>
> Otherwise the reply answers my questions, I don't have further questions.

---

> ### Author Response · Authors · 2023-11-23
>
> Thank you for your response. Yes, we indeed mean Feldman et al. (2020).  You are correct that the condition is $\log(1/\delta)\geq Ne^{-\epsilon_0}$, and we state it as $\epsilon_0$ is larger than the order of $\log N$ in our previous response for simplicity. We will emphasize the results using amplification vs. non-amplification under different regimes of $\epsilon_0$ in the revision to make it more clear. Thank you for your suggestions.

---

### Official Review · Reviewer_BWiw · 2023-11-01

**Soundness:** 4 excellent
**Presentation:** 3 good
**Contribution:** 2 fair
**Rating:** 3
**Confidence:** 3

**Summary:**

This paper analyzes a variant of DP-FTRL for linear regression in the overparameterized regime. Here the goal is to produce a parameter that predicts well on new data. There are many existing works on DP linear regression but all existing analyses have sample complexities that grow with the dimension. Investigation into the behavior of nonprivate algorithms in the overparameterized setting is relatively recent as well, and this paper builds heavily on recent work of Zou et al., who analyze one-pass SGD in a nearly identical setting.

This work makes two modifications to standard DP-FTRL. The first is adaptive clipping, using a subroutine of Liu et al. The second modification allows the use of non-isotropic noise, with the noise covariance set via some prior knowledge. Ideally, the noise covariance would depend on the covariance of the $x$'s. Privately estimating this requires $N=\Omega(d^{3/2})$ examples, even for Gaussian distributions [see 1, 2, 3; these aren't cited but should be]. This is outside the regime of interest. However, one might have access to unlabeled public data points, which would yield useful prior knowledge.

The main result, a theorem about the algorithm's accuracy, is rather inscrutable. We get two main corollaries: one with isotropic noise and one with noise based on this large public data set. The key feature about these bounds is that they are expressed in terms of the spectrum of the covariance of the $x$'s and do not directly depend on the dimension. We get further corollaries simplifying the results when the eigenvalues of the covariance matrix decay sufficiently quickly.

[1] Dwork, Cynthia, et al. "Analyze gauss: optimal bounds for privacy-preserving principal component analysis." Proceedings of the forty-sixth annual ACM symposium on Theory of computing. 2014.

[2] Kamath, Gautam, Argyris Mouzakis, and Vikrant Singhal. "New lower bounds for private estimation and a generalized fingerprinting lemma." Advances in Neural Information Processing Systems 35 (2022): 24405-24418.

[3] Narayanan, Shyam. "Better and Simpler Lower Bounds for Differentially Private Statistical Estimation." arXiv preprint arXiv:2310.06289 (2023).

**Strengths:**

Linear regression is a fundamental statistical task. It is interesting that we can achieve nontrivial guarantees in the overparameterized setting. The results are original.

The paper is structured well, making it easy to understand how the results fit together.

**Weaknesses:**

I feel the submission has central weaknesses around the discussion of results and technical contribution. I advocate for rejection but remain open-minded.

I worry that the technical contribution represents a limited increment, combining standard ideas from privacy with the analysis of Zou et al. The paper (pages 2-3) lists as a contribution that "We develop new analytical tools... which can be of independent interest." After reading the paper, I am not sure what these tools are.

I observe two issues with the discussion of results. First are comparisons with the work of Varshney et al. and Liu et al. The abstract says "Our proposed method significantly improves upon existing differentially private methods for linear regression." It does not qualify this statement by saying "in some regimes," so it seems reasonable to interpret this claim as saying the analysis dominates that of prior work. I am not sure this is the case, especially because the analysis here depends poorly on the bound $\lVert w_0 - w^* \rVert_2$ and the magnitude of the initial residuals (via $\ell$).

The second issue with the discussion of results is around lower bounds. The paper claims, for example on page 2, that the algorithm achieves "sharp excess risk for private overparameterized linear regression." Unless I missed it, this submission contains no lower bounds. It appears the discussion of "sharpness" comes from Zou et al., but (i) they do not consider privacy and (ii) they prove that their *analysis* of SGD is sharp. They do not prove a lower bound for the task. I believe it is likely that a casual reader might be misled. It is possible I misunderstand the results and, if so, I would happily change my mind.

**Questions:**

What do you mean when you say that, for example, Corollary 4.3 gives a sharp excess risk bound?

Can you point to and discuss the technical hurdles your analysis had to overcome? What are the analytical tools you developed?

Are there parameter regimes where any of the algorithms of Varshney et al. or Liu et al. have asymptotically lower error than your algorithm with isotropic noise?

Before reading this paper, my impression was that overparameterized linear regression was a theoretical proving ground for understanding implicit bias. Is the problem of overparameterized linear regression (without, for example, sparsity assumptions) of practical interest?

---

> ### Author Response · Authors · 2023-11-19
> **Response 1/2**
>
> We thank the reviewer for their comments and suggestions. We address the reviewer’s comments as follows and hope the reviewer will reconsider their score.
>
> >see 1, 2, 3; these aren't cited but should be
>
> Thanks for suggesting these relevant papers, but we would like to point out that the third paper you mentioned is released on arXiv **after** the ICLR submission deadline. We will add and discuss them in both the introduction section and Section 3.2 in the revision.
>
> >technical hurdles and analytical tools you developed
>
> Firstly, we propose to use DP-FTRL based method instead of the DP-SGD based method  to prevent the need to use shuffling/random sampling to get good dependency on privacy parameters. Additionally, to achieve problem- and algorithm- dependent excess risk bound, we resort on the fact that DP-FTRL can be written as a gradient update (Sec B.2.2) to actually more closely follow the analysis from Zou et al (2021b) instead of following the DP-FTRL based analysis. However, directly applying their analysis will lead to an unfavorable dependence on the problem dimension $d$ when the additive noise is introduced. Therefore, we have developed highly non-trivial new techniques to address this issue and get an excess risk bound that can get rid of the explicit dimension dependence. In the following, we summarize the key technique hurdles in our analysis and the new analytical tools we have developed to address these difficulties:
>
> 1. Although DP-FTRL can be written as a gradient update (Sec B.2.2), the additive noise at each iteration $\mathbf{\xi}$ (see equation (B.4)) is correlated with previous iterations. This is significantly different from DP-SGD, where independent random noise is added at each iteration. Therefore, we cannot simply factor out the additive noise using the independence of the additive noise like the one used in the DP-SGD analysis. This correlation will introduce extra errors (see the second equation under equation (B.14)), which are challenging to be upper bounded and have never been studied in the analysis of Zou et al (2021b). To overcome this difficulty, we propose to characterize the properties of the additive noise generated by the tree aggregation procedure, and develop new proof techniques (see Sec B.4) as well as new results (see Lemma B.12) to characterize these extra error terms (see equation (B.16) and Theorem B.5). These new techniques and results enable us to derive the new bias-variance decomposition expression for each iteration in our proposed DP-FTRL based method (Theorem B.5).
>
>  2. Given the new bias-variance decomposition expression (Theorem B.5) for each iteration, we still cannot directly apply the proof techniques in Zou et al (2021b) to obtain the desired excess risk bound. The reason is that the extra errors introduced by the private mechanism depend on the general covariance matrix $\mathbf{\Sigma}$ and will affect the variance error (see Lemma B.6). As we discussed under Lemma B.6, the proof techniques in Zou et al (2021b) will make the variance error explode since the general covariance matrix $\mathbf{\Sigma}$ will no longer be well aligned with $\mathbf{H}$, a condition crucial for the proofs in in Zou et al (2021b)’s proofs. Therefore, to provide a tight upper bound on the variance error, we derive new results (Lemma B.7 and Lemma B.8) to handle the general covariance matrix $\mathbf{\Sigma}$ case.
>
> In summary, to obtain the excess risk bound of our proposed method, we need to develop new proof techniques that can handle the correlated noise and the variance error introduced by the general covariance matrix of the additive noise. We believe our new analytical tools could be of independent interest due to their effectiveness in addressing the challenges posed by correlated noise and general covariance matrices.

---

> > ### Author Response · Authors · 2023-11-19
> > **Response 2/2**
> >
> > >comparisons with the work of Varshney et al. and Liu et al….
> >
> > First of all, we do not agree with your comment on poorly depend on $||w_0-w^*||_2$. Actually, our results have nothing to do with $\ell_2$ norm bound ($||w_0-w^*||_2$), but only depend on the $H$ norm ($||w_0-w^*||_H$), which is the same as Varshney et al. (2022). Our bound is decomposed into the error in the head and tail subspaces of $H$, which is similar to the bound developed in Zou et al (2021b), Tsigler & Bartlett (2020). The $\ell_2$ norm type bound only appears in the headspace, while the bound on the tail subspace only appears in $H$ norm. Then, noting that the smallest eigenvalue in the headspace can be lower bounded by $1/(N\eta)$, implying that the $\ell_2$ norm bound in the headspace can be further relaxed to $H$ norm bound with an additional multiplication factor $N\eta$. Consequently, our bound is nonvacuous as long as $||w_0-w^*||_H$ is upper bounded, while $||w_0-w^*||_2$ is allowed to be infinite.
> >
> >
> > Secondly, the explicit dependence of our results on the estimated residual $\ell$ comes from our proposed method of adaptively estimating the gradient norm at each iteration.
> > It's important to note that the results in Varshney et al. (2022), Liu et al. (2023) will also have the similar dependence (they also propose to adaptively estimate the gradient norm). However, in their final results, they derive a feasible upper bound for this term by assuming the data distribution has a nice property, i.e., the condition number of the covariance matrix is finite. Our results and those in Varshney et al. (2022), Liu et al. (2023) may be not directly comparable due to these subtleties.
> >
> > Lastly, we would like to clarify that our bound is at least more reasonable and meaningful in the high-dimensional or even infinite-dimensional setting. Our bound, stated as a function directly with respect to the data covariance and noise covariance, can still provide nonvaucous excess risk bound even when $d$ is very large or even infinite (see our Corollary 4.4 and Corollary 4.6). In contrast, the prior bounds in  Varshney et al. (2022), Liu et al. (2023)  have an explicit dependence on $d$, which clearly become vacuous in the high-dimensional or infinite-dimensional case.
> >
> > Therefore, as pointed out by Reviewer jHEC, we will revise our comments about the relationship of our work with that of Varshney et al. (2022), Liu et al. (2023) to ensure a fair placement of our work in the literature. For example, rather than claiming that our method outperforms previous method, we will state that our method provides the first problem- and algorithm- dependent excess risk bound for private linear regression.
> >
> > Given the technique contributions and the fair placement of our work, we believe our work makes significant contributions to the field of private linear regression and private overparameterized learning.
> >
> > > the meaning of sharp and lower bounds
> >
> > The sharp we mean actually our bound could be potentially sharper than existing results because we remove the explicit dependence on the problem dimension.
> > We believe that we can provide a utility lower bound for our algorithm. When the covariance of the additive noise is $\mathbf{\Sigma}=\mathbf{I}$, we believe our upper and lower bounds can be tight up to log factors. In the more general case, there may exist a gap because we may need to handle the non-commute matrices in the lower bound proofs, which may necessitate additional relaxations to address this challenge. The motivation of our paper is to develop a problem-dependent framework to study the private linear regression that can handle general data covariance and general non-isotropic additive noise to achieve DP. Our results can potentially lead to better noise designs for achieving DP and obtain better utility guarantees. Adding the lower bound to our paper is a plus but this will not affect the main contributions of our paper. We would like to add the discussions about the lower bound in the revision.
> >
> > >Is the problem of overparameterized linear regression of practical interest?
> >
> > Overparameterized model is very important and has been widely used in practice. Therefore, how to protect sensitive information in the overparameterized model is a very important and meaningful problem. Recent empirical studies (Li et al. (2022b), Yu et al. (2022), De et al. (2022), Mehta et al. (2022)) have observed that DP-SGD for fine-tuning pre-trained large models can yield promising performance on downstream language and vision tasks. Therefore, we believe that understanding the overparameterized linear model is a starting point for us to further understand the performance of privacy-preserving methods on other more complex overparameterized models. Additionally, our paper can motivate practitioners to design better noise to achieve DP by (privately) considering the statistical property of the data.

---

> > > ### Comment · Reviewer_BWiw · 2023-11-21
> > >
> > > I have read and considered your response. My opinion of the submission has not changed substantially.
> > >
> > > The submission made central claims about the tightness of the results and comparisons to prior work that I feel would likely have misled readers. After the reviews, the rebuttal largely walks back these claims. I find this concerning. I'm not confident that I will agree with the way future versions discuss these results [1].
> > >
> > > I am still concerned that the technical work is only incremental. The rebuttal highlights what doesn't follow from prior work, but I still don't understand the challenges faced (e.g., why are these issues not amenable to existing tools or proof techniques?) or the new analytical tools produced (e.g., what new ideas allowed you to solve these problems?).
> > >
> > > [1] As a primary example, the rebuttal claims the "first problem- and algorithm- dependent excess risk bound for private linear regression." What does this mean? What would it mean for a bound to be algorithm-independent?

---

> > > > ### Author Response · Authors · 2023-11-22
> > > >
> > > > Thanks for your response, we would like to further clarify your concerns as follows.
> > > >
> > > > >why are these issues not amenable to existing tools or proof techniques
> > > >
> > > > As we have highlighted in our previous response: (1) existing tool cannot handle the extra errors introduced by the additive noise in the variance error, and it will lead to an exploding bound; and (2) existing method can only handle the case where the iterates form a markov chain. However, this is not the case in our setting as the additive noises are correlated at different iterations.
> > > >
> > > > >what new ideas allowed you to solve these problems?
> > > >
> > > > To solve (1): we develop new proof techniques,  utilizing a new reference quantity that is also stated as a function of the data covariance and sample size, to control the variance error dynamics, rather than the previous results.
> > > >
> > > > To solve (2): we develop a new technique that can handle their correlations. The key is to maintain a dynamic set that includes the noises correlated with the noise in the current iteration, and then precisely characterize how these correlations affect the convergence.
> > > >
> > > > >As a primary example, the rebuttal claims the "first problem- and algorithm- dependent excess risk bound for private linear regression." What does this mean?
> > > >
> > > > The “first problem- and algorithm- dependent excess risk bound for private linear regression” means that our excess risk bound depends on unique properties of the specific problem (such as data covariance) and the particular training algorithm (one-pass SGD-based algorithm) that allow us to achieve potentially better utility and privacy trade-offs,  thus could lead to sharper results compared to the worst-case guarantees (e.g., the worse problem within a problem set). We will be more specific in the revision.

---

### Official Review · Reviewer_jHEC · 2023-11-02

**Soundness:** 3 good
**Presentation:** 2 fair
**Contribution:** 2 fair
**Rating:** 6
**Confidence:** 3

**Summary:**

This submission proposes algorithms and analysis for differentially private linear regression, with focus on dimension-independent guarantees as to have guarantees in the over-parameterized regime (that is, when the dimension is significantly larger than the number of data points). The authors propose an algorithm based on DP-FTRL with adaptive gradient clipping and Gaussian mechanism with non-identity covariance matrix estimated from public data when available to improve the error. Based on techniques from related work, the authors show that the price we pay for privacy is not too large and show dimension-independent rates for a few distributions of the data.

**Strengths:**

**Interesting combination of recent topics**: The submission joins a few different interesting directions of recent work: work on DP linear regression together with recent efforts of showing "benign overfitting" in the data. Although we should intuitively expect that DP-fying linear regression should not ruin benign overfitting, fleshing out the details is interesting and a contribution up to the standards of most large ML conferences. In fact, by skimming the appendix I could see that there are a few challenges in extending the benign overfitting results to the private case, and the authors do a good job on overcoming these obstacles;

**Interesting use of public data**: One of the contributions of the submission is allowing the use of public data to improve on the Gaussian mechanism by using a covariance matrix that better matches (or, in other words, preconditions) the distribution of the data. Probably the most interesting part is showing how this change can positively affect the dimension-independent rates derived by the end of the submission.

**Weaknesses:**

**Missing context for results in the introduction**: One aspect of the submission that is unfortunate is that in the abstract and introduction the authors claim that their convergence rates "improve on previous work on DP linear regression", even more so when using general covariance matrices in the Gaussian mechanism. However, these claims should be toned down a bit since there is some context missing. For example:
- (Major problem) the covariance matrix used in the Gaussian mechanism to achieve the best rates requires access to (unlabeled) public data. This is not ever mentioned in the abstract or introduction. As far as one can tell, the best rates stated in the introduction are achieved only based on the original data, but this is not the case and definitely should be explicitly mentioned in the abstract and in the introduction;
- (Minor problem) Although it is true that the authors improve on rates of previous work, it seems to me that there is some nuance missing. For example, the work of Liu et al. (2023) had as a goal to have algorithms that are robust to a small portion of adversarial corruptions, right? Moreover, if I am not mistake both Varshney et al. (2022) and Liu et al. (2023) assume less about the data distribution (maybe except for Theorem 4.2, but in that case it is not clear if one can compare the bounds in general). Finally, I believe it is true that the constants hidden make it so that beyond requiring $d > N$, one needs $N$ to be significantly large for the rates to actually improve on the ones with dependency on $d$. I do not think these are points that invalidate the contribution of the submission, but I do believe they should be discussed for a fair placement of the contributions of the submission in the literature.

**Apparently incremental work**: Although the results are interesting, they are somewhat incremental since they rely quite heavily on previous results and it is not clear what are the extra technical difficulties. In fact, I believe a great way to make the submission much stronger is to better delineate what are the technical challenges when trying to combine the several techniques the authors use. More specifically, if I understood it correctly, the main contribution is extending the benign overfitting rates from Zou et al. (2021b) to the private setting by "practically" using DP-SGD with the adaptive gradient clipping also used in Liu et al. (2023). I say practically since the authors instead use DP-FTRL (Kairouz et al., 2021) to prevent the need to use shuffling/random sampling to get good dependency on privacy parameters, but they resort on the fact that DP-FTRL can be written as a gradient update (Sec B.2.2) to actually more closely follow the analysis from Zou et al (2021b). So it becomes hard to separate what are the technical difficulties that the authors needed to overcome to make the analysis work well.

A great example if the beginning of the proof of Theorem 4.1 in Sec B.1. The first part of the algorithm is only to show that tree aggregation is private. However, Tree Aggregation to privately release partial sums is already well studied and it is not clear why the analysis needs to be redone. The use of a general $\Sigma$ matrix in the Gaussian mechanism should not warrant, I believe, an entire new analysis of tree aggregation: adding noise with distribution $\mathcal{N}(0, \Sigma)$ to $x$ is equivalent to adding noise with distribution $\mathcal{N}(0, I)$ to $\Sigma^{-1/2} x$ and then post-processing by multiplying everthing by $\Sigma^{1/2}$. This confusion extends to other parts of the analysis since I cannot untangle how much follows from previous work and what are the key technical insights given by the authors. I believe the discussion period will be a great opportunity for the authors to elucidate a bit some of this confusion (which might be simply due to a lack of background from my part or limited time to dedicate to the paper).

**Questions:**

Ultimately, I believe this paper is mostly correct and that its contribution is interesting to the DP community. Yet, I believe better placing the contributions in the literature (including stating use or not of public data) and clarifying the key technical contributions would strengthen the submission by an order of magnitude. Removing the burden of untangling the key technical contributions from previous work and the key technical challenges overcome by the current paper would certainly make us appreciate the results a lot more. Here are a few points I hope the authors can help me get some clarity on:

1) Do you believe my assessment on the minor problem I mentioned on the lack of context is fair? Feel free to disagree, but even if I am wrong, this already tells that the authors should try to clarify the placement of their contributions in the literature.
2) On my second part, would the authors be able to briefly mention some of the technical challenges they had to overcome? I believe the analysis is not simply following the analysis of benign overfitting from 2021 modulo carrying extra terms due to privacy, but from the appendix it is hard to untangle the key technical contributions (as mentioned before, for example, I am not sure the privacy analysis of tree aggregation needs to be proven again). The tree aggregation example might be a good point to clarify whether a new analysis was actually required or not.
3) (Minor) A more technical question: Theorem 4.2 specifies a few constants that depend on $\mathrm{Tr}(H)$, which we assume we do not know, right? Is it standard to have the algorithm parameters in DP linear regression depend on the unknown matrix $H$?
4) (Minor) This is certainly a minor question, but the authors assume that $\mathrm{Tr}(H)$ is no bigger than a constant, if I understand correctly. Although this seems to hold for the

Finally, here are direct suggestions that do not necessarily require comments from the authors (I tried to sort them from more important to less important):
- I think it is paramount that the authors make it clear that the tighter convergence rates depend on a matrix estimated from **public data**. This should not be left to be mentioned only half-way through the paper;
- I believe the text neves defines explitly from which matrix the eigenvalues $\lambda_i$ mentioned are from, but this notion is used throughout the entire text (except for a parenthesis in page 7). This should be clearly stated when you introduce the matrix $H$, since this is extremely important for the readers to understand many of the results;
- At some point the authors call FTRL "follow-the-perturbed-leader" but it should be "follow-the-regularized-leader". The former is a term for a very different algorithm;
- Theorem 4.2 is not very readable in its current form, and uses a lot of space in the main paper. Either a discussion should be added on the relevance/meaning of some of these terms, and/or the statement should be simplified. Using space in the main paper with so much notation that does not communicate much would probably be better off being moved to the appendix. Of course, the authors should feel free to disagree with me on this point, but the current discussion after the theorem does not seem to require all the notation used;
- Right after Assumption 2.2 the text references Assumption 2.3, but I think it should reference Assumption 2.2;
- Define a few of the terms you use early on. It takes a few paragraphs for the text to explicitly say that overparameterized setting simply means $N < d$. I am not sure what "sharp risk" means in the bold text in the introduction.

---

> ### Author Response · Authors · 2023-11-19
> **Response 1/3**
>
> We thank the reviewer for their careful and detailed comments, as well as the insightful suggestions. We address the reviewer’s concerns as follows.
>
> >(Major problem) access to (unlabeled) public data is not ever mentioned in the abstract or introduction.
>
> We indeed mention in the abstract that “Furthermore, when **unlabeled public data** is available, we can design a better noise covariance matrix structure to improve the utility. ” We will also add comments about using unlabeled public data in the introduction section when we state our main contributions (the second bullet).
>
> >(Minor problem) a fair placement of the contributions of the submission in the literature
>
> We agree with the reviewer that we need to have a fair placement of our work in the literature. It is true that our results and the results in Varshney et al. (2022), Liu et al. (2023) may be not directly comparable. For example, our results have an explicit dependence on the estimated residual $\ell$ while Varshney et al. (2022), Liu et al. (2023) get rid of this dependence by establishing a feasible upper bound for this term, assuming that the data distribution has a nice property, i.e., the condition number of the data covariance matrix is finite.
>
> However, we would also like to clarify that our results are at least more reasonable and meaningful in the high-dimensional or even infinite-dimensional setting. Our results, stated as a function directly with respect to the data covariance and noise covariance, can still provide nonvaucous excess risk bounds even when $d$ is very large or even infinite (see our Corollary 4.4 and Corollary 4.6). In contrast, the prior bounds in  Varshney et al. (2022), Liu et al. (2023)  have an explicit dependence on $d$, which clearly become vacuous in the high-dimensional or infinite-dimensional case. Therefore, we will revise our comments about the relationship of our work with that of Varshney et al. (2022), Liu et al. (2023) to ensure a fair placement of our work in the literature. For example, rather than claiming that our method outperforms previous method, we will state that our method provides the first problem- and algorithm- dependent excess risk bound for private linear regression.

---

> > ### Author Response · Authors · 2023-11-19
> > **Response 2/3**
> >
> > >technical difficulties and analytical tools
> >
> > As you said, the reason that we propose to use DP-FTRL based method instead of the DP-SGD based method is to prevent the need to use shuffling/random sampling to get good dependency on privacy parameters. The main reason that we do not follow the DP-FTRL based analysis is that it cannot ensure us to achieve problem- and algorithm- dependent excess risk bound. Therefore, to achieve the desired excess risk bound, we instead resort on the fact that DP-FTRL can be written as a gradient update (Sec B.2.2) to actually more closely follow the analysis from Zou et al (2021b).  However, directly applying their analysis will lead to an unfavorable dependence on the problem dimension $d$ when the additive noise is introduced. Therefore, we have developed highly non-trivial new techniques to address this issue and get an excess risk bound that can get rid of the explicit dimension dependence. In the following, we summarize the key technique difficulties in our analysis and the new analytical tools we have developed to address these difficulties:
> >
> > 1. Although DP-FTRL can be written as a gradient update (Sec B.2.2), the additive noise at each iteration $\mathbf{\xi}$ (see equation (B.4)) is correlated with previous iterations. This is significantly different from DP-SGD, where independent random noise is added at each iteration. Therefore, we cannot simply factor out the additive noise using the independence of the additive noise like the one used in the DP-SGD analysis. This correlation will introduce extra errors (see the second equation under equation (B.14)), which are challenging to be upper bounded and have never been studied in the analysis of Zou et al (2021b). To overcome this difficulty, we propose to characterize the properties of the additive noise generated by the tree aggregation procedure, and develop new proof techniques (see Sec B.4) as well as new results (see Lemma B.12) to characterize these extra error terms (see equation (B.16) and Theorem B.5). These new techniques and results enable us to derive the new bias-variance decomposition expression for each iteration in our proposed DP-FTRL based method (Theorem B.5).
> >
> >  2. Given the new bias-variance decomposition expression (Theorem B.5) for each iteration, we still cannot directly apply the proof techniques in Zou et al (2021b) to obtain the desired excess risk bound. The reason is that the extra errors introduced by the private mechanism depend on the general covariance matrix $\mathbf{\Sigma}$ and will affect the variance error (see Lemma B.6). As we discussed under Lemma B.6, the proof techniques in Zou et al (2021b) will make the variance error explode since the general covariance matrix $\mathbf{\Sigma}$ will no longer be well aligned with $\mathbf{H}$, a condition crucial for the proofs in in Zou et al (2021b)’s proofs. Therefore, to provide a tight upper bound on the variance error, we derive new results (Lemma B.7 and Lemma B.8) to handle the general covariance matrix $\mathbf{\Sigma}$ case.
> >
> > In summary, to obtain the excess risk bound of our proposed method, we need to develop new proof techniques that can handle the correlated noise and the variance error introduced by the general covariance matrix of the additive noise. We believe our new analytical tools could be of independent interest due to their effectiveness in addressing the challenges posed by correlated noise and general covariance matrices.
> >
> > >why the analysis of tree aggregation needs to be redone.
> >
> > It's important to note that the tree aggregation protocol used in our method is slightly different from those used in prior work (e.g., Guha Thakurta & Smith (2013),  Kairouz et al. (2021)).
> > Specifically, our method proposes to add different amounts of noise to each node in the tree (see line 7 in Algorithm 1 and line 11 in Algorithm 3) while the previous work proposes to add the same amount of noise to each node in the tree. This is due to the adaptive clipping step (see line 6 in Algorithm 1) used in our method, leading to different norm bounds for different nodes. Therefore, we cannot directly apply existing results (such as those in Guha Thakurta & Smith (2013),  Kairouz et al. (2021) ) to provide the privacy guarantee of our tree aggregation protocol. We will clarify this in the revision.

---

> > > ### Author Response · Authors · 2023-11-19
> > > **Response 3/3**
> > >
> > > >Is it standard to have the algorithm parameters in DP linear regression depend on the unknown matrix
> > >
> > > It is very standard in the DP literature to set algorithm parameters to some unknown values in order to derive utility guarantees (e.g., Varshney et al. (2022), Liu et al. (2023)) as long as the privacy guarantee (i.e., Theorem 4.1 in our paper) does not depend on any unknown values.
> > >
> > > >assume that $\mathrm{Tr}(\mathbf{H})$ is no bigger than a constant
> > >
> > > We can also present our results in terms of Tr(H). The reason that we choose to assume $\mathrm{Tr}(\mathbf{H})$ to be a constant is to simplify our final results, especially in the polynomial and exponential decay settings. Moreover, considering bounded $\mathrm{Tr}(\mathbf{H})$ is the necessary condition to enable the analysis in the infinity dimensional case.

---

> > > > ### Comment · Reviewer_jHEC · 2023-11-22
> > > > **Great response, common points among reviews and current opinion**
> > > >
> > > > I would like to thank the authors for the thorough and careful replies to all the reviewers concerns. Also, I apologize for taking some time to reply to the author's response. Although this period is meant as a discussion period, I did not have any burning questions after skimming through the replies, and to write a meaningful reply I needed to spend time reading the long replies (not only to my review but to the other reviews as well).
> > > >
> > > > It seems that the concerns about fair placement of the contributions of the paper in the literature and lack of clarity on the technical novelty are concerns that most/all of the reviewers had, including me. So I will try to summarize where I stand on these points after reading the other reviews and the responses.
> > > >
> > > > *Fair placement of the contributions of the paper*: On part 1 of the response to my review you do a great job on better comparing your work to the ones of Varshney et al. (2022) and Liu et al. (2023), and a polished version of this certainly deserved to be in the paper. I'd just be a bit careful with wording. In the reply you say "(...) our results are at least more reasonable and meaningful in the high-dimensional or even infinite-dimensional setting". I understand what you meant, but I'd add the caveat that this is true *asymptotically in $d$ and $n$*. I do not know enough from these other works to be able to state whether your claim is actually true or not, but it seems somewhat likely that the dimension-dependent bounds from other work might have smaller constants hidden in the big-Oh notation. Although this does not affect the relevance of the results in the asymptotic case, it warrants the fair qualifier of the claim that maybe $d$ and $n$ need to be quite big for the dimension-independent bounds to be better. Moreover, the paper always seems to assume we are dealing with finite $d$, and making the jump to infinite dimensions certainly deserves justification. I would not claim the results hold in infinite dimensions unless you explicitly dedicate time discussing why it is not a problem to handle infinite-dimensional spaces (and even how this would look like)
> > > >
> > > > Again, this is somewhat of a suggestion on how to frame your contributions. From what I can gather from the other reviewers, it seems that most of us felt that some of the claim deserved to be toned down and better contextualized. I guess that, in summary, your results do not need to be the first or best in class to be great. I still believe these results are interesting, and better contextualizing them would make readers better appreciate your work!
> > > >
> > > > *Technical difficulties*: I truly believe the authors did a wonderful job in the responses on highlighting some of the technical difficulties. Indeed, we can see that in parts of the appendix (such as after Lemma B.6) this is already done, but it is hard for the reviewers of a conference to go through the appendix, I hope the authors can understand this. Probably a summary of these points (as done in these responses, but more polished) in the main paper would make it clearer what are the difficulties when extending the benign overfitting results to your setting. Again, I'd be careful with some of the claims. For example, when you say "We believe our new analytical tools could be of independent interest due to their effectiveness in addressing the challenges posed by correlated noise and general covariance matrices." This might be true, but it is hard to pin down what are these tools since, as written, these tools seem to be very specific to the setting you are used. And in fact cleaning up the results to extract this tools for general use might be another paper, and I don't think these techniques need to be of independent interest for the readers to appreciate the work done by the authors. As in the previous point, be careful with the claims, and do not worry about claiming things that are actually not absolutely clear from your work.
> > > >
> > > > As a related minor note, the clarification on why you need to analyze tree aggregation again is interesting. I do not have enough knowledge of the literature to verify the claims, but it seems reasonable and explains the need of a new analysis.
> > > >
> > > > Finally, on a few of the other points:
> > > > - *Use of public data*: Thanks for pointing out that you do mention use of public data in the abstract. But this does not alleviate the problem of it not being mentioned beyond that point until sec 4. This certainly worsened the aspect of the paper "overclaiming" that most/all reviewers complained about.
> > > > - *Clarifications on dependence on problem parameters*: Thanks for clarifying to me how reasonable it is to have the dependency on problem parameters.
> > > >
> > > > (...continues in the next response)

---

> ### Comment · Reviewer_jHEC · 2023-11-22
> **Continuing**
>
> - *Meaning of sharp*: This was yet another confusion with the claims that reviewer BWiw mentioned and I have to agree that I was confused as well. I think the authors do a good job explaining what you meant by sharp, but as you could see, "sharp" seems to be a bit of a overloaded term and led to a lot of confusion, and it does not seem it was a good use of the term in the paper.
>
> - *Dependence on $\lVert w_0 - w^* \rVert$*: I saw the author's response to reviewer BWiw on the dependence on the norm dependence, I think this was an interesting discussion that I had completely missed on my first reading of the paper. I cannot quite see how the value of this norm can be infinite (maybe you meant "go to infinity with $d$"?), but the difference between depending on $|| w_0 - w^* ||_2 $ and on $|| w_0 - w^* ||_H$ seems worth discussing. I got a bit lost on the wording used (headspace and tailspace have very loose meanings). So this is a very minor suggestion and take it with a grain of salt, but it seems to be an interesting point of discussion and maybe it deserves a more careful discussion than the reply we had for BWiw.
>
> - *Practical interest*: This is me commenting on BWiw's comment asking whether this is of practical interest, since the main focus of this paper seems to be theoretical. I'd advise the authors to not try to push too much trying to justify your results in practical terms since its biggest value seems to be theoretical, which is perfectly fine.

---

> > ### Comment · Reviewer_jHEC · 2023-11-22
> > **Concluding thoughts**
> >
> > So in conclusion, the other reviews does make me believe more strongly that the results are incremental (an extension of Zou et al. (2021b) with a combination of results from Liu et al. (2023) with some non-trivial steps to extend the results), but still interesting and worht publishing. In fact, we should be careful to not take "incremental" as a demeaning property. Modulo the claim being properly contextualized and accurately discusses, I do believe these results are of interest to the DP community.
> >
> > I agree with BWiw that, in its current state, the paper can mislead some readers, so the revisions suggested are paramount.

---

> > > ### Author Response · Authors · 2023-11-23
> > >
> > > Thank you so much for your detailed comments and suggestions. We will clarify fair placement of the contributions of the paper in the literature and the technical novelty in the revision.

---

### Official Review · Reviewer_mnfV · 2023-11-10

**Soundness:** 3 good
**Presentation:** 3 good
**Contribution:** 2 fair
**Rating:** 3
**Confidence:** 5

**Summary:**

The paper proposed a new variant of the DP-FTRL algorithm with a novel noise, a general covariance matrix instead of traditional identical matrix. Specifically, the tree aggregation protocol and designed noise matrix are employed to obtain better trade-offs between privacy and utility; The residual estimator is to use the private empirical variance estimator of the residual to control the gradient.

**Strengths:**

- The paper introduces a novel designed noise covariance matrix, which could provide improved privacy and utility trade-offs compared to using isotropic noise.
- The paper is well-organized and the use of pseudocode for the algorithms helps in understanding the technical details.

**Weaknesses:**

- The proposed designed noise covariance matrix is also limited and may be unpractical since it needs the unlabeled data first to generate an estimation of $\mathbf{H}$.
- The statement of Algorithm 3 is not clear. Binary representation $b_i$ is not clear in line 13. Also, Line 7 in algorithm 3 shows "$n_j$ is the last node in $\mathbf{p}$ that is a left child" and add noise when $n_i \in \mathbf{p}_j$, while Page 15 states "adding Gaussian noise to the nodes along this path from m to the first left child", which is contradict. $\mathbf{p}_j$ is a set and first left child is a node. Hope the author give more explanations.
- typo: Page 5, update rule should be \sum_{i=0}^t g_i instead of \sum_{j=0}^t.

**Questions:**

-The authors should add experiments to support their theory
-The authors should show the disadvantage of DP-AMBSSGD: DP-Adaptive-Mini-Batch-Shuffled-SGD, I believe it can also handle the overparameterized case
-The author should add lower bounds

---

> ### Author Response · Authors · 2023-11-19
>
> We thank the reviewer for their comments, and we address the reviewer's concerns as follows.
> >The proposed designed noise covariance matrix is also limited and may be unpractical since it needs the unlabeled data first to generate an estimation of $\mathbf{H}$
>
> The idea of using public data in the DP methods has been widely used in the literature, e.g., Kairouz et al. (2020), Yu et al. (2022), and Zhou et al. (2021). In addition, getting unlabeled public data is generally cheaper than labeled training data. Therefore, we do not think the proposed designed noise covariance matrix is limited and impractical. Additionally, if we do not have a public dataset, we can use the isotropic noise $\mathbf{\Sigma}=\mathbf{I}$. In this case, we also develop an problem- and algorithm- dependent analysis for the excess risk bound (see Corollary 4.3, Corollary 4.4), which are stated as the entire data covariance matrix instead of the problem dimension, thus can be potentially better than the existing works (Varshney et al. (2022), Liu et al. (2023)) that rely on the dimension explicitly.
>
> Moreover, we also want to emphasize that our excess risk bound is general, and can be employed for any noise covariance. Therefore, one can definitely seek to develop private algorithms using part of the training data points to estimate $\mathbf{H}$ (regardless of its estimation error) and then apply our general theoretical framework to get the excess risk bound accordingly. In fact,  as long as the estimate can well approximate the large eigenvalue of $\mathbf{H}$ (i.e., the estimate is better than $\mathbf{I}$ in terms of estimating $\mathbf{H}$), it will result in better utility guarantees compared to that obtained by using isotropic noise. We will add the discussion in the revision.
>
> > Binary representation is not clear in line 13. The contradiction between Line 7 and page 15
>
> The binary representation mentioned in line 13 refers to expressing the number  $t$ as a binary number using $k$ bits, i.e., $\\{b_1,....,b_k\\}$, where $b_1$ is the most significant bit. We will add more explanations about the binary representation in the revision. There is no contradiction between line 7 in Algorithm 3 and Page 15. The set $\mathbf{p}_j$ stores nodes in a top-down sequence, from the root to the leaf. Since the first node $n_j$ stored in $\mathbf{p}_j$ is the last left child node in $\mathbf{p}$ (top-down order), it is equivalent to state that $\mathbf{p}_j$ stores the nodes from the leaf node $m_t$ to the first left child, i.e., $n_j$, along the path (down-top order). We will clarify this in the revision.
>
> > typo
>
> Thanks for pointing out this typo, we will fix it in the revision.
>
> >Comparison to DP-Adaptive-Mini-Batch-Shuffled-SGD
>
> We would like to emphasize that the main contributions of our work lie at the theoretical side. We theoretically show our method can achieve the problem- and algorithm- dependent excess risks for private linear regression, which is directly stated as the data covariance and noise covariance, rather than the problem dimension. Therefore, our bound can still be reasonable and meaningful even in the infinite dimension case. This is a distinct departure from existing results, such as those presented by Varshney et al. (2022) and Liu et al. (2023), which are dependent on the problem dimension. Clearly, those bounds will be vacuous when $d$ is large.
> Additionally, according to the theoretical guarantee of DP-Adaptive-Mini-Batch-Shuffled-SGD, it cannot handle the overparameterized case.
>
> >The author should add lower bounds
>
> We believe that it is possible to have an algorithm-dependent lower bound that is also stated as a function of both data covariance and the covariance of the additive noise. When the covariance of the additive noise is $\mathbf{\Sigma}=\mathbf{I}$, we believe our upper and lower bounds can be tight up to logarithmic factors. In the more general case, there may exist a gap because we may need to handle the non-commute matrices  in the lower bound proofs, which may necessitate additional relaxations to address this challenge. The motivation of our paper is to develop a problem-dependent framework to study the private linear regression that can handle general data covariance and general non-isotropic additive noise to achieve DP. Our results can potentially lead to better noise designs for achieving DP and obtain better utility guarantees. Adding the lower bound to our paper is a plus but this will not affect the main contributions of our paper. We would like to add the discussions about the lower bound in the revision.

---

> ### Comment · Reviewer_mnfV · 2023-11-22
>
> Thanks for your response, but I still find there are many issues.
> (1) Yes, private learning with public data is a common assumption. However, all the papers you mentioned are for DP-ERM rather than DP linear regression. I believe it is possible to get similar results even without public data for overparameterized linear regression. For example https://arxiv.org/abs/2206.01836. So the assumption in the paper is unacceptable unless the authors could show a lower bound.
>
> (2) Even for the original DP linear regression paper, we can also get a similar upper bound for the overparameterized case, please carefully read the paper https://arxiv.org/pdf/2207.04686.pdf, for example, page 41, their bound only depends on the trace of the Hessian so it is possible to get a similar bound.
>
> (3) No experiments. In the non-private case, all previous papers on overparameterized linear regression have experimental results. However, this paper does not have any experimental study.
>
> (4) The proof high depends on the previous studies on overparameterized linear regression. I cannot see any  difficulty in the proof.
>
> So I will not change the score unless the author could show it is impossible to achieve similar results without public data.

---

> ### Author Response · Authors · 2023-11-22
>
> Thanks for your response, I believe the reviewer's concerns mainly come from the misunderstanding of our results and other results in the literature, and we would like to clarify it as follows.
>
> >(1) Yes, private learning with public data is a common assumption. However, all the papers you mentioned are for DP-ERM rather than DP linear regression. I believe it is possible to get similar results even without public data for overparameterized linear regression.
>
> We have **clearly** state in our paper that we can get similar results **without using any public data** by adding random noise with the identity matrix (see Corollary 4.3 and Corollary 4.4). The reason that we propose to use public unlabeled data is to show that the excess risk bounds can be further improved. In addition, since our developed analytical framework can handle the additive noise with general covariance matrices, we can also use the DP approximation of $H$ to get the result. The detailed comparisons between our method with and without unlabeled public data are illustrated in Table 1. We have discussed the paper you mentioned in detail in the related work section. Note that their results cannot be applied to linear regression since they assume the per-example objective loss to be Lipschitz everywhere.
>
> >(2) Even for the original DP linear regression paper, we can also get a similar upper bound for the overparameterized case, please carefully read the paper. https://arxiv.org/pdf/2207.04686.pdf, for example, page 41, their bound only depends on the trace of the Hessian so it is possible to get a similar bound.
>
> It is **obvious** that the results in the page 41 of https://arxiv.org/pdf/2207.04686.pdf will lead to an exploding bound in the overparameterized case. To be more specific, their results depend on terms $\mathrm{Tr}(H^{-1})$ and $\mathrm{Tr}(H^{-1}\Sigma)$ ($H$ may have very small eigenvalues, e.g., polynomial or exponential, and $\Sigma$  resembles $H$), which will definitely lead to much worse results or even exploding results in the overparameterized case. Therefore, the original DP linear regression paper **cannot** get similar upper bounds for the overparameterized case.
>
> >(3) No experiments. In the non-private case, all previous papers on overparameterized linear regression have experimental results. However, this paper does not have any experimental study.
>
> Our paper’s main contribution is on the theoretical side. However, we would like to add experiments in the revision.
>
> >(4) The proof high depends on the previous studies on overparameterized linear regression. I cannot see any difficulty in the proof.
>
> As we have mentioned in the response to Reviewer jHEC and Reviewer BWiw, we have summarized the Technical difficulties as well as new proof techniques. Previous techniques cannot handle these difficulties in our problem and we develop highly non-trivial solutions to address them.  Here we present a brief summary of these difficulties and new proof techniques.
>
> Technical difficulties:
> 1. Existing tool cannot handle the extra errors introduced by the additive noise in the variance error, and it will lead to an exploding bound.
>
> 2. Existing method can only handle the case where the iterates form a markov chain. However, this is not the case in our setting as the additive noises are correlated at different iterations.
>
> New proof techniques:
> 1. We develop new proof techniques, utilizing a new reference quantity that is also stated as a function of the data covariance and sample size, to control the variance error dynamics, rather than the previous results.
>
> 2. We develop a new technique that can handle their correlations. The key is to maintain a dynamic set that includes the noises correlated with the noise in the current iteration, and then precisely characterize how these correlations affect the convergence.

---

> > ### Comment · Reviewer_mnfV · 2023-11-22
> >
> > For the case where there is no private data, I do not think you are right. Since you assume x is sub-Gaussian, that is your noise added to the covariance matrix is proportional to d and this is unavoidable. That means your noise scale will be extremely large, then how you can privatize the covariance matrix without affecting your final result?
> >
> > For the theoretical analysis, unfortunately, I cannot agree with that. I checked carefully, it's idea is very similar to Zou et al and previous linear regression paper although here you used the tree mechanism instead. The idea of the proof is similar.

---

> ### Author Response · Authors · 2023-11-23
>
> Thanks for your response, we would like to further address your questions as follows.
>
> >For the case where there is no private data, I do not think you are right. Since you assume x is sub-Gaussian, that is your noise added to the covariance matrix is proportional to d and this is unavoidable.
>
> Our results depend on the data covariance and the covariance of the additive noise (see Theorem 4.2), and this is key that we can derive our results. Even if you add random noise with an identity matrix you can still benefit from good data covariance, as the excess risk is characterized via $H$-norm and $\\|\epsilon\\|_H\sim \mathrm{Tr}(H)<< d$ when $\epsilon$ is generated from isotropic Gaussian. And this is why we say that we derive the problem-dependent excess risk.
> Similar results have been established in the literature for other problems (Song et al., 2021; Ma et al., 2022; Li et al., 2022a) using the random noise with an identity matrix (these results cannot be applied to private linear regression).
>
> >That means your noise scale will be extremely large, then how you can privatize the covariance matrix without affecting your final result?
>
> When we add isotropic additive noise, we don’t need to privatize the covariance matrix or use public data.
>
> >For the theoretical analysis, unfortunately, I cannot agree with that. I checked carefully, it's idea is very similar to Zou et al and previous linear regression paper although here you used the tree mechanism instead. The idea of the proof is similar.
>
> We disagree with your comments. Note that most analysis of differentially private algorithms will somehow rely on the analysis of the underlying non-private method (e.g., DP-SGD), and that is why our analysis looks similar to the analysis in Zou et al. Note that even in the previous work (Varshney et al.), it also relies on the underlying non-private analysis (bias-variance decomposition analysis proposed in [1] ).
>
> Reference:
>
> 1. Jain et al. A markov chain theory approach to characterizing the minimax optimality of stochastic gradient descent (for least squares).  In FSTTCS, 2017

---

### Author Response · Authors · 2023-11-21

Dear Reviewers,

Thank you so much for your time and efforts in reviewing our paper! We have provided detailed responses to your comments in the threads below. As the reviewer-author discussion period is ending soon, we would be grateful to receive your feedback on whether our responses have adequately addressed your concerns. If you have any further questions, we are eager and more than happy to address them. If not, we would greatly appreciate it if you could kindly reconsider your scores.

Thanks!

Best regards, Authors of Paper 7718

---

### Meta-Review · Area_Chair_5YGB · 2023-12-08

**Metareview:**

Unfortunately, the reviewers were not enthusiastic about the paper. In this case, I would side with the reviewer's decision since most of the reviewers for the paper are experts in the area. There were two major concerns: i) Technical novelty, and ii) correctness of the claims made in the paper (see the comments from Bwiw). We would recommend the authors to update the paper after carefully taking into account the reviewer concerns.

**Justification For Why Not Higher Score:**

None of the reviewers were in strongly favor of the paper, and there were concerns about the claims made which remained unresolved.

**Justification For Why Not Lower Score:**

NA

---

### Decision · Program_Chairs · 2024-01-16

Reject